# A neuronal least-action principle for real-time learning in cortical circuits

**Walter Senn[1]\*[†], Dominik Dold[1,2,3†], Akos F Kungl[1,2], Benjamin Ellenberger[1,4], Jakob Jordan[1,5], Yoshua Bengio[6], João Sacramento[7], Mihai A Petrovici[1,2†]**

[1]Department of Physiology, University of Bern, Bern, Switzerland; [2]Kirchhoff-Institute for Physics, Heidelberg University, Heidelberg, Germany; [3]European Space Research and Technology Centre, European Space Agency, Noordwijk, Netherlands; [4]Insel Data Science Center, University Hospital Bern, Bern, Switzerland; [5]Electrical Engineering, Yale University, New Haven, United States; [6]MILA, University of Montreal, Montreal, Canada; [7]Department of Computer Science, ETH Zurich, Zurich, Switzerland

**\*For correspondence:**
walter.senn@unibe.ch

[†]These authors contributed equally to this work

**Competing interest:** The authors declare that no competing interests exist.

## eLife assessment

This manuscript describes a potentially **important** theoretical framework to link predictive coding, error-based learning, and neuronal dynamics. The provided evidence is **solid**, but some details would benefit from additional clarification. The exposition of the manuscript is targeted for a specialist audience.

**Abstract** One of the most fundamental laws of physics is the principle of least action. Motivated by its predictive power, we introduce a neuronal least-action principle for cortical processing of sensory streams to produce appropriate behavioral outputs in real time. The principle postulates that the voltage dynamics of cortical pyramidal neurons prospectively minimizes the local somato-dendritic mismatch error within individual neurons. For output neurons, the principle implies minimizing an instantaneous behavioral error. For deep network neurons, it implies the prospective firing to overcome integration delays and correct for possible output errors right in time. The neuron-specific errors are extracted in the apical dendrites of pyramidal neurons through a cortical microcircuit that tries to explain away the feedback from the periphery, and correct the trajectory on the fly. Any motor output is in a moving equilibrium with the sensory input and the motor feedback during the ongoing sensory-motor transform. Online synaptic plasticity reduces the somatodendritic mismatch error within each cortical neuron and performs gradient descent on the output cost at any moment in time. The neuronal least-action principle offers an axiomatic framework to derive local neuronal and synaptic laws for global real-time computation and learning in the brain.

## Introduction

Wigner's remark about the 'unreasonable effectiveness' of mathematics in allowing us to understand physical phenomena *Wigner, 1960* is famously contrasted by Gelfand's quip about its 'unreasonable ineffectiveness' in doing the same for biology (*Borovik, 2021*). Considering the component of randomness that is inherent to evolution, this may not be all that surprising. However, while this argument holds just as well for the brain at the cellular level, ultimately brains are computing devices. At the level of computation, machine learning, and neuroscience have revealed near-optimal strategies for information processing and storage, and evolution is likely to have found similar principles through trial and error (*Hassabis et al., 2017*). Thus, we have reason to hope for the existence of fundamental

principles of cortical computation that are similar to those we have found in the physical sciences. Eventually, it is important for such approaches to relate these principles back to brain phenomenology and connect function to structure and dynamics.

In physics, a fundamental measure of 'effort' is the action of a system, which nature seeks to 'minimize.' Given an appropriate description of interactions between the system's constituents, the least-action principle can be used to derive the equations of motion of any physical system (*Feynman et al., 2011*; *Coopersmith, 2017*). Here, we suggest that in biological information processing, a similar principle holds for prediction errors, which are of obvious relevance for cognition and behavior.

Based on such errors, we formulate a neuronal least-action (NLA) principle which can be used to derive neuronal dynamics and map them to observed dendritic morphologies and cortical microcircuits. Within this framework, local synaptic plasticity at basal and apical dendrites can be derived by stochastic gradient descent on errors. The errors that are minimized refer to the errors in output neurons that are typically thought to represent motor trajectories, planned and encoded in cortical motor areas and ultimately in the spinal cord and muscles. In the context of motor control, a phenomenological 'minimal action principle' has previously been proposed that guides the planning and execution of movements (*Feldman and Levin, 2009*). Our neuronal least-action principle reformulates and formalizes the classical equilibrium point hypothesis (*Latash, 2010*) in a dynamical setting, linking it to optimality principles in sensory-motor control (*Todorov, 2004*).

Other attempts exist to link biological information processing and neural networks with the least-action principle, for instance by directly learning to reproduce a given trajectory (*Amirikian and Lukashin, 1992*), by minimizing the physical action for the muscle force generation by motor unit recruitment (*Senn et al., 1995*), minimizing cognitive prediction errors (*Alonso et al., 2012*), minimizing output errors with a weight-change regularization (*Betti and Gori, 2016*), minimizing psychomotor work (*Fox and Kotelba, 2018*), minimizing data transport through a network (*Karkar et al., 2021*), minimizing the discrimination information (*Summers, 2021*), or minimizing the free energy (*Friston, 2010*; *Friston et al., 2022*). Apart from the latter, however, these attempts remain far from the biology that seems to resist a formalization with the tool of physics – at least, when applied too strictly.

The fundamental novelty of our NLA principle is the way it deals with time. In physics, bodies interact based on where they are now, irrespective of what happens in the future. Living systems, instead, interact based on what could happen in the near future, and react early to stay alive. This difference is also mirrored in the way our NLA principle looks for an error-minimizing trajectory of brain states. We postulate that the brain trades with near-future states and seeks for a path that minimizes errors of these future states at any moment in time. Looking ahead towards what will likely happen allows the network for correcting the internal trajectory of deep neurons early enough so that the delayed output moves along the desired path. The notion of looking into the future to gate a dynamical system is also central in optimal control theory (as expressed by the Bellman equation, see e.g. *Todorov, 2006*). Yet, starting with a neuronal action is more principled as it includes the derivation of the dynamical system itself that will be optimally controlled.

The insight into the time structure of biological information processing allows us to express a simple form of a total 'mismatch energy' for our cortical neuronal networks, from which we derive the dynamic neuronal and synaptic laws.In short, the mismatch energy within a single pyramidal neuron is the squared prediction error between basal dendrites and the soma, together with the apical dendrites receiving a top-down feedback. The apical dendrites calculate a local prospective prediction error that looks ahead in time and overcomes neuronal integration delays (*Figure 1a*). As a consequence, the output neurons are corrected on the fly by the prospective error processing, pushing them in real time closer to the desired path. In addition, the prospective errors are suited for gradient learning of the sensory synapses on the basal dendrites. This gradient learning is proven to reduce the error in the output neurons at any moment in time.

The NLA principle builds on and integrates various ingredients from existing work and theories. Output neurons, be they motor neurons or decision-making neurons, are postulated to be 'nudged' towards the desired target time course by additional synaptic input to the soma or the proximal apical dendrite, as described by *Urbanczik and Senn, 2014*. The cortical microcircuit with lateral 'inhibition' that seeks to cancel the top-down feedback in order to extract the apical error is inspired by *Sacramento et al., 2018* and *Haider et al., 2021*. The energy-based approach for describing

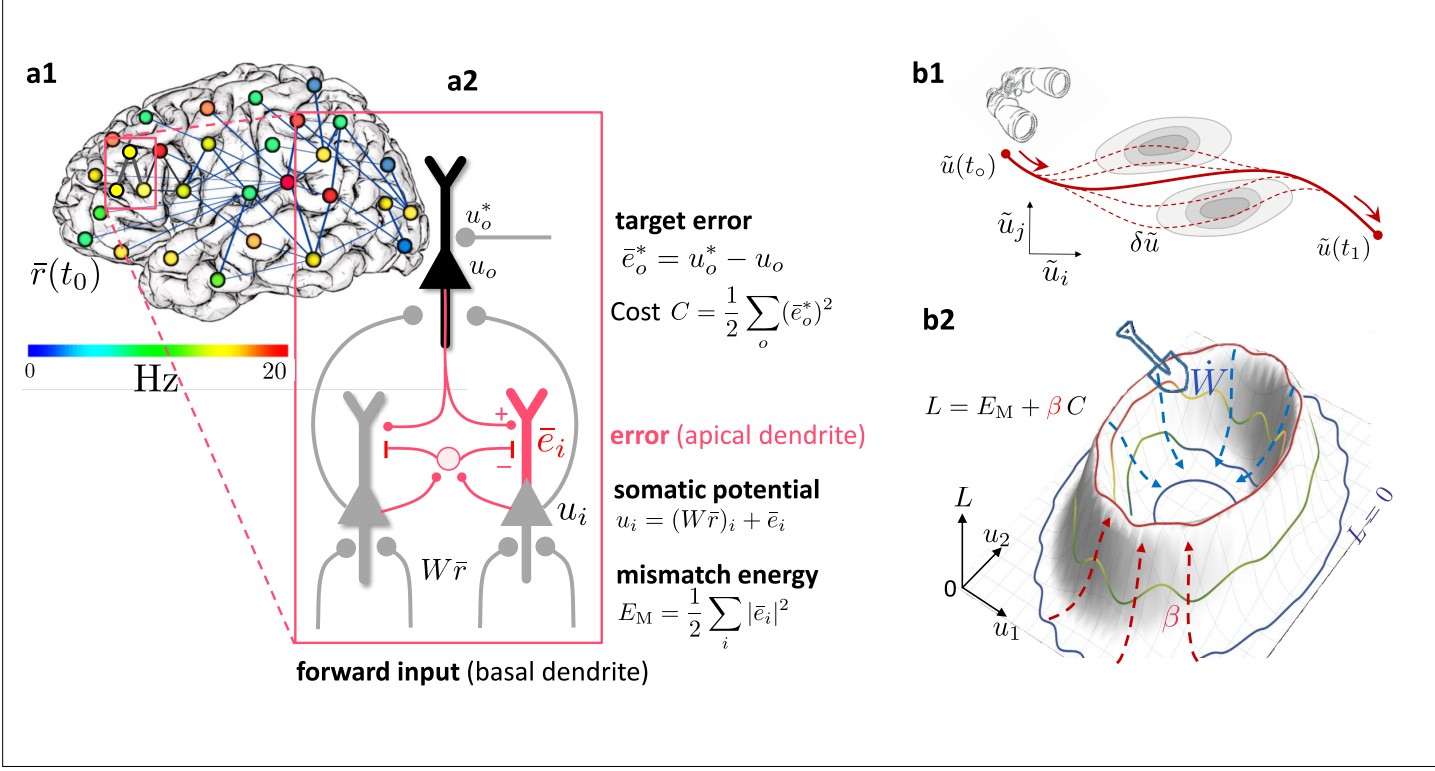

**Figure 1.** Somato-dendritic mismatch energies and the neuronal least-action (NLA) principle. (**a1**) Sketch of a cross-cortical network of pyramidal neurons described by NLA. (**a2**) Correspondence between elements of NLA and biological observables such as membrane voltages and synaptic weights. (**b1**) The NLA principle postulates that small variations $\delta\tilde{u}$ (dashed) of the trajectories $\tilde{u}$ (solid) leave the action invariant, $\delta A = 0$. It is formulated in the look-ahead coordinates $\tilde{u}$ (symbolized by the spyglass) in which 'hills' of the Lagrangian (shaded gray zones) are foreseen by the prospective voltage so that the trajectory can turn by early enough to surround them. (**b2**) In the absence of output nudging ($\beta = 0$), the trajectory $u(t)$ is solely driven by the sensory input, and prediction errors and energies vanish ($L = 0$, outer blue trajectory at bottom). When nudging the output neurons towards a target voltage ($\beta > 0$), somatodendritic prediction errors appear, the energy increases (red dashed arrows symbolising the growing 'volcano') and the trajectory $u(t)$ moves out of the $L = 0$ hyperplanes, riding on top of the 'volcano' (red trajectory). Synaptic plasticity $\dot{W}$ reduces the somatodendritic mismatch along the trajectory by optimally 'shoveling down the volcano' (blue dashed arrows) while the trajectory settles in a new place on the $L = 0$ hyperplane (inner blue trajectory at bottom).

error-backpropagation for weak nudging is borrowed from the Equilibrium Propagation algorithm (*Scellier and Bengio, 2017*) that we generalize from a steady-state algorithm to real-time computation in cross-cortical microcircuits. Our theory covers both cases of weak and strong output nudging. For strong nudging, it likewise generalizes the least-control principle (*Meulemans et al., 2022*) and the prospective configuration algorithm (*Song et al., 2024*) from a steady-state to a dynamic real-time version, linking to optimal feedback control (*Todorov and Jordan, 2002*). Finally, the apical activity of our pyramidal neurons can be seen in the tradition of predictive coding (*Rao and Ballard, 1999*), where cortical feedback connections try to explain away lower-level activities. Yet, different from classical predictive coding, our prediction errors are integrated with the soma, and these errors are prospective in time. The errors extrapolate from current to future activities, so that their integration improves the network output in real time. The combination of an energy-based model with prospective coding in which neuronal integration delays are compensated on the fly enters also in *Haider et al., 2021*.

The paper is organized as follows: we first define the prospective somatodendritic mismatch error, construct out of this the mismatch energy of a network, and 'minimize' this energy to obtain the error-corrected, prospective voltage dynamics of the network neurons. We then show that the prospective error coding leads to an instantaneous and joint processing of low-pass filtered input signals and backpropagated errors. Applied to motor control, the instantaneous processing is interpreted as a moving equilibrium hypothesis according to which sensory inputs, network state, motor commands, and muscle feedback are in a self-consistent equilibrium at any point of the movement. We then

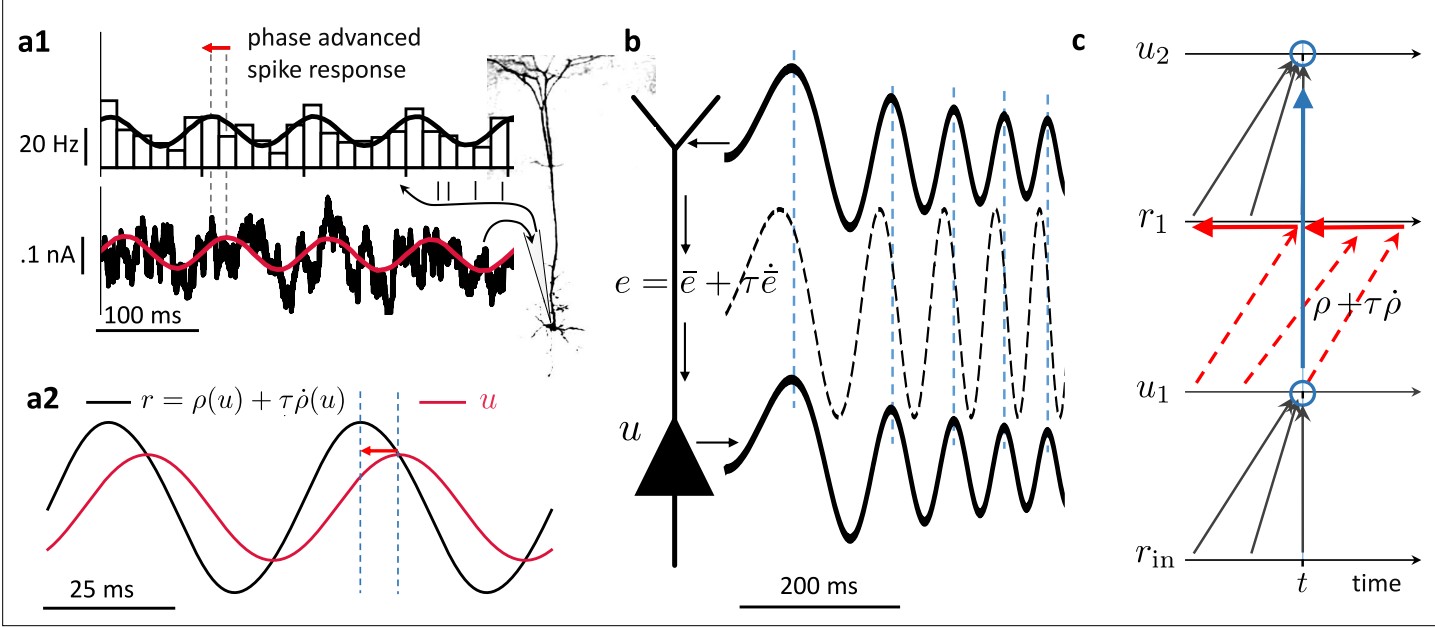

**Figure 2.** Prospective coding in cortical pyramidal neurons enables instantaneous voltage-to-voltage transfer. (**a1**) The instantaneous spike rate of cortical pyramidal neurons (top) in response to sinusoidally modulated noisy input current (bottom) is phase-advanced with respect to the input adapted from *Köndgen et al., 2008*. (**a2**) Similiarly, in neuronal least-action (NLA), the instantaneous firing rate of a model neuron ($r = \rho(u) + \tau\dot\rho(u)$, black) is phase-advanced with respect to the underlying voltage ($u$, red, postulating that the low-pass filtered rate is a function of the voltage, $\bar r = \rho(u)$). (**b**) Dendritic input in the apical tree (here called $\bar e$) is instantaneously causing a somatic voltage modulation ($u$, modeling data from *Ulrich, 2002*). The low-pass filtering with $\tau$ along the dendritic shaft is compensated by a lookahead mechanism in the dendrite ($e = \bar e + \tau\dot{\bar e}$). In (*Ulrich, 2002*) a phase advance is observed even with respect to the dendritic input current, not only the dendritic voltage, although only for slow modulations (as here). (**c**) While the voltage of the first neuron ($u_1$) integrates the input rates $r_{\rm in}$ from the past (bottom black upward arrows), the output rate $r_1$ of that first neuron looks ahead in time, $r_1 = \rho(u_1) + \tau\dot\rho(u_1)$ (red dashed arrows pointing into the future). The voltage of the second neuron ($u_2$) integrates the prospective rates $r_1$ (top black upwards arrows). By doing so, it inverts the lookahead operation, resulting in an instantaneous transfer from $u_1(t)$ to $u_2(t)$ (blue arrow and circles).

derive a local learning rule that globally minimizes the somato-dendritic mismatch errors across the network, and show how this learning can be implemented through error-extracting cortical microcircuits and dendritic predictive plasticity.

## Results

### Somato-dendritic mismatch errors and the Lagrangian of cortical circuits

We consider a network of neurons – identified as pyramidal cells – with firing rates $r_i(t)$ in continuous time $t$. The somatic voltage $u_i$ of pyramidal neuron $i$ is driven by the close-by basal input current, $\sum_j W_{ij}r_j$, with presynaptic rates $r_j$ and synaptic weights $W_{ij}$, and an additional distal apical input $e_i$ that will be learned to represent a prospective prediction error at any moment in time (*Figure 1a*). While in classical rate-based neuron models the firing rate $r_i$ of a neuron is a function of the somatic voltage, $\rho(u_i)$, the NLA principle implies that the effective firing rate of a cortical neuron is prospective. More concretely, the formalism derives a firing rate that linearly extrapolates from $\rho(u_i)$ into the future with the temporal derivative, $r_i = \rho(u_i) + \tau\dot\rho(u_i)$, where $\dot\rho(u_i)$ represents the temporal derivative of $\rho(u_i(t))$. There is experimental evidence for such prospective coding in cortical pyramidal neurons where the instantaneous rate $r_i$ is in fact not only a function of the underlying voltage, but also a function of how quickly that voltage increases (see *Figure 2a*).

The second central notion of the theory is the prospective error $e_i$, that we interpret as prospective somato-dendritic mismatch error in the individual network neurons, $e_i = (u_i + \tau\dot u_i) - \sum_j W_{ij}r_j$. It is defined as a mismatch between the prospective voltage, $u_i + \tau\dot u_i$, and the weighted prospective input rates, $\sum_j W_{ij}r_j$. In the same way, as the firing rates $r_j$ linearly extrapolate into the future given the

current-voltages $u_j$ of the presynaptic neurons $j$, the postsynaptic error is based on the linear extrapolation of its current voltage $u_i$ using its temporal derivative, $u_i + \tau \dot{u}_i$. If the prospective error $e_i$ is low-pass filtered with time constant $\tau$, it takes the form $\bar{e}_i = u_i - \sum_j W_{ij} \bar{r}_j$, where $\bar{r}_j$ is the corresponding low-pass filtered firing rate of the presynaptic neuron $j$ (that becomes a function of the presynaptic voltage, $\bar{r}_j = \rho(u_j)$, see Methods, Sect. Euler-Lagrange equations as inverse low-pass filters). We refer to $\bar{e}_i$ as a somato-dendritic mismatch error of neuron that, as compared to $e_i$, is non-prospective and instantaneous.

We next interpret the mismatch error $\bar{e}_i$ in terms of the morphology and biophysics of pyramidal neurons with basal and apical dendrites. While the error $e_i$ is formed in the apical dendrite, this error is low-pass filtered and added to the somatic voltage $u_i$, that is also driven by the low-pass filtered basal input $\sum_j W_{ij} \bar{r}_j$, so that $u_i = \sum_j W_{ij} \bar{r}_j + \bar{e}_i$. From the perspective of the basal dendrites, the low-pass filtered apical error $\bar{e}_i$ can be calculated as the difference between the somatic voltage and the own local low-pass filtered input, $\bar{e}_i = u_i - \sum_j W_{ij} \bar{r}_j$. The somatic voltage $u_i$ is assumed to be sampled in the basal dendrite by the backpropagating acting potentials (*Urbanczik and Senn, 2014*; *Spicher et al., 2017*). The apical error now appears as a 'somato-basal' mismatch error, that both are summarized as a somato-dendritic mismatch error. It tells the difference between 'what a neuron does,' which is based on the somatic voltage $u_i$, and 'what the basal inputs think it should do,' which is based on its own input $\sum W_{ij} \bar{r}_j$ (*Figure 1a2*). The two quantities may deviate because neuron $i$ get additional 'unpredicted' apical inputs from higher-area neurons that integrate with the somatic voltage $u_i$. What cannot be predicted in $u_i$ by the sensory-driven basal input remains as somato-basal (somato-dendritic) mismatch error $\bar{e}_i$.

Associated with this mismatch error is the somatodendritic mismatch energy defined for each network neuron $i \in \mathcal{N}$ as the squared mismatch error,

$$E_i^M = \tfrac{1}{2} \bar{e}_i^2 = \tfrac{1}{2} \left( u_i - \sum_j W_{ij} \bar{r}_j \right)^2 . \tag{1}$$

On a subset of output neurons of the whole network, $\mathcal{O} \subseteq \mathcal{N}$, a cost is defined as a function of the somatic voltage and some instructive reference signal such as targets or a reward. When a target trajectory $u_o^*(t)$ is available, the cost is defined at each time point as a squared target error,

$$C_o = \tfrac{1}{2} (\bar{e}_o^*)^2 = \tfrac{1}{2} \left( u_o^* - u_o \right)^2 \tag{2}$$

Much more general mismatch energies and cost functions are conceivable, for instance, errors of the form $\bar{e}_i = u_i - f_i(\boldsymbol{u}, t)$ for general functions $f_i$ of the voltage vector $\boldsymbol{u}$ and of time, encompassing conductance-based neurons, but also further dynamic variables can be included such as threshold adaptation (see Appendix 6). The cost represents a performance measure for the entire network that produces the output voltages $u_o(t)$ in response to some input rates $\boldsymbol{r}_{\text{in}}(t)$. The cost directly relates to behavioral or cognitive measures such as the ability of an animal or human to perform a particular task in real time. The target could be provided by explicit external supervision, for example, target movements in time encoded by $u_o^*(t)$, it could represent an expected reward signal, or it could arise via self-supervision from other internal prediction errors.

We define the Lagrangian (or 'total energy') of the network as a sum across all mismatch energies and costs, weighted by the nudging strength $\beta$ of the output neurons,

$$L = \sum_{i \in \mathcal{N}} E_i^M + \beta \sum_{o \in \mathcal{O}} C_o = \frac{1}{2} \sum_{i \in \mathcal{N}} \left( u_i - \sum_j W_{ij} \bar{r}_j \right)^2 + \frac{\beta}{2} \sum_{o \in \mathcal{O}} \left( u_o^* - u_o \right)^2 . \tag{3}$$

The low-pass filtered presynaptic rates, $\bar{r}_j$, also encompass the external input neurons. While in classical energy-based approaches, $L$ is called the total energy, we call it the 'Lagrangian' because it will be integrated along real and virtual voltage trajectories as done in variational calculus (leading to the Euler-Lagrange equations, see below and Appendix 6). We 'prospectively' minimize $L$ locally across a voltage trajectory, so that, as a consequence, the local synaptic plasticity for $W_{ij}$ will globally reduce the cost along the trajectory (Theorem 1 below).

Due to the prospective coding, the Lagrangian can be minimal at any moment in time while the network dynamics evolve. This is different from the classical predictive coding (*Rao and Ballard, 1999*) and energy-based approaches (*Scellier and Bengio, 2017*; *Song et al., 2024*), where a stimulus

needs to be fixed in time while the network relaxes to a steady state, and only there the prediction error is minimized (see Appendix 3).

## The least-action principle expressed for prospective firing rates

Motivated by the prospective firing in pyramidal neurons, we postulate that cortical networks strive to look into the future to prevent instantaneous errors. Each neuron tries to move along a trajectory that minimizes its own mismatch error $\bar{e}_i$ across time (*Figure 1b*). The 'neuronal currency' with which each neuron 'trades' with others to choose its own error-minimizing trajectory is the future discounted membrane potential,

$$\tilde{u}(t) = \frac{1}{\tau} \int_t^\infty u(t') \, e^{-\frac{t' - t}{\tau}} \, \mathrm{d}t' \, . \tag{4}$$

The prospective voltages $\tilde{u}$ are the 'canonical coordinates' entering the NLA principle, and in these prospective coordinates the overall network searches for a 'least-action trajectory'. Since from $\tilde{u}$ we can recover the instantaneous voltage via $u = \tilde{u} - \tau\dot{\tilde{u}}$ (see Appendix 2), we can replace $u$ in the Lagrangian and obtain $L$ as a function of our new prospective coordinates $\tilde{u}$ and the 'velocities' $\dot{\tilde{u}}$, i.e., $L = L\left[\tilde{\boldsymbol{u}}, \dot{\tilde{\boldsymbol{u}}}\right]$, where bold fonts represent vectors. Inspired by the least-action principle from physics, we define the neuronal action $A$ as a time-integral of the Lagrangian,

$$A = \int_{t_1}^{t_2} L\left[\tilde{\boldsymbol{u}}(t), \dot{\tilde{\boldsymbol{u}}}(t)\right] \, \mathrm{d}t \, . \tag{5}$$

The NLA principle postulates that the trajectory $\tilde{\boldsymbol{u}}(t)$ keeps the action $A$ stationary with respect to small variations $\delta\tilde{\boldsymbol{u}}$ (*Figure 1b1*). In other words, nature chooses a trajectory such that, when deviating a little bit from it, say by $\delta\tilde{\boldsymbol{u}}$, the value of $A$ will not change (or at most up to second order in the variation), formally $\delta A = 0$. The motivation to search for a trajectory that keeps the action stationary is borrowed from physics. The motivation to search for a stationary trajectory by varying the near-future voltages $\tilde{\boldsymbol{u}}$, instead of $\boldsymbol{u}$, is assigned to the evolutionary pressure in biology to 'think ahead of time.' To not react too late, internal delays involved in the integration of external feedback need to be considered and eventually need to be overcome. In fact, only for the 'prospective coordinates' defined by looking ahead into the future, even when only virtually, will a real-time learning from feedback errors become possible (as expressed by our Theorems below).

The equations of motion that keep the action stationary with respect to these prospective coordinates are known to satisfy the Euler-Lagrange equations.

$$\frac{\partial L}{\partial \tilde{u}_i} - \frac{\mathrm{d}}{\mathrm{d}t} \frac{\partial L}{\partial \dot{\tilde{u}}_i} = 0. \tag{6}$$

Applying these equations to our Lagrangian yields a prospective version of the classical leaky integrator voltage dynamics, with rates $\boldsymbol{r}$ and errors $\boldsymbol{e}$ that are looking into the future (Methods, Sects. Euler-Lagrange equations as inverse low-pass filters, Deriving the network dynamics from the Euler-Lagrange equations),

$$\tau\dot{\boldsymbol{u}} = -\boldsymbol{u} + \boldsymbol{W}\boldsymbol{r} + \boldsymbol{e}, \tag{7a}$$

$$\bar{\boldsymbol{e}} = \bar{\boldsymbol{r}}'_{\text{net}} \cdot \boldsymbol{W}_{\text{net}}^{\mathrm{T}} \bar{\boldsymbol{e}} + \beta\bar{\boldsymbol{e}}^*. \tag{7b}$$

The ' ' ' denotes the component-wise product, and the weight matrix splits into weights from input neurons and weights from network neurons, $\boldsymbol{W} = (\boldsymbol{W}_{\text{in}}, \boldsymbol{W}_{\text{net}})$. While for output neurons a target error can be defined, $\bar{e}_o^* = u_o^* - u_o$, for non-output neurons $i$ no target exists and we hence set $\bar{e}_i^* = 0$. In a control theoretic framework, the neuronal dynamics (*Equation 7a*) represent the state trajectory, and the adjoint error dynamics *Equation 7b* represent the integrated costate trajectory (*Todorov, 2006*).

From the point of view of theoretical physics, where the laws of motion derived from the least-action principle contain an acceleration term (as in Newton's law of motion, like $m\ddot{x} = -x + F$ for a harmonic oscillator), one may wonder why no second-order time derivative appears in the NLA dynamics. As an intuitive example, consider driving into a bend. Looking ahead in time helps us to reduce the lateral acceleration by braking early enough, as opposed to braking only when the lateral acceleration is

already present. This intuition is captured by minimizing the neuronal action $A$ with respect to the discounted future voltages $\tilde{u}_i$ instead of the instantaneous voltages $u_i$. Keeping up an internal equilibrium in the presence of a changing environment requires looking ahead and compensating early for the predicted perturbations. Technically, the acceleration disappears because the Euler-Lagrange operator (*Equation 6*) turns into a lookahead-gradient operator, $\frac{\partial}{\partial \tilde{u}_i} - \frac{d}{dt}\frac{\partial}{\partial \dot{\tilde{u}}_i} = \left(1 + \frac{d}{dt}\right)\frac{\partial}{\partial u_i}$, since the $\ddot{\tilde{u}}_i$ is absorbed via $\dot{\tilde{u}}_i - \tau\ddot{\tilde{u}}_i = \dot{u}_i$ (see Methods, Sect. Euler-Lagrange equations as inverse low-pass filters, and Appendix 6 for the link to the least-action principle in physics).

Mathematically, the voltage dynamics in *Equation 7a* specifies an implicit differential equation since $\dot{u}(t)$ also appears on the right-hand side. This is because the prospective rates $r = \rho(u) + \tau\dot{\rho}(u)$ include $\dot{u}$ through $\dot{\rho}(u) = \rho'(u)\cdot\dot{u}$. Likewise, the prospective errors $e = \bar{e} + \tau\dot{\bar{e}}$, with $\bar{e}$ given in *Equation 7b* and plugged into *Equation 7a*, imply $\dot{u}$ through $\dot{\bar{e}}(u) = \bar{e}'(u)\cdot\dot{u}$. Nevertheless, the voltage dynamics can be stably run by replacing $\dot{u}(t)$ on the right-hand side of *Equation 7a* with the temporal derivative $\dot{u}(t - dt)$ from the previous time step (technically, the Hessian $(\mathbf{1} - \mathbf{W}\rho' - \bar{e}')$ is required to be strictly positive definite, see Methods Sect. From implicit to explicit differential equations and Appendix 3). This ensures that the voltage dynamics of *Equation 7a*, *Equation 7b* can be implemented in cortical neurons with a prospective firing and a prospective dendritic error (see *Figure 2*).

The error expression in *Equation 7b* is reminiscent of error backpropagation *Rumelhart et al., 1986* and can in fact be related (Methods, Sect. Deriving the error backpropagation formula). Formally, the errors are backpropagated via transposed network matrix, $\mathbf{W}_{net}^{T}$, modulated by $\bar{r}_i'$, the derivative of $\bar{r}_i = \rho(u_i)$ with respect to the underlying voltage. While the transpose can be constructed with various local methods see *Akrout et al., 2019*; *Max et al., 2022* in our simulations we mainly adhere to the phenomenon of feedback alignment (*Lillicrap et al., 2016*) and consider fixed and randomized feedback weights $\mathbf{B}$ (unless stated differently). Recent control theoretical work is exploiting the same prospective coding technique as expressed in *Equation 7a*, *Equation 7b* to tackle general time-varying optimization problems see *Simonetto et al., 2020* for a review and Appendix 3 for the detailed connection.

## Prospective coding in neurons and instantaneous propagation

The prospective rates and errors entering via $r$ and $e$ in the NLA (*Equation 7a*) are consistent with the prospective coding observed in cortical pyramidal neurons in vitro (*Köndgen et al., 2008*). Upon sinusoidal current injection into the soma, the somatic firing rate is advanced with respect to its voltage (*Figure 2a*), effectively compensating for the delay caused by the current integration. Likewise, sinusoidal current injection in the apical tree causes a lag-less voltage response in the soma (*Figure 2b*, *Ulrich, 2002*). While the rates and errors in general can be reconstructed from their low-pass filterings via $r = \bar{r} + \tau\dot{\bar{r}}$ and $e = \bar{e} + \tau\dot{\bar{e}}$, they become prospective in time because $\bar{r}$ and $\bar{e}$ are themselves instantaneous functions of the voltage $u$, and hence $r$ and $e$ depend on $\dot{u}$. The derivative of the membrane potential implicitly also appears in the firing mechanism of Hodgkin-Huxley-type conductances, with a quick depolarization leading to a stronger sodium influx due to the dynamics of the gating variables (*Hodgkin and Huxley, 1952*). This advances the action potential as compared to a firing that would only depend on $u$, not $\dot{u}$, giving an intuition of how such a prospective coding may arise. A similar prospective coding has been observed for retinal ganglion cells (*Palmer et al., 2015*) and cerebellar Purkinje cells (*Ostojic et al., 2015*), making a link from the visual input to the motor control.

To understand the instantaneous propagation through the network, we low-pass filter the dynamic equation $u + \tau\dot{u} = \mathbf{W}r + e$ (obtained by rearranging *Equation 7a*), with $\bar{e}$ given by *Equation 7b*, to obtain the somatic voltage $u = \mathbf{W}\bar{r}(u) + \bar{e}(u)$. At any point in time, the voltage is in a moving equilibrium between forward and backpropagating inputs. Independently of the network architecture, whether recurrent or not, the output is an instantaneous function of the low-pass filtered input and a putative correction towards the target, $u_o(t) = F_W(\bar{r}_{in}(t), \bar{e}_o^*(t))$, see *Figure 2C* and Methods, Sect. Proving theorem 1 (rt-DeEP). The mapping again expresses an instantaneous propagation of voltages throughout the network in response to both, the low-pass filtered input $\bar{r}_{in}$ and feedback error $\bar{e}_o^*$. This instantaneity is independent of the network size, and in a feed-forward network is independent of its depths (see also *Haider et al., 2021*, where the instantaneity is on the rates, not the voltages). In the absence of the look-ahead activity, each additional layer would slow down the network relaxation time.

Notice that an algorithmic implementation of the time-continuous dynamics of a $N$-layer feed-forward network would still need $N$ calculation steps until information from layer 1 reaches layer $N$. However, this does not imply that an analog implementation of the prospective dynamics will encounter delays. To see why, consider a finite step-change $\Delta u_1$ in the voltage of layer 1. In the absence of the look-ahead, $\Delta u_1$ was mapped within the infinitesimal time interval $dt$ to an infinitesimal change $du_2$ in the voltages of layer 2. But with a prospective firing rate, $r_1 = \rho(u_1) + \tau \rho'(u_1) \cdot \dot{u}_1$, a step-change $\Delta u_1$ translates to a delta-function in $r_1$, this in turn to a step-change in the low-pass filtered rates $\Delta \bar{r}_1$, and therefore within $dt$ to a step-change $\Delta u_2$ in the voltages $u_2$ of the postsynaptic neurons (*Figure 2c*). Iterating this argument, a step-change $\Delta u_1$ propagates 'instantaneously' through $N$ layers within the 'infinitesimal' time interval $N dt$ to a step-change $\Delta u_N$ in the last layer. When run in a biophysical device in continuous time that exactly implements the dynamical *Equation 7a*, the implementation becomes an instantaneous computation (since $dt \to 0$). Yet, in a biophysical device, information has to be moved across space. This typically introduces further propagation delays that may not be captured in our formalism where low-pass filtering and prospective coding cancel each other exactly. Nevertheless, analog computation in continuous time, as formalized here, offers an idea to 'instantaneously' realize an otherwise time-consuming numerical recipe run on time-discrete computing systems that operate with a finite clock cycle.

## Prospective control and the moving equilibrium hypothesis

Crucially, at the level of the voltage dynamics (*Equation 7a*) the correction is based on the prospective error $e$. This links our framework to optimal control theory and motor control where delays are also taken into account, so that a movement can be corrected early enough (*Wolpert and Ghahramani, 2000*; *Todorov and Jordan, 2002*; *Todorov, 2004*). The link between energy-based models and optimal control was recently drawn for strong nudging ($\beta \to \infty$) to learn individual equilibrium states (*Meulemans et al., 2022*). Our prospective error $e(t)$ appears as a 'controller' that, when looking at the output neurons, pushes the voltage trajectories toward the target trajectories. Depending on the nudging strength $\beta$, the control is tighter or weaker. For infinitely large $\beta$, the voltages of the output neurons are clamped to the time-dependent target voltages, $u_o = u_o^*$ (implying $e_o^* = 0$), while their errors, $\bar{e}_o = u_o - (W\bar{r})_o$, instantaneously correct all network neurons. For small $\beta$, the output voltages are only weakly controlled, and they are dominated by the forward input, $u_o \approx (W\bar{r})_o$.

To show how the NLA principle with the prospective coding globally maps to cortico-spinal circuits we consider the example of motor control. In the context of motor control, our network mapping $u_o = F_W(\bar{r}_{\text{in}}, \bar{e}_o^*)$ can be seen as a forward internal model that quickly calculates an estimate of the future muscle length $u_o$ based on some motor plans, sensory inputs, and the current proprioceptive feedback (*Figure 3a*). Forward models help to overcome delays in the execution of the motor plan by predicting the outcome, so that the intended motor plans and commands can be corrected on the fly (*Kawato, 1999*; *Wolpert and Ghahramani, 2000*).

The observation that muscle spindles prospectively encode the muscle length and velocity (*Dimitriou and Edin, 2010*) suggests that the prospective coding in the internal forward model mirrors the prospective coding in the effective forward pathway. This forward pathway leads from the motor plan to the spindle feedback, integrating also cerebellar and brainstem feedback (*Kawato, 1999*). Based on the motor plans, the intended spindle lengths and the effective muscle innervation are communicated via a descending pathway to activate the $\gamma$- and $\alpha$-motoneurons, respectively (*Li et al., 2015*). The mapping from the intended arm trajectory to the intended spindle lengths via $\gamma$-innervation is mainly determined by the joint geometry. The mapping from the intended arm trajectory to the force-generating $\alpha$-innervation, however, needs to also take account of the internal and external forces, and this is engaging our network $W$.

When we prepare an arm movement, spindles in antagonistic muscle pairs that measure the muscle length are tightened or relaxed before the movement starts (*Papaioannou and Dimitriou, 2021*). According to the classical equilibrium-point hypothesis (*Feldman and Levin, 2009*; *Latash, 2010*), top-down input adjusts the activation threshold of the spindles through ($\gamma$-) innervation from the spinal cord so that slight deviations from the equilibrium position can be signaled (*Figure 3a*). We postulate that this $\gamma$-innervation acts also during the movement, setting an instantaneous target $u_o^*(t)$ for the spindle lengths. The effective lengths of the muscle spindles is $u_o$, and the spindles are prospectively signaling back the deviation from the target through the $I_a$-afferents (*Dimitriou and*

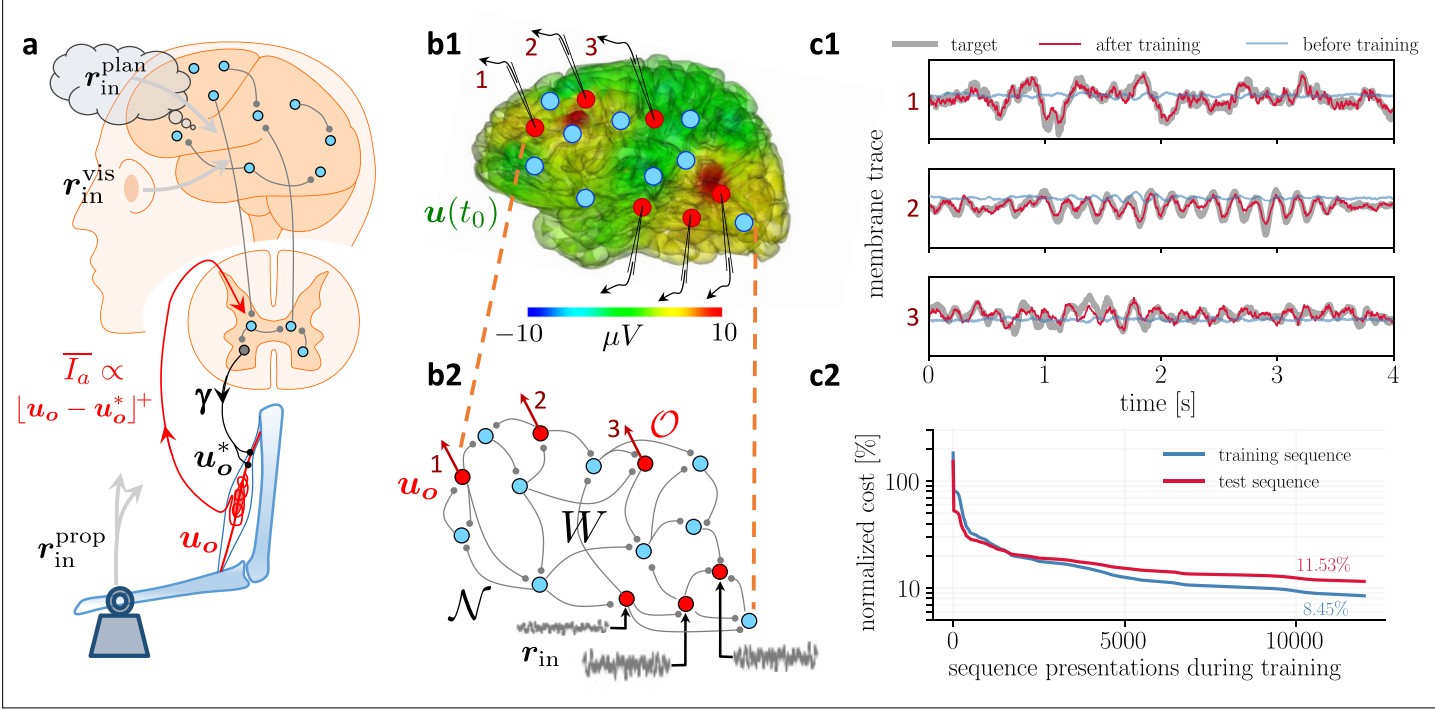

**Figure 3.** Moving equilibrium hypothesis for motor control and real-time learning of cortical activity. (**a**) A voluntary movement trajectory can be specified by the target length of the muscles in time, $\boldsymbol{u_o}^*$, encoded through the $\gamma$-innervation of muscle spindles, and the deviation of the effective muscle lengths from the target, $\boldsymbol{u_o} - \boldsymbol{u_o}^* = -\bar{\boldsymbol{e}}_{\boldsymbol{o}}^*$. The $I_a$-afferents emerging from the spindles prospectively encode the error, so that their low-pass filtering is roughly proportional to the length deviation, truncated at zero (red). The moving equilibrium hypothesis states that the low-pass filtered input $\bar{\boldsymbol{r}}_{in}$, composed of the movement plan $\bar{\boldsymbol{r}}_{in}^{plan}$ and the sensory input (here encoding the state of the plant e.g., through visual and proprioceptive input, $\bar{\boldsymbol{r}}_{in}^{vis}$ and $\bar{\boldsymbol{r}}_{in}^{prop}$), together with the low-pass filtered error feedback from the spindles, $\bar{\boldsymbol{e}}_{\boldsymbol{o}}^*$, instantaneously generate the muscle lengths, $\boldsymbol{u_o} = \boldsymbol{F}_W(\bar{\boldsymbol{r}}_{in}, \bar{\boldsymbol{e}}_{\boldsymbol{o}}^*)$, and are thus at any point in time in an instantaneous equilibrium (defined by *Equation 7a*, *Equation 7b*). (**b1**) Intracortical intracortical electroencephalogram (iEEG) activity recorded from 56 deep electrodes and projected to the brain surface. Red nodes symbolize the 56 iEEG recording sites modeled alternately as input or output neurons, and blue nodes symbolize the 40 'hidden' neurons for which no data is available, but used to reproduce the iEEG activity. (**b2**) Corresponding NLA network. During training, the voltages of the output neurons were nudged by the iEEG targets (black input arrows, but for all red output neurons). During testing, nudging was removed for 14 out of these 56 neurons (here, represented by neurons 1, 2, 3). (**c1**) Voltage traces for the 3 example neurons in a2, before (blue) and after (red) training, overlaid with their iEEG target traces (gray). (**c2**) Total cost, integrated over a window of 8 s of the 56 output nodes during training with sequences of the same duration. The cost for the test sequences was evaluated on a 8 s window not used during training.

*Edin, 2010*; *Dimitriou, 2022*). The low-pass filtered $I_a$-afferents may be approximated by a threshold-nonlinearity, $\overline{I_a} = \beta \lfloor \boldsymbol{u_o} - \boldsymbol{u_o}^* \rfloor^+$, with $\beta$ being interpreted as spindle gain (*Latash, 2018*). Combining the feedback from agonistic and antagonistic muscle pairs allows for extracting the scaled target error $\beta \bar{\boldsymbol{e}}_{\boldsymbol{o}}^* = \beta(\boldsymbol{u_o}^* - \boldsymbol{u_o})$. Taking account of the prospective feedback, we postulate the *moving equilibrium hypothesis* according to which the instructional inputs, $\bar{\boldsymbol{r}}_{in}$, the spindle feedback, $\beta \bar{\boldsymbol{e}}_{\boldsymbol{o}}^*$, and the muscle lengths, $\boldsymbol{u_o}$, are at any point of the movement in a dynamic equilibrium. The moving equilibrium hypothesis extends the classical equilibrium-point hypothesis from the spatial to the temporal domain (for a formal definition of a moving equilibrium see Methods, Sect. From implicit to explicit differential equations).

Prediction errors are also reduced when motor units within a muscle are recruited according to the size principle (*Senn et al., 1997*), which itself was interpreted in terms of the physical least-action principle (*Senn et al., 1995*). With regard to the interpretation of the prospective feedback error $\boldsymbol{e}_{\boldsymbol{o}}^*$ as spindle activity, it is worth noticing that in humans the spindle activity is not only ahead of the muscle activation (*Dimitriou and Edin, 2010*), but also share the property of a motor error (*Dimitriou, 2016*). The experiments show that during the learning of a gated hand movement, spindle activity is initially stronger when making movement errors, and it returns back to baseline with the success of learning. This observation is consistent with the NLA principle, saying that the proprioceptive prediction errors

are minimized through the movement learning. We next address how the synaptic strengths $W$ involved in producing the muscle length can be optimally adapted to capture this learning.

## Local plasticity at basal synapses minimizes the global cost in real time

The general learning paradigm starts with input time series $r_{\text{in}(t),i}$ and target time series $u_o^*(t)$, while assuming that the target series are an instantaneous function of the low-pass filtered input series, $\boldsymbol{u_o^*}(t) = \boldsymbol{F}^*(\bar{\boldsymbol{r}}_{\text{in}}(t))$. The low-pass filtering in the individual inputs could be with respect to any time constant $\tau_{\text{in},i}^*$ (that may also be learned, see Appendix 2). Yet, for simplicity, we assume the same time constant $\tau$ for low-pass filtering the rates of the network neurons and input neurons. The goal of learning is to adapt the synaptic strengths $W$ in the student network so that this moves towards the target mapping, $\boldsymbol{F}_W \rightarrow \boldsymbol{F}^*$. The local synaptic plasticity will also reduce the global cost $C$ defined on the output neurons $o$ in terms of the deviation of the voltage from the target, $u_o^* - u_o$ (*Equation 2*).

The problem of changing synaptic weights to correct the behavior of downstream neurons, potentially multiple synapses away, is typically referred to as the credit assignment problem and is notoriously challenging in physical or biological substrates operating in continuous time. A core aspect of the NLA principle is how it relates the global cost $C$ to the Lagrangian $L$ and eventually to somato-dendritic prediction errors $\bar{e}$ that can be reduced through local synaptic plasticity $\dot{W}$. We define this synaptic plasticity as a partial derivative of the Lagrangian with respect to the weights, $\dot{W} \propto -\frac{\partial L}{\partial W} = \bar{e}\,\bar{r}^{\text{T}}$. Since the somatodendritic mismatch error is $\bar{e} = \boldsymbol{u} - W\bar{r}$, this leads to the local learning rule of the form 'postsynaptic error times low-pass filtered presynaptic rate',

$$\dot{W} = \eta\,(\boldsymbol{u} - W\bar{r})\,\bar{r}^{\text{T}}. \tag{8}$$

The plasticity rule runs simultaneously to the neuronal dynamics in the presence of a given nudging strength $\beta$ that tells how strongly the voltage of an output neuron is pushed towards the target, $u_o \rightarrow u_o^*$. The learning rule is local in space since $W\bar{r}$ is represented as a voltage of the basal dendrites, and the somatic voltage $\boldsymbol{u}$ may be read out at the synaptic site on the basal dendrite from the back-propagating action potentials that sample $\boldsymbol{u}$ at a given time (*Urbanczik and Senn, 2014*). The basal voltage $W\bar{r}$ becomes the dendritic prediction of the somatic activity $\boldsymbol{u}$, interpreting *Equation 8* as 'dendritic predictive plasticity'.

We have derived the neuronal dynamics as a path that keeps the action stationary. Without an external teaching signal, the errors vanish, and the voltage trajectory wriggles on the bottom of the energy landscape ($L = 0$, *Figure 1b2*). If the external nudging is turned on, $\beta > 0$, errors emerge and hills grow out of the landscape. The trajectory still tries to locally minimize the action, but it is lifted upwards on the hills ($L > 0$, *Figure 1b2*). Synaptic plasticity reshapes the landscape so that, while keeping $\beta$ fixed, the errors are reduced and the landscape again flattens. The transformed trajectory settles anew in another place (inside the 'volcano' in 1b2). Formally, the local plasticity rule (*Equation 8*) is shown to perform gradient descent on the Lagrangian and hence on the action. In the energy landscape picture, plasticity 'shovels off' energy along the voltage path so that this is lowered most efficiently. The error that is back-propagated through the network tells at any point on the voltage trajectory how much to 'dig' in each direction, i.e., how to adapt the basal input in each neuron in order to optimally lower the local error.

The following theorem tells that synaptic plasticity $\dot{W}$ pushes the network mapping $\boldsymbol{u_o} = \boldsymbol{F}_W(\bar{r}_{\text{in}})$ towards the target mapping $\boldsymbol{u_o^*} = \boldsymbol{F}^*(\bar{r}_{\text{in}})$ at any moment in time. The convergence of the mapping is a consequence of the fact the plasticity reduces the Lagrangian $L = E^{\text{M}} + \beta C$ along its gradient.

## Theorem 1 (real-time dendritic error propagation, rt-DeEP)

Consider an arbitrary network $W$ with voltage and error dynamics following *Equation 7a*, *Equation 7b*. Then the local plasticity rule $\dot{W} \propto \bar{e}\,\bar{r}^{\text{T}}$ *Equation 8*, acting at each moment along the voltage trajectories, is gradient descent

(i) on the Lagrangian $L$ for any nudging strength $\beta \geq 0$, i.e., $\bar{e}\,\bar{r}^{\text{T}} = -\frac{\mathrm{d}L}{\mathrm{d}W}$, with $\lim\limits_{\beta \to \infty} \bar{e}\,\bar{r}^{\text{T}} = -\frac{\mathrm{d}E^{\text{M}}}{\mathrm{d}W} \propto \dot{W}$.

(ii) on the cost $C$ for small nudging, $\beta \to 0$, while up-scaling the error to $\frac{1}{\beta}\bar{e}$, i.e., $\lim\limits_{\beta \to 0} \frac{1}{\beta}\bar{e}\,\bar{r}^{\text{T}} = -\frac{\mathrm{d}C}{\mathrm{d}W} \propto \dot{W}$.

The gradient statements hold at any point in time (long enough after initialization), even if the input trajectories $\boldsymbol{r}_{\text{in}}(t)$ contain delta functions and the target trajectories $\boldsymbol{u_o^*}(t)$ contain step functions.

Loosely speaking, the NLA enables the network to localize in space and time an otherwise global problem: what is good for a single neuron (the local plasticity) becomes good for the entire network (the gradient on the global cost). Learning is possible at any point in time along the trajectory because the NLA inferred a prospective voltage dynamics expressed in prospective firing rates $r_i$ and prospective errors $e_i$ of the network neurons. In the limit of strong nudging ($\beta \to \infty$), the learning rule performs gradient descent on the mismatch energies $E^M{}_i$ in the individual neurons. If the network architecture is powerful enough so that after learning all the mismatch energies vanish, $E^M{}_i = 0$, then the cost will also vanish, $C = \frac{1}{2}\|u_o^* - u_o\|^2 = 0$. This is because for the output neurons, the mismatch error includes the target error (*Equation 7b*). In the limit of weak nudging ($\beta \to 0$), the learning rule performs gradient descent on $C$, and with this also finds a local minimum of the mismatch energies.

In the case of weak nudging and a single steady-state equilibrium, the NLA algorithm reduces to the Equilibrium Propagation algorithm (*Scellier and Bengio, 2017*) that minimizes the cost $C$ for a constant input and a constant target. In the case of strong nudging and a single steady-state equilibrium, the NLA principle reduces to the Least-Control Principle (*Meulemans et al., 2022*) that minimizes the mismatch energy $E^M$ for a constant input and a constant target, with the apical prediction error becoming the prediction error from standard predictive coding (*Rao and Ballard, 1999*). While in the Least-Control Principle, the inputs and outputs are clamped to fixed values, the output errors are backpropagated and the network equilibrates in a steady state where the corrected network activities reproduce the clamped output activities. This state is called the 'prospective configuration' in *Song et al., 2024* because neurons deep in the network are informed about the distal target already during the inference, and are correspondingly adapted to be consistent with this distal target. In the NLA principle, after an initial transient, the network always remains in the moving equilibrium due to the prospective coding. While inputs and targets dynamically change, the network moves along a continuous sequence of prospective configurations.

In the motor control example, the theorem tells that a given target motor trajectory $u_o^*(t)$ is learned to be reproduced with the forward model $u_o(t) = F_W(\bar{r}_{in}(t))$, by applying the dendritic predictive plasticity for the network neurons (*Equation 8*). We next exemplify the theory by looking into the brain, reproducing cortical activity, and showing how a multi-layer cortical network can learn a sensory-motor mapping while staying in a moving equilibrium throughout the training.

## Reproducing intracortical EEG recordings and recognizing handwritten digits

As an illustration, we consider a recurrently connected network that learns to represent intracortical electroencephalogram (iEEG) data from epileptic patients (*Figure 3b*). For each electrode, we assign a neuron within this network to represent the activity of the cell cluster recorded in the corresponding iEEG signal via its membrane potential. During learning, a randomly selected subset of electrode neurons are nudged towards the target activity from recorded data while learning to be reproduced by the other neurons. After learning, we can present only a subset of electrode neurons with previously unseen recordings and observe how the activity of the other neurons closely matches the recordings of their respective electrodes (*Figure 3c*). The network derived from NLA is thus able to learn complex correlations between signals evolving in real-time by embedding them in a recurrent connectivity structure.

As an example of sensory-motor processing in the NLA framework, we next consider a well-studied image recognition task, here reformulated in a challenging time-continuous setting, and interpreted as a motor task where 1 out of 10 fingers has to be bent upon seeing a corresponding visual stimulus (see *Figure 3*). In the context of our moving equilibrium hypothesis, we postulate that during the learning phase, but not the testing phase, an auditory signal identifies the correct finger and sets the target spindle lengths of 10 finger flexors, $u_o^*(t)$. The target spindle length encodes the desired contraction of a flexor muscle in the correct finger upon the visual input $r_{in}(t)$, and a corresponding relaxation for the nine incorrect fingers.

We train a hierarchical three-layer network on images of handwritten digits (MNIST, *LeCun, 1998*), with image presentation times between $0.5\tau$ (=5 ms) and $20\tau$ (=200 ms, with $\tau = 10$ the membrane time constant). *Figure 4a-c* depict the most challenging scenario with the shortest presentation time. Synaptic plasticity is continuously active, despite the network never reaching a temporal steady state (*Figure 4b1*). Due to the lookahead firing rates in the NLA, the mismatch errors $\bar{e}_i(t)$ represent the

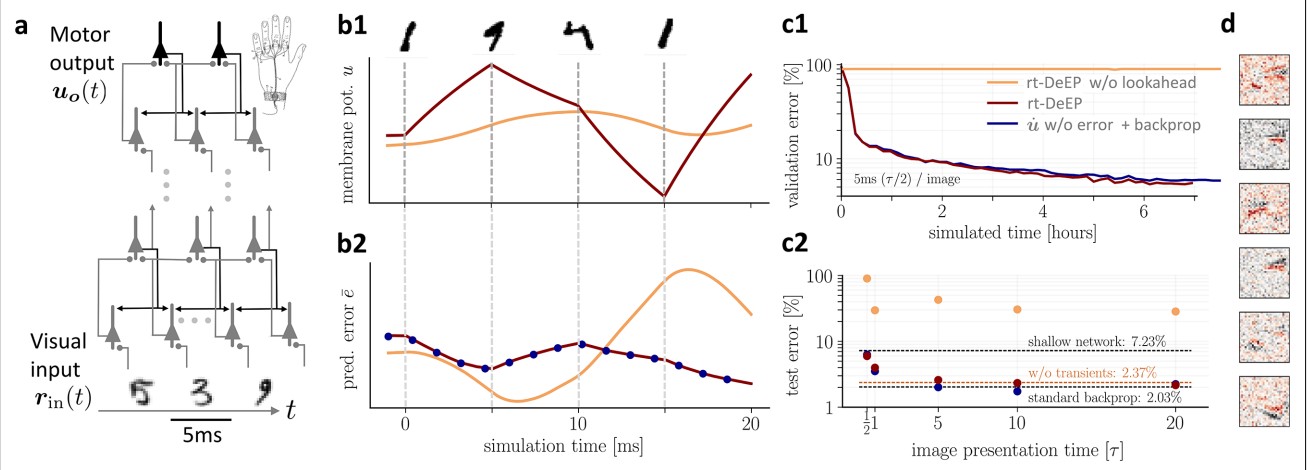

**Figure 4.** On-the-fly learning of finger responses to visual input with real-time dendritic error propagation (rt-DeEP). (**a**) Functionally feedforward network with handwritten digits as visual input ($r_{in}^{(2)}(t)$ in *Figure 3a*, here from the MNIST data set, 5 ms presentation time per image), backprojections enabling credit assignment, and activity of the 10 output neurons interpreted as commands for the 10 fingers (forward architecture: 784×500×10 neurons). (**b**) Example voltage trace (**b1**) and local error (**b2**) of a hidden neuron in neuronal least-action (NLA) (red) compared to an equivalent network without lookahead rates (orange). Note that neither network achieves a steady state due to the extremely short input presentation times. Errors are calculated via exact backpropagation, i.e., by using the error backpropagation algorithm on a pure feedforward NLA network at every simulation time step (with output errors scaled by $\beta$), shown for comparison (blue dots). (**c**) Comparison of network models during and after learning. Color scheme as in (**b**). (**c1**) The test error under NLA evolves during learning on par with classical error backpropagation performed each Euler $dt$ based on the feedforward activities. In contrast, networks without lookahead rates are incapable of learning such rapidly changing stimuli. (**c2**) With increasing presentation time, the performance under NLA further improves, while networks without lookahead rates stagnate at high error rates. This is caused by transient, but long-lasting misrepresentation of errors following stimulus switches: when plasticity is turned off during transients and is only active in the steady state, comparably good performance can be achieved (dashed orange). (**d**) Receptive fields of 6 hidden-layer neurons after training, demonstrating that even for very brief image presentation times (5ms), the combined neuronal and synaptic dynamics are capable of learning useful feature extractors such as edge filters.

correct gradient and propagate without lag throughout the network. As a consequence, our mismatch errors are almost equal to the errors obtained from classical error backpropagation applied at each time step to the purely forward network (i.e. the network that suppresses the error-correction $\bar{e}$ of the voltage and instead considers the 'classical' voltage $u_l = W_l \rho(u_{l-1})$ only, see blue dots in *Figure 4b2*). The network eventually learned to implement the mapping $u_o = F_W(\bar{r}_{in}) \approx u_o^*$ with a performance comparable to error-backpropagation at each $dt$, despite the short presentation time of only 5 ms (*Figure 4c1*). The approximation is due to the fact that the NLA learns an instantaneous mapping from the low-pass filtered input rates $\bar{r}_{in}$ to the output voltage $u_o$, while the mapping from the original input rates $r_{in}$ to the voltages $u_1$ of the first-layer neurons (and hence also to the output voltages $u_o$) is delayed by $\tau_{in}$. Since in the simulations, the target voltages $u_o^*$ were switched instantaneously with $r_{in}$ (and not with $\bar{r}_{in}$), however, a mismatch error between $u_o$ and $u_o^*$ remains for stimulus presentation times shorter than $\tau_{in}$ (*Figure 4c2*). The Latent Equilibrium (*Haider et al., 2021*) avoids these temporal limitations by implementing an instantaneous mapping on the rates instead on the voltages (Methods, Sect. From implicit to explicit differential equations).

The instantaneous voltage propagation relieves an essential constraint of previous models of bio-plausible error backpropagation (e.g. *Scellier and Bengio, 2017*; *Whittington and Bogacz, 2017*; *Sacramento et al., 2018*), with reviews (*Richards et al., 2019*; *Whittington and Bogacz, 2019*; *Lillicrap et al., 2020*): without lookahead firing rates, networks need much longer to correctly propagate errors across layers, with each layer roughly adding another membrane time constant of 10 ms, and thus cannot cope with realistic input presentation times. In fact, in networks without lookahead output, learning is only successful if plasticity is switched off while the network dynamics did not reach a stationary state during a stimulus presentation interval (*Figure 4c2*). Notice also that the prospective coding is necessary to keep the network activity stable for an instantaneous processing of the sensory input. If, in the absence of prospective coding, we would only shrink the membrane time constant to 0, the recurrent error processing would become unstable (see Appendix 3).

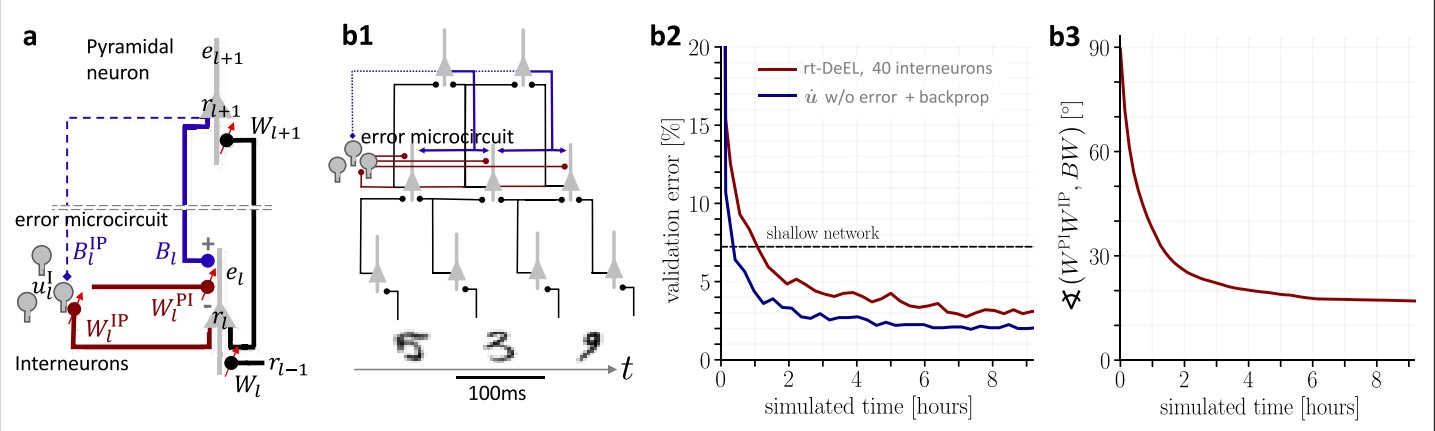

**Figure 5.** Hierarchical plastic microcircuits implement real-time dendritic error learning (rt-DeEL). (**a**) Microcircuit with 'top-down' input (originating from peripheral motor activity, blue line) that is explained away by the lateral input via interneurons (dark red), with the remaining activity representing the error $\bar{e}_l$. Plastic connections are denoted with a small red arrow and nudging with a dashed line. (**b1**) Simulated network with 784-300-10 pyramidal-neurons and a population of 40 interneurons in the hidden layer used for the MNIST learning task where the handwritten digits have to be associated with the 10 fingers. (**b2**) Test errors for rt-DeEL with joint tabula rasa learning of the forward and lateral weights of the microcircuit. A similar performance is reached as with classical error backpropagation. For comparability, we also show the performance of a shallow network (dashed line). (**b3**) Angle derived from the Frobenius norm between the lateral pathway $W^{IP}_l W^{PI}_l$ and the feedback pathway $B_l W_{l+1}$. During training, both pathways align to allow correct credit assignment throughout the network. Indices are dropped in the axis label for readability.

## Implementation in cortical microcircuits

So far, we did not specify how errors $e$ appearing in the differential equation for the voltage (*Equation 7a*) are transmitted across the network in a biologically plausible manner. Building on *Sacramento et al., 2018*, we propose a cortical microcircuit to enable this error transport, with all neuron dynamics evolving according to the NLA principle. Although the idea applies to arbitrarily connected networks, we use the simpler case of functionally feedforward networks to illustrate the flow of information in these microcircuits (*Figure 5a*).

For such an architecture, pyramidal neurons in area $l$ (that is a 'layer' of the feedforward network) are accompanied by a pool of interneurons in the same layer (area). The dendrites of the interneurons integrate in time (with time constant $\tau$) lateral input from pyramidal neurons of the same layer ($r_l$) through plastic weights $W^{IP}_l$. Additionally, interneurons receive 'top-down nudging' from pyramidal neurons in the next layer through randomly initialized and fixed back projecting synapses $B^{IP}_l$ targeting the somatic region, and interneuron nudging strength $\beta^I$. The notion of 'top-down' originates from the functionally feed-forward architecture leading from sensory to 'higher cortical areas.' In the context of motor control, the highest 'area' is the last stage controlling the muscle lengths, being at the same time the first stage for the proprioceptive input (*Figure 3a*).

According to the biophysics of the interneuron, the somatic membrane potential becomes a convex combination of the two types of afferent input (*Urbanczik and Senn, 2014*),

$$u^I_l = (1 - \beta^I)W^{IP}_l \bar{r}_l + \beta^I B^{IP}_l u_{l+1}. \tag{9}$$

In the biological implementation, the feedback input is mediated by the low-pass filtered firing rates $\bar{r}_{l+1} = \rho(u_{l+1})$, not by $u_{l+1}$ as expressed in the above equation. Yet, we argue that for a threshold-linear $\rho$ the 'top-down nudging' by the rate $\bar{r}_{l+1}$ is effectively reduced to a nudging by the voltage $u_{l+1}$. This is because errors are only backpropagated when the slope of the transfer function is positive, $r'_{l+1} > 0$, and hence when the upper-layer voltage is in the linear regime. For more general transfer functions, we argue that short-term synaptic depression may invert the low-pass filtered presynaptic rate back to the presynaptic membrane potential, $\bar{r}_{l+1} \rightarrow u_{l+1}$, provided that the recovery time constant $\tau$ matches the membrane time constant (see end of Results and Appendix 1).

Apical dendrites of pyramidal neurons in each layer receive top-down input from the pyramidal population in the upper layer through synaptic weights $B_l$. These top-down weights could be learned to predict the lower-layer activity (*Rao and Ballard, 1999*) or to become the transpose of the forward

weight matrix ($\boldsymbol{B}_l = \boldsymbol{W}_{l+1}^{\mathrm{T}}$, **Max et al., 2022**), but for simplicity, we randomly initialized them and keep them fixed (**Lillicrap et al., 2020**). Besides the top-down projections, the apical dendrites also receive lateral input via an interneuron population in the same layer through synaptic weights $-\boldsymbol{W}^{\mathrm{PI}}{}_l$ that are plastic and will be learned to obtain suitable dendritic errors. The '-' sign is suggestive of these interneurons to subtract away the top-down input entering through $\boldsymbol{B}_l$ (while the weights can still be positive or negative). Assuming again a conversion of rates to voltages, also for the inhibitory neurons that may operate in a linear regime, the overall apical voltage becomes

$$\bar{e}_l^A = \boldsymbol{B}_l \boldsymbol{u}_{l+1} - \boldsymbol{W}^{\mathrm{PI}}{}_l \boldsymbol{u}^{\mathrm{I}}{}_l. \tag{10}$$

What cannot be explained away from the top-down input $\boldsymbol{B}_l \boldsymbol{u}_{l+1}$ by the lateral feedback, $-\boldsymbol{W}^{\mathrm{PI}}{}_l \boldsymbol{u}^{\mathrm{I}}{}_l$, remains as dendritic prediction error $\bar{e}_l^A$ in the apical tree (**Figure 5a**). If the top-down and lateral feedback weights are learned as outlined next, these apical prediction errors take the role of the backpropagated errors in the classical backprop algorithm.

To adjust the interneuron circuit in each layer ('area'), the synaptic strengths from pyramidal-to-interneurons, $\boldsymbol{W}^{\mathrm{IP}}{}_l$, are learned to minimize the interneuron mismatch energy, $E_l^{\mathrm{IP}} = \frac{1}{2}\|\boldsymbol{u}_l^{\mathrm{I}} - \boldsymbol{W}_l^{\mathrm{IP}} \bar{\boldsymbol{r}}_l\|^2$. The interneurons, while being driven by the lateral inputs $\boldsymbol{W}^{\mathrm{IP}} \bar{\boldsymbol{r}}_l$, learn to reproduce the upper-layer activity that also nudges the interneuron voltage. Learning is accomplished if the upper-layer activity, in the absence of an additional error on the upper layer, is fully reproduced in the interneurons by the lateral input.

Once the interneurons learn to represent the 'error-free' upper-layer activity, they can be used to explain away the top-down activities that also project to the apical trees. The synaptic strengths from the inter-to-pyramidal neurons, $\boldsymbol{W}^{\mathrm{PI}}{}_l$, are learned to minimize the apical mismatch energy, $E_l^{\mathrm{PI}} = \frac{1}{2}\|\bar{e}_l^A\|^2 = \frac{1}{2}\|\boldsymbol{B}_l \boldsymbol{u}_{l+1} - \boldsymbol{W}^{\mathrm{PI}}{}_l \boldsymbol{u}^{\mathrm{I}}{}_l\|^2$. While in the absence of an upper-layer error, the top-down activity $\boldsymbol{B}_l \boldsymbol{u}_{l+1}$ can be fully cancelled by the interneuron activity $\boldsymbol{W}^{\mathrm{PI}}{}_l \boldsymbol{u}^{\mathrm{I}}{}_l$, a neuron-specific error will remain in the apical dendrites of the lower-level pyramidal neurons if there is an error endowed in the upper-layer neurons. Gradient descent learning on these two energies results in the learning rules for the P-to-I and I-to-P synapses,

$$\dot{\boldsymbol{W}}_l^{\mathrm{IP}} = \eta^{\mathrm{IP}} \left( \boldsymbol{u}_l^{\mathrm{I}} - \boldsymbol{W}_l^{\mathrm{IP}} \bar{\boldsymbol{r}}_l \right) \bar{\boldsymbol{r}}_l^{\mathrm{T}} \quad \text{and} \quad \dot{\boldsymbol{W}}_l^{\mathrm{PI}} = \eta^{\mathrm{PI}} \left( \boldsymbol{B}_l \boldsymbol{u}_{l+1} - \boldsymbol{W}_l^{\mathrm{PI}} \boldsymbol{u}_l^{\mathrm{I}} \right) \boldsymbol{u}_l^{\mathrm{I}^{\mathrm{T}}}. \tag{11}$$

The following theorem on dendritic error learning tells that the plasticity in the lateral feedback loop leads to an appropriate error representation in the apical dendrites of pyramidal neurons.

## Theorem 2 (real-time dendritic error learning, rt-DeEL)

Consider a cortical microcircuit composed of pyramidal and interneurons, as illustrated in **Figure 5a**, with more interneurons in layer ('cortical area') $l$ than pyramidal neurons in layer $l+1$, and with adaptable pyramidal-to-inhibitory weights $\boldsymbol{W}^{\mathrm{IP}}{}_l$ within the same layer that are nudged through top-down weights $\boldsymbol{B}^{\mathrm{IP}}{}_l$, see Methods, Sect. Proving theorem 2 (rt-DeEL). Then, for suitable top-down nudging, learning rates, and initial conditions, the inhibitory-to-pyramidal synapses $\boldsymbol{W}^{\mathrm{PI}}{}_l$ within each layer $l$ (**Equation 11**) evolve such that the lateral feedback circuit aligns with the bottom-up-top-down feedback circuit,

$$\boldsymbol{W}^{\mathrm{PI}}{}_l \boldsymbol{W}^{\mathrm{IP}}{}_l = \boldsymbol{B}_l \boldsymbol{W}_{l+1}. \tag{12}$$

After this horizontal-to-vertical circuit alignment, the apical voltages $\bar{e}_l^A = \boldsymbol{B}_l \boldsymbol{u}_{l+1} - \boldsymbol{W}^{\mathrm{PI}}{}_l \boldsymbol{u}^{\mathrm{I}}{}_l$ of the layer-$l$ pyramidal neurons (**Equation 13**) represent the '$B$-backpropagated' errors, $\bar{e}_l^A = \boldsymbol{B}_l \bar{e}_{l+1}$. When modulated by the postsynaptic rate derivatives, $\bar{\boldsymbol{r}}_l' = \rho'(\bar{\boldsymbol{u}}_l)$, the apical voltages yield the appropriate error signals

$$\bar{e}_l = \boldsymbol{u}_l - \boldsymbol{W}_l \bar{\boldsymbol{r}}_{l-1} = \bar{\boldsymbol{r}}_l' \cdot \bar{e}_l^A = \bar{\boldsymbol{r}}_l' \cdot \boldsymbol{B}_l \bar{e}_{l+1} \tag{13}$$

for learning the forward weights $\boldsymbol{W}_l$ according to $\dot{\boldsymbol{W}}_l \propto \bar{e}_l \bar{\boldsymbol{r}}_{l-1}^{\mathrm{T}}$ **Equation 8**.

The back projecting weights can also be learned by a local real-time learning rule to become transpose of the forward weights, $\boldsymbol{B}_l = \boldsymbol{W}_{l+1}^{\mathrm{T}}$ (**Max et al., 2022**). In this case, the error signals $\bar{e}_l$ learned in the apical dendrites according to the above Theorem (**Equation 13**) represent the gradient errors $\bar{e}$ appearing in the real-time dendritic error propagation (rt-DeEP, Theorem 1). There, the errors $\bar{e}$ drive

the gradient plasticity of the general weight matrix $W$, split up here into the forward weights $W_l$ to a layer $l$ (for $l = 1, .., N$).

## Simultaneously learning apical errors and basal signals

Microcircuits following these neuronal and synaptic dynamics are able to learn the classification of hand-written digits from the MNIST dataset while learning the apical signal representation (*Figure 5b1, b2*). In this case, feedforward weights $W_l$ and lateral weights $W^{PI}_l$ and $W^{IP}_l$ are all adapted simultaneously. Including the $\dot{W}^{IP}_l$-plasticity (by turning on the interneuron nudging from the upper layer, $\beta^I > 0$ in *Equation 9*), greatly speeds up the learning.

With and without $\dot{W}^{IP}_l$-plasticity, the lateral feedback via interneurons (with effective weight $W^{IP}_l W^{PI}_l$) learns to align with the forward-backward feedback via upper layer pyramidal neurons (with effective weight $B_l W_{l+1}$, *Figure 5b3*). The microcircuit extracts the gradient-based errors (*Equation 13*), while the forward weights use these errors to reduce these errors to first minimize the neuron-specific mismatch errors, and eventually the output cost.

Since the apical voltage $\bar{e}^A_l$ appears as a postsynaptic factor in the plasticity rule for the interneurons ($\dot{W}^{PI}_l$), this I-to-P plasticity can be interpreted as Hebbian plasticity of inhbitory neurons, consistent with earlier suggestions (*Vogels et al., 2011*; *Bannon et al., 2020*). The plasticity $\dot{W}^{IP}_l$ of the P-to-I synapses, in the same way as the plasticity for the forward synapses $\dot{W}_l$, can be interpreted as learning from the dendritic prediction of somatic activity (*Urbanczik and Senn, 2014*).

Crucially, by choosing a large enough interneuron population, the simultaneous learning of the lateral microcircuit and the forward network can be accomplished without fine-tuning of parameters. As an instance in case, all weights shared the same learning rate. Such stability bolsters the biophysical plausibility of our NLA framework and improves over the previous, more heuristic approach (*Sacramento et al., 2018*; *Mesnard et al., 2019*). The stability may be related to the nested gradient descent learning according to which somatic and apical mismatch errors in pyramidal neurons, and somatic mismatch errors in inhibitory neurons are minimized.

Finally, since errors are defined at the level of membrane voltages (*Equation 11*), synapses need a mechanism by which they can recover the presynaptic voltage errors from their afferent firing rates. While for threshold-linear transfer functions the backpropagated voltage errors translate into rate errors (Appendix 1), more general neuronal nonlinearities must be matched by corresponding synaptic nonlinearities. *Pfister et al., 2010* have illustrated how spiking neurons can leverage short-term synaptic depression to estimate the membrane potential of their presynaptic partners. Here, we assume a similar mechanism in the context of our rate-based neurons. The monotonically increasing neuronal activation function, $\bar{r}_{l+1} = \rho(u_{l+1})$, can be approximately compensated by a vesicle release probability that monotonically decreases with the low-pass filtered presynaptic rate $\bar{r}_{l+1}$ (see Appendix 1 and *Appendix 1—figure 1*). If properly matched, this leads to a linear relationship between the presynaptic membrane potential $u_{l+1}$ and the postsynaptic voltage contribution.

## Discussion

We introduced a least-action principle to neuroscience for deriving the basic laws of the voltage and synaptic dynamics in networks of cortical neurons. The approach is inspired by the corresponding principle in physics where basic laws of motion are derived across the various scales. While in physics the action is defined as the time-integral of the kinetic minus potential energy, we define the action as the time-integral of instantaneous somatodendritic mismatch errors across network neurons plus a behavioral error. The 'kinetics' of a voltage trajectory only arises because we postulate that the action along a trajectory is minimized with respect to future voltages, not the instantaneous voltage, as would be done in physics. The postulate implies a prospective voltage dynamics that look ahead in time, together with prospective local errors, in order to minimize the action and hence the somatodendritic mismatch errors. The prospective errors nudge the firing of pyramidal neurons deep in the brain, so that motor neurons improve the output of the network right in time. A putative behavioral error, encoded in the motor feedback, propagates back through the network and produces prospective corrections of the pyramidal neuron activities that effectively manifest in instantaneous corrections of the motor trajectory. Through this prospective coding, the sensory stream, the deep network activity, and the motor feedback are in sync at any moment in time. We formulated the

dynamic synchronization as a 'moving equilibrium hypothesis', referring to the classical equilibrium point hypothesis for motor control (*Feldman and Levin, 2009*; *Latash, 2010*). More generally, the brain activity formed by the prospective firing of cortical pyramidal neurons is in a moving equilibrium while converting sensory input streams into motor outputs, consistent with prospective sensory processing in the human cortex (*Blom et al., 2020*).

Because the neuronal dynamics derived from the global NLA principle is in a moving equilibrium, the prospective dendritic errors that globally correct the output trajectory are also suited to instruct local synaptic plasticity in the dendrites. In fact, working down the apical errors by adapting the sensory-driven synapses on the basal dendrites reduces the global output errors in real time. The apical errors are extracted from the top-down feedback via lateral 'inhibition' that tries to cancel the top-down signal. This top-down feedback includes activity from a putative erroneous motor output that was not foreseen by the local inhibition and thus survives as a local apical error. Given the prospective coding of the pyramidal neurons, the dendritic errors are also prospective and thus able to induce the correct error-minimizing plasticity online, while stimuli and targets continuously change.

## The NLA principle as a bottom-up theory from neurons to behavior

To show that the NLA principle offers a viable program for a formalization of neuroscience following the example of physics, we exemplified its ramifications in dendritic computation, cortical microcircuits, synaptic plasticity, motor control, and sensory-based decision-making. The crucial point of our axiomatization is that it connects the local neuronal errors to the global behavioral errors right in the formulation of the principle, eventually leading to local gradient-based plasticity rules. Because the formulation builds upon computations that can be realized in single neurons and dendrites to produce a behavioral output, the NLA principle can be seen as a bottom-up theory of behavior. It is articulated in terms of apical and basal dendrites, somatic firing, network connectivity and behavioral outputs that jointly minimize their errors. This contrasts the related free energy principle, for instance, that leads to a top-down theory of behavior by starting with the statistical, but the more universal, notion of a free energy. It postulates that any self-organizing system, that is at a statistical equilibrium with its environment, must minimize its free energy (*Friston, 2010*; *Friston et al., 2022*), and from there work down its way to neurons and dendrites (*Bastos et al., 2012*; *Kiebel and Friston, 2011*).

Starting with a single Lagrangian function that specifies the form of the somatodendritic prediction errors leaves some freedom for the interpretation and the implementation of the emerging dynamical equations for the voltages. We interpret errors to be represented in the apical dendrites of pyramidal neurons while sensory input targets the basal dendrites, but other dendritic configurations are conceivable (*Mikulasch et al., 2023*) that apply also to non-pyramidal neurons. We have chosen a specific interneuron circuitry to extract our apical errors, but other microcircuits or error representations might also be considered (*Keller and Mrsic-Flogel, 2018*). On the other hand, the derived gradient-based synaptic plasticity is tightly linked to the specific form of the somatodendritic prediction errors expressed in the Lagrangian and its interpretation, making specific predictions for synaptic plasticity (as outlined below). The 'external' feedback entering through the cost function offers additional freedom to model behavioral interactions. We considered an explicit time course of a target voltage in motor neurons, for instance imposed by the feedback from muscle spindles that are themselves innervated by a prospective top-down signal to control muscle lengths (*Papaioannou and Dimitriou, 2021*; *Dimitriou, 2022*). But the cost may also link to reinforcement learning and express a delayed reward feedback delivered upon a behavioral decision of an agent acting in a changing environment (*Friedrich et al., 2011*; *Friedrich and Senn, 2012*).

A fundamental difficulty arises when the neuronal implementation of the Euler-Lagrange equations requires an additional microcircuit with its own dynamics. This is the case for the suggested microcircuit extracting the local errors. Formally, the representation of the apical feedback errors first needs to be learned before the errors can teach the feedforward synapses on the basal dendrites. We showed that this error learning can itself be formulated as minimizing an apical mismatch energy. What the lateral feedback through interneurons cannot explain away from the top-down feedback remains an apical prediction error. Ideally, while the network synapses targeting the basal tree are performing gradient descent on the global cost, the microcircuit synapses involved in the lateral feedback are performing gradient descent on local error functions, both at any moment in time. The simulations show that this intertwined system can in fact learn simultaneously with a common learning rate that

is properly tuned. The cortical model network of inter- and pyramidal neurons learned to classify handwritten digits on the fly, with 10-digit samples presented per second. Yet, the overall learning is more robust if the error learning in the apical dendrites operates in phases without output teaching but with corresponding sensory activity, as may arise during sleep (see e.g., *Deperrois et al., 2022*; *Deperrois et al., 2024*).

## The NLA principle integrates classical theories for cortical processing and learning

The prospective variational principle introduced with the NLA allows for integrating previous ideas on formalizing the processing and learning in cortical networks. Four such classical lines of theories come together. (*i*) The first line refers to the use of an energy function to jointly infer the neuronal dynamics and synaptic plasticity, originally formulated for discrete-time networks (*Hopfield, 1982*; *Ackley et al., 1985*), and recently extended to continuous-time networks (*Scellier and Bengio, 2017*). (*ii*) The second line refers to understanding error-backpropagation in the brain (*Rumelhart et al., 1986*; *Xie and Seung, 2003*; *Whittington and Bogacz, 2017*; *Whittington and Bogacz, 2019*; *Lillicrap et al., 2020*). (*iii*) The third line refers to dendritic computation and the use of dendritic compartmentalization for various functions such as nonlinear processing (*Schiess et al., 2016*; *Poirazi and Papoutsi, 2020*) and deep learning (*Guerguiev et al., 2017*; *Sacramento et al., 2018*; *Haider et al., 2021*). (*iv*) The fourth line refers to *predictive coding* (*Rao and Ballard, 1999*) and active inference (*Pezzulo et al., 2022*) to improve the sensory representation and motor output, respectively.

(i) With regard to *energy functions*, the NLA principle adds a variational approach to characterize continuous-time neuronal trajectories and plasticity. Variational approaches are studied in the context of optimal control theory where a cost integral is minimized across time, constrained to some network dynamics (*Todorov and Jordan, 2002*; *Meulemans et al., 2021*). The NLA represents a unifying notion that allows us to infer both, the network dynamics and its optimal control from a single Lagrangian. The error we derive represents prospective control variables that are applied to the voltages of each network neuron so that they push the output neurons towards their target trajectory. The full expression power of this control theoretic framework has yet to be proven when it is extended to genuine temporal processing that includes longer time constants, for instance, inherent in a slow threshold adaptation (*Bellec et al., 2020*). The NLA principle can also treat the case of strong feedback studied so far in relaxation networks only (*Meulemans et al., 2022*; *Song et al., 2024*). Our rt-DeEP Theorem makes a statement for real-time gradient descent learning while the network is in a moving equilibrium, linking to motor learning in the presence of perturbing force fields (*Herzfeld et al., 2014*) or perturbing visual inputs (*Dimitriou, 2016*).

(ii) With regard to *error-backpropagation*, the NLA principle infers a local error that originates from the error in the output neurons. In the current version, the NLA relies on feedback alignment (*Lillicrap et al., 2016*) to obtain a local apical error without postulating plasticity in the top-down synapses. Other works have explored learning the feedback weights (*Akrout et al., 2019*; *Kunin et al., 2020*), notably within a phase-less real-time learning framework as considered here (*Max et al., 2022*). It would be promising to combine these ideas to obtain a fully plastic microcircuit that adjusts from scratch to various real-time learning tasks.

(iii) With regard to *dendritic computation*, the NLA principle extends the idea of dendritic error representation (*Sacramento et al., 2018*; *Mesnard et al., 2019*) by the prospective coding of both errors and firing rates (*Figure 2*). As a consequence, the various dendritic delays are compensated and synaptic plasticity can operate at any moment, without need to wait for network relaxations (*Scellier and Bengio, 2017*; *Song et al., 2024*). In the present framework, the input rates are low-pass filtered by variable input time constants ($\tau_{\text{in}}$) before the induced voltages are instantaneously processed in the network. While this offers the possibilities for a temporal processing of the inputs, step changes in the input rates cannot instantaneously propagate as made possible in *Haider et al., 2021*. The dendritic error representation has also been applied to spike-based learning in recurrent networks (*Mikulasch et al., 2021*).

(iv) With regard to the principle of *predictive coding* (*Rao and Ballard, 1999*; *Pezzulo et al., 2022*), the NLA offers a form of predictive error-coding in the apical tree of sensory pyramidal neurons that shapes the sensory representation. Both the predictive coding and the NLA

principle imply an error-based adaptation of the lower and higher cortical representations. However, while predictive coding is based on propagating errors, the NLA principle directly propagates the error-corrected representations. Based on the backpropagated activities, it extracts in each area the prospective errors to reshape the local representations and induce synaptic plasticity. Otherwise, the freedom in defining the cost function in the NLA, and the inclusion of active inference in predictive coding (*Pezzulo et al., 2022*), make the two frameworks equally powerful. Both frameworks can explain the learning from motor and sensory prediction errors. In fact, our example of motor control minimizes the proprioceptive error formed by the target muscle lengths minus the effective muscle lengths. Active inference likewise minimizes the mismatch between an internal motor target and the proprioceptive inputs caused by the own actions. In turn, the cost function in the NLA principle may also capture the sensory prediction errors, where the ground truth lies in the external inputs that are learned to be matched by the sensory expectation. Hence, while being functionally equivalent, the neuronal interpretations are different: predictive coding focuses on the propagation of errors using the sensory- and motor representations as local auxiliary quantities, and the NLA principle directly integrates errors in 'auxiliary dendrites' to propagate sensory or motor representations.

## The NLA integrates and predicts features of synapses, dendrites, and circuits

Motivated by the predictive power of the least-action principle in physics, we ask about experimental confirmation and predictions of the NLA principle. Given its axiomatic approach, it appears astonishing to find various preliminary matches at the dendritic, somatic, interneuron, synaptic, and even behavioral levels. Some of these are:

1. the prospective coding of pyramidal neuron firing (*Köndgen et al., 2008*);
2. the prospective processing of apical signals while propagating to the soma (*Ulrich, 2002*);
3. the basal synaptic plasticity on pyramidal neurons and synaptic plasticity on interneurons, driven by the postsynaptic activity that is 'unexplained' by the distal dendritic voltage (*Urbanczik and Senn, 2014*) – while partly consistent with spike-timing dependent plasticity (*Sjöström et al., 2001*; *Spicher et al., 2017*), the postulated dendritic voltage-dependence of the plasticity rules still awaits an experimental inspection;
4. the Hebbian homeostatic plasticity of interneurons targeting the apical dendritic tree of pyramidal neurons (*Bannon et al., 2020*);
5. the short-term synaptic depression at top-down synapses targetting inhibitory neurons and apical dendrites (akin to *Abbott et al., 1997*), but with a faster recovery time constant that invert the presynaptic activation function (see also *Pfister et al., 2010*);
6. the modulation of the apical contribution to the somatic voltage by the slope of the somatic activation function for instance by downregulating apical NMDA receptors with increasing rate of backpropagating action potentials, *Theis et al., 2018*; and
7. the involvement of muscle spindles in the prospective encoding of motor errors during motor learning, with the $\gamma$-innervation setting the target length of the spindles (*Dimitriou, 2016*; *Papaioannou and Dimitriou, 2021*; *Vargas et al., 2023*).

More experimental and theoretical work is required to substantiate these links and test specific predictions, such as the apical error representation in cortical pyramidal neurons.

Overall, our approach adapts the least-action principle from physics to be applied to neuroscience, and couples it with a normative perspective on the prospective processing of neurons and synapses in global cortical networks and local microcircuits. Given its physical underpinnings, the approach may inspire the rebuilding of computational principles of cortical neurons and circuits in neuromorphic hardware (*Bartolozzi et al., 2022*). A step in this direction, building on the instantaneous computational capabilities by slowly integrating neurons, has made done by *Haider et al., 2021*. Given its aspiration for a theoretical framework in neurobiology, a next challenge would be to generalize the NLA principle to spiking neurons (*Gerstner and Kistler, 2002*; *Brendel et al., 2020*) with their potential for hardware implementation (*Zenke and Ganguli, 2018*; *Göltz et al., 2021*; *Cramer et al., 2022*), to include attentional mechanisms in terms of dendritic gain modulation (*Larkum et al., 2004*) with a putative link to self-attention in artificial intelligence (*Vaswani et al., 2017*), to add second-order errors to cope with certainties (*Granier et al., 2023*), and to incorporate longer temporal processing

as, for instance, offered by neuronal adaptation processes (*La Camera et al., 2006*) or realistically modelled dendrites (*Chavlis and Poirazi, 2021*).

## Methods
### Euler-Lagrange equations as inverse low-pass filters

The theory is based on the look-ahead of neuronal quantities. In general, the look-ahead of a trajectory $x(t)$ is defined via lookahead operator applied to $x$,

$$\left(1 + \tau \tfrac{\mathrm{d}}{\mathrm{d}t}\right)x = x + \tau\dot{x}. \tag{14}$$

The lookahead operator is the inverse of the low-pass filter operator denoted by a bar,

$$\bar{x}(t) = \frac{1}{\tau}\int_{-\infty}^{t} x(t')e^{-\frac{t-t'}{\tau}}\,\mathrm{d}t'\,. \tag{15}$$

This low-pass filtering can also be characterized by the differential equation $\tau\dot{\bar{x}}(t) = -\bar{x}(t) + x(t)$, see Appendix 2. Hence, applying the low-pass filtering to $x$ and then the lookahead operator $\left(1 + \tau \tfrac{\mathrm{d}}{\mathrm{d}t}\right)$ to $\bar{x}(t)$, and using the Leibnitz rule for differentiating an integral, we calculate $\left(1 + \tau \tfrac{\mathrm{d}}{\mathrm{d}t}\right)\bar{x}(t) = x(t)$. In turn, applying first the lookahead, and then the low-pass filtering, also yields the original trace back, $\overline{\left(1 + \tau \tfrac{\mathrm{d}}{\mathrm{d}t}\right)x} = \bar{x} + \tau\dot{\bar{x}} = x$.

We consider an arbitrary network architecture with network neurons that are recurrently connected and that receive external input through an overall weight matrix $W = (W_{\text{in}}, W_{\text{net}})$, aggregated column-wise. The instantaneous presnyaptic firing rates are $r = (r_{\text{in}}, r_{\text{net}})^{\text{T}}$, interpreted as a single-column vector. A subset of network neurons are output neurons, $\mathcal{O} \subseteq \mathcal{N}$, for which target voltages $u^*$ may be imposed. Rates and voltages may change in time $t$. Network neurons are assigned a voltage $u$, generating the low-pass filtered rate $\bar{r}_{\text{net}} = \rho(u)$, and a low-pass filtered error $\bar{e} = u - W\bar{r}$. We further define output errors $\bar{e}_o^* = u_o^* - u_o$ for $o \in \mathcal{O}$, and $\bar{e}_i^* = 0$ for non-output neurons $i \in \mathcal{N} \setminus \mathcal{O}$. With this, the Lagrangian from *Equation 3* takes the form

$$L = \frac{1}{2}\|\bar{e}\|^2 + \frac{\beta}{2}\|\bar{e}^*\|^2. \tag{16}$$

We next use that $u = \tilde{u} - \tau\dot{\tilde{u}}$, with the $\tilde{\phantom{u}}$ operator defined in *Equation 4*, to write out the Lagrangian $L$ in the canonical coordinates $(\tilde{u}, \dot{\tilde{u}})$ as (see also *Equation 3*)

$$L = \frac{1}{2}\sum_{i \in \mathcal{N}}\left[\tilde{u}_i - \tau\dot{\tilde{u}}_i - \sum_j W_{ij}\rho(\tilde{u}_j - \tau\dot{\tilde{u}}_j)\right]^2 + \frac{\beta}{2}\sum_{o \in \mathcal{O}}\left[u_o^* - (\tilde{u}_o - \tau\dot{\tilde{u}}_o)\right]^2\,. \tag{17}$$

The neuronal dynamics is derived from requiring a stationary action (see *Equation 5*), which is generally solved by the Euler-Lagrange equations $\frac{\partial L}{\partial \tilde{u}_i} - \frac{\mathrm{d}}{\mathrm{d}t}\frac{\partial L}{\partial \dot{\tilde{u}}_i} = 0$ (see *Equation 6*). Because $\tilde{u}$ only arises in $L$ in the compound $\tilde{u} - \tau\dot{\tilde{u}}$, the derivative of $L$ with respect to $\tilde{u}$ is identical to the derivative with respect to $\tau\dot{\tilde{u}}$,

$$\frac{\partial L}{\partial \dot{\tilde{u}}_i} = -\tau\frac{\partial L}{\partial \tilde{u}_i}. \tag{18}$$

Using the lookahead operator *Equation 14*, the Euler-Lagrange equations can then be rewritten as

$$\frac{\partial L}{\partial \tilde{u}_i} + \tau\frac{\mathrm{d}}{\mathrm{d}t}\frac{\partial L}{\partial \tilde{u}_i} = \left(1 + \tau\frac{\mathrm{d}}{\mathrm{d}t}\right)\frac{\partial L}{\partial \tilde{u}_i} = 0. \tag{19}$$

Since $L(\tilde{u}, \dot{\tilde{u}}) = L(u)$ and $u = \tilde{u} - \tau\dot{\tilde{u}}$, the derivative of $L$ with respect to $\tilde{u}$ is the same as the derivative of $L$ with respect to $u$,

$$\frac{\partial L}{\partial \tilde{u}_i} = \frac{\partial L}{\partial u_i}\,.$$

Plugging this into *Equation 19*, the Euler-Lagrange equations become a function of $u$ and $\dot{u}$,

$$\left(1 + \tau \frac{\mathrm{d}}{\mathrm{d}t}\right) \frac{\partial L}{\partial u_i} = 0. \tag{20}$$

Notice that, if we had directly calculated $\frac{\partial L}{\partial \bar{u}_i} - \frac{\mathrm{d}}{\mathrm{d}t} \frac{\partial L}{\partial \dot{\bar{u}}_i} = 0$, the second-order time derivative $\ddot{u}_i$ of the discounted future voltage would be absorbed in a first-order time derivative of the voltage. The reason is that $\dot{\bar{u}}_i - \tau \ddot{\bar{u}}_i = \dot{u}_i$, and $\ddot{\bar{u}}_i$ only arises in this combination because the Lagrangian $L = L(\boldsymbol{u})$ is only a function of $\boldsymbol{u}$ and not of $\dot{\boldsymbol{u}}$. Hence, the acceleration term $\ddot{u}_i$ disappears, while a voltage derivative $\dot{u}_i$ appears.

The solution of this differential *Equation 20* is $\frac{\partial L}{\partial u_i} = c_i\, e^{-\frac{t-t_0}{\tau}}$, and hence any trajectory $(\tilde{u}_i, \dot{\tilde{u}}_i)$ which satisfy the Euler-Lagrange equations will hence cause $\frac{\partial L}{\partial u_i}$ to converge to zero with a characteristic time scale of $\tau$. Since we require that the initialisation is at $t_0 = -\infty$, we conclude that $\frac{\partial L}{\partial u_i} = 0$, as required in the rt-DeEP Theorem. For a table with all the mathematical abbreviations see Methods-*Table 1*.

## Deriving the network dynamics from the Euler-Lagrange equations

We now derive the equations of motion from the Euler-Lagrange equations. Noticing that $\boldsymbol{u}$ enters in $\bar{\boldsymbol{e}} = \boldsymbol{u} - \boldsymbol{W}\bar{\boldsymbol{r}}$ twice, directly and through $\bar{\boldsymbol{r}}_{\mathrm{net}} = \rho(\boldsymbol{u})$, and once in the output error $\bar{\boldsymbol{e}}^*$, we calculate from 16, using $\bar{\boldsymbol{r}}(\boldsymbol{u}) = (\bar{\boldsymbol{r}}_{\mathrm{in}}, \rho(\boldsymbol{u}))^{\mathrm{T}}$ and $\boldsymbol{W} = (\boldsymbol{W}_{\mathrm{in}}, \boldsymbol{W}_{\mathrm{net}})$,

$$\frac{\partial L}{\partial \boldsymbol{u}} = \bar{\boldsymbol{e}} - \bar{\boldsymbol{\epsilon}} - \beta \bar{\boldsymbol{e}}^* , \quad \text{with} \quad \bar{\boldsymbol{\epsilon}} = \bar{\boldsymbol{r}}'_{\mathrm{net}} \boldsymbol{W}_{\mathrm{net}}^{\mathrm{T}} \bar{\boldsymbol{e}}. \tag{21}$$

Remember that for non-output neurons $i$ no target exists, and for those we set $\bar{\boldsymbol{e}}_i^* = 0$. Next, we apply the lookahead operator to this expression, as required by the Euler-Lagrange *Equation 19*. In general $\left(1 + \tau \frac{\mathrm{d}}{\mathrm{d}t}\right) \bar{x} = \bar{x} + \tau \dot{\bar{x}} = x$, and we set for $\bar{x}$ the expression on the right-hand side of *Equation 21*, $\bar{x} = \bar{\boldsymbol{e}} - \bar{\boldsymbol{\epsilon}} - \beta \bar{\boldsymbol{e}}^*$, which at the same time is $\bar{x} = \frac{\partial L}{\partial \boldsymbol{u}}$. Hence, the Euler-Lagrange equations in the form of *Equation 20*, $\left(1 + \tau \frac{\mathrm{d}}{\mathrm{d}t}\right)\bar{x} = 0$, translate into

$$\left(1 + \tau \frac{\mathrm{d}}{\mathrm{d}t}\right) \frac{\partial L}{\partial \boldsymbol{u}} = 0 \iff \boldsymbol{e} - \boldsymbol{\epsilon} - \beta \boldsymbol{e}^* = 0 \iff \tau \dot{\boldsymbol{u}} = -\boldsymbol{u} + \boldsymbol{W}\boldsymbol{r} + \boldsymbol{e}. \tag{22}$$

To move from the middle to the last equality we replaced $\boldsymbol{e}$ with $\boldsymbol{e} = \left(1 + \tau \frac{\mathrm{d}}{\mathrm{d}t}\right)\bar{\boldsymbol{e}} = \boldsymbol{u} + \tau \dot{\boldsymbol{u}} - \boldsymbol{W}\boldsymbol{r}$. In the last equality we interpret $\boldsymbol{e}$ as the sum of the two errors, $\boldsymbol{e} = \boldsymbol{\epsilon} + \beta \boldsymbol{e}^*$, again using the middle equality. This proves *Equation 7a*, *Equation 7b*.

Notice that the differential equation $\tau \dot{\boldsymbol{u}} = \ldots$ in *Equation 22* represents an implicit ordinary differential equation as on the right-hand side not only $\boldsymbol{u}$, but also $\dot{\boldsymbol{u}}$ appears (in $\boldsymbol{r}$ and $\boldsymbol{e}$). The uniqueness of the solution $\boldsymbol{u}(t)$ for a given initial condition is only guaranteed if it can be converted into an explicit ordinary differential equation (see Sect. Appendix 3).

In taking the temporal derivative we assumed small learning rates such that terms including $\dot{W}_{ij}$ can be neglected. The derived dynamics for the membrane potential of a neuron $u_i$ in *Equation 22* show the usual leaky behavior of biological neurons. However, both presynaptic rates $\bar{r}_i$ and prediction errors $\bar{e}_i$ enter the equation of motion with lookaheads, i.e., they are advanced ($r_i = \bar{r}_i + \tau \dot{\bar{r}}_i$ and $e_i = \bar{e}_i + \tau \dot{\bar{e}}_i$), cancelling the low-pass filtering. Since $\dot{\bar{r}}_i = \rho'(u_i)\dot{u}_i$, the rate and error, $r_i$ and $e_i$, can also be seen as nonlinear extrapolations from the voltage and its derivative into the future.

The instantaneous transmission of information throughout the network at the level of the voltages can now be seen by low-pass filtering *Equation 22* with initialization far back in the past,

$$\boldsymbol{u} = \overline{\boldsymbol{u} + \tau \dot{\boldsymbol{u}}} = \overline{\boldsymbol{W}\boldsymbol{r} + \boldsymbol{e}} = \boldsymbol{W}\bar{\boldsymbol{r}}(\boldsymbol{u}) + \bar{\boldsymbol{e}} , \tag{23}$$

with column vector $\bar{\boldsymbol{r}}(\boldsymbol{u}) = (\bar{\boldsymbol{r}}_{\mathrm{in}}, \rho(\boldsymbol{u}))^{\mathrm{T}}$ and $\bar{\boldsymbol{e}} = \bar{\boldsymbol{r}}'_{\mathrm{net}} \boldsymbol{W}_{\mathrm{net}}^{\mathrm{T}} \bar{\boldsymbol{e}} + \beta \bar{\boldsymbol{e}}^*$. Hence, solving the voltage dynamics for $\boldsymbol{u}$ (*Equation 7a*), with apical voltage $\boldsymbol{e} = \bar{\boldsymbol{e}} + \tau \dot{\bar{\boldsymbol{e}}}$ derived from *Equation 7b*, yields the somatic voltage $\boldsymbol{u}$ satisfying the self-consistency *Equation 23* at any time. In other words, $\boldsymbol{u}$ and $\bar{\boldsymbol{e}}$ 'propagate instantaneously'.

## Deriving the error backpropagation formula

For clarity, we derive the error backpropagation algorithm for layered networks here. These can be seen as a special case of a general network with membrane potentials $\boldsymbol{u}$ and all-to-all weight matrix $\boldsymbol{W}$ (as introduced in Appendix 8), where the membrane potentials decompose into layerwise membrane

**Table 1.** Mathematical symbols.

| Mathematical expression | Naming | Comment |
|---|---|---|
| $u_i$ | Instantaneous (somatic) voltage | only for network neurons |
| $r_i = \rho(u_i) + \tau\dot\rho(u_i)$ | Instantaneous firing rate of neuron $i$ | that looks linearly ahead in time |
| $\bar{r}(t) = \frac{1}{\tau}\int_{-\infty}^{t} r(t')e^{-\frac{t-t'}{\tau}}\,\mathrm{d}t'$ | Definition of low-pass filtering | See **Equation 15** |
| $\bar{r}_i = \rho(u_i) = \overline{r_i + \tau\,\dot r_i}$ | Low-pass filtered firing rate | postulated to be a function of $u_i$ |
| $r = \bar{r} + \tau\dot{\bar{r}}$ | Self-consistency eq. | for low-pass filtered rate |
| $r_{\mathrm{in}}$ | Input rate vector, column | projects to selected neurons |
| $\bar{r}_{\mathrm{in}}$ | Low-pass filter input rates | instantaneously propagates |
| $e_i = (u_i + \tau\dot u_i) - \sum_j W_{ij}r_j$ | Prospective error of neuron $i$ | in apical dendrite |
| $\bar{e}_i = u_i - \sum_j W_{ij}\bar{r}_j$ | Error of neuron $i$ | in soma |
| $E^{\mathrm{M}}_i = \frac{1}{2}\bar{e}_i^2 = \frac{1}{2}\left(u_i - \sum_j W_{ij}\bar{r}_j\right)^2$ | Mismatch energy in neuron $i$ | between soma and basal dendrite |
| $u_o^*$ | Target voltage for output neuron $o$ | could impose target on $r_o$ or $\bar{r}_o$ |
| $\bar{e}_o^* = u_o^* - u_o$ | Error of output neuron $o$ | also called target error |
| $C_o = \frac{1}{2}(\bar{e}_o^*)^2$ | Cost contribution of output neuron $o$ | between soma and basal dendrite |
| $L = \sum_{i\in\mathcal{N}} E^{\mathrm{M}}_i + \beta\sum_{o\in\mathcal{O}} C_o$ | Lagrangian | output $\mathcal{O}\subset$ network $\mathcal{N}$ |
| $\tilde{u}(t) = \frac{1}{\tau}\int_t^{\infty} u(t')e^{(t-t')/\tau}\,\mathrm{d}t'$ | Discounted future voltage | prospective coordinates for NLA |
| $u = \tilde{u} - \tau\dot{\tilde{u}}$ | Self-consistency eq. | for discounted future voltage |
| $A = \int_{t_1}^{t_2} L\left[\tilde{u}(t),\dot{\tilde{u}}(t)\right]\,\mathrm{d}t$ | Neuronal Least Action (NLA) | expressed in prospect. coordinates |
| $\frac{\partial L}{\partial\tilde{u}_i} - \frac{\mathrm{d}}{\mathrm{d}t}\frac{\partial L}{\partial\dot{\tilde{u}}_i} = \left(1 + \frac{\mathrm{d}}{\mathrm{d}t}\right)\frac{\partial}{\partial u_i}L = 0$ | Euler-Lagrange equations | turned into lookahead operator |
| $W_{\mathrm{in}}$ | weights from input neurons $r_{\mathrm{in}}$ | $\dim(\mathcal{N})\times\dim(r_{\mathrm{in}})$, most $0$ |
| $W_{\mathrm{net}}$ | weights between network neurons | $\dim(\mathcal{N})\times\dim(\mathcal{N})$ |
| $W = (W_{\mathrm{in}}, W_{\mathrm{net}})$ | total weight matrix | $\dim(\mathcal{N})\times\left(\dim(r_{\mathrm{in}})+\dim(\mathcal{N})\right)$ |
| $r = (r_{\mathrm{in}}, r_{\mathrm{net}})^{\mathrm{T}}$ | instantaneous firing rate vector | column (indicated by transpose) |
| $\dot W \propto \bar{e}\,\bar{r}^{\mathrm{T}}$ | Plasticity of $W$ | $\bar{\mathbf{e}}$ is a column, $\bar{r}^{\mathrm{T}}$ a row vector |
| $u_o^*(t) = F^*(\bar{r}_{\mathrm{in}}(t))$ | Target function formulated for $\bar{r}_{\mathrm{in}}(t)$ | a functional of $r_{\mathrm{in}}(t)$ |
| $u_o(t) = F_W(\bar{r}_{\mathrm{in}}(t), \bar{e}_o^*(t))$ | Func. implemented by forward network | instant. func. of $\bar{r}_{\mathrm{in}}(t)$, not $r_{\mathrm{in}}(t)$ |
| $N$ | Layers in forward network, w/o or $r_{\mathrm{in}}$ | Last-layer voltages: $u_N = u_o$ |
| $W_l^{IP}$ | Weights from pyr to interneurons | lateral, within layer $l$ |

*Table 1 continued on next page*

*Table 1 continued*

| Mathematical expression | Naming | Comment |
|---|---|---|
| $W_l^{PI}$ | Weights from inter- to pyr'neurons | lateral, within layer $l$ |
| $W_l$ | Bottom-up weights from layer $l-1$ to $l$ | between pyramidal neurons |
| $B_l$ | Top-down weights from layer $l+1$ to $l$ | between pyramidal neurons |
| $\bar{e}_l^A = B_l u_{l+1} - W^{PI}{}_l u^I{}_l$ | Low-pass filtered apical error in layer $l$ | top-down minus lateral feedback |
| $\bar{e}_l = \bar{r}_l' \cdot \bar{e}_l^A = \bar{r}_l' \cdot B_l\, \bar{e}_{l+1}$ | Somato-basal prediction error | is correct error for learning |
| $E_l^{IP} = \frac{1}{2}\|u^I{}_l - W^{IP}{}_l \bar{r}_l\|^2$ | Interneuron mismatch energy | minimized to learn $W_l^{IP}$ |
| $E_l^{PI} = \frac{1}{2}\|B_l u_{l+1} - W^{PI}{}_l u^I{}_l\|^2$ | Apical mismatch energy | minimized to learn $W_l^{PI}$ |
| $\eta$, $\eta^{IP}$, $\eta^{PI}$ | Learning rates for plasticity of… | …$W_l$ ; $W_l^{IP}$ ; $W_l^{PI}$ |
| $H = \frac{\partial^2 L}{\partial u^2} = 1 - W_{net}\rho' - \bar{e}'$ | Hessian, $\frac{\partial^2 L}{\partial u^2} = \frac{\partial f}{\partial u}$. If pos. definite | $\Rightarrow$ stable dynamics |
| $f(u,t) = \frac{\partial L}{\partial u} = u - W\bar{r}(u) - \bar{e}(u)$ | Corrected error | becomes $0$ with $\tau$ |
| $f(u,t) + \tau \dot{f}(u,t) = 0$ | Euler-Lagrange equations | satisfy $f(u,t) = f_0\, e^{-(t-t_0)/\tau}$ |
| $f(u,t) = 0$ | Always the case after transient | exponentially decaying with $\tau$ |
| $\dot{u} = -\frac{1}{\tau}H^{-1}(u)\left(f(u) + \tau\frac{\partial f}{\partial t}\right)$ | Explicit diff. eq. | obtained by solving for $\dot{u}$ |
| $g(u,t) = -\frac{1}{\tau}H^{-1}(u)\left(f(u) + \tau\frac{\partial f}{\partial t}\right)$ | Used to write the explicit diff. eq. | $\dot{u} = g(u,t)$ |
| $G(y,\dot{u}) = \left(1 + \tau\frac{d}{dt}\right)\frac{\partial L}{\partial u} = f + \tau\dot{f}$ | Used for contraction anaylsis, *Equation 53* | $y = (r_{in}, u_o^*, u)$ |
| $M$, $K$ | Used to iteratively converge to $\dot{u}$ | see *Equation 46* |
| $\breve{u} = u + \tau\dot{u}$ | Linear lookahead voltage | Latent Equilibrium, Appendix 4 |

potential vectors $u_l$ and the weight matrix into according to block diagonal matrices $W_l$ (with $W_l$ being the weights that project into layer $l$).

Assuming a network with $N$ layers, by low-pass filtering the equations of motion we get

$$u_l = W_l\bar{r}_{l-1} + \bar{e}_l, \tag{24}$$

for all $l \in 1,..,N$, with the output error $\bar{e}_o$ of the general recurrent network becoming the error in the last layer, that itself is the target error, $\bar{e}_o = \bar{e}_N = \beta\bar{e}^* = \beta\left(u_N^* - u_N\right)$. The error $\bar{e} = \bar{\epsilon} + \beta\bar{e}^*$, that we obtain from the general dynamics with $\bar{\epsilon} = \bar{r}_{net}' W_{net}^T \bar{e}$, see *Equations 21 and 22*, translates to an iterative formula for the error at the current layer $l$ given the error at the downstream layer $l+1$, inherited from the drive $\bar{r}_l = \rho(u_l)$ of that downstream layer via $W_{l+1}$,

$$\bar{e}_l = \bar{r}_l' W_{l+1}^T \bar{e}_{l+1} \quad \text{for} \quad l < N. \tag{25}$$

and $\bar{e}_N = \beta\bar{e}^*$ for the output layer. The learning rule that reduces $\bar{e}_l$ by gradient descent is proportional to this error and the presynaptic rate, as stated by Theorem 1, is

$$\dot{W}_l \propto \left(u_l - W_l\bar{r}_{l-1}\right)\bar{r}_{l-1}^T = \bar{e}_l\,\bar{r}_{l-1}^T, \tag{26}$$

for $l = 1...N$. *Equations 25 and 26* together take the form of the error backpropagation algorithm, where an output error is iteratively propagated through the network and used to adjust the weights in order to reduce the output cost $C$. From this, it is easy to see that without output nudging (i.e. $\beta = 0$), the output error vanishes and consequently all other prediction errors vanish as well, $\bar{e}_l = u_l - W_l \bar{r}_l = 0$ for all $l \leq N$. This also means that in the absence of nudging, no weight updates are performed by the plasticity rule.

The learning rule for arbitrary connectivities is obtained in the same way by dropping the layer-wise notation. In this case, low-pass filtering the equations of motion yields $u = W\bar{r} + \bar{e}$, as calculated in 23, and the low-pass filtered error $\bar{e} = \bar{e} + \beta \bar{e}^* = \bar{r}'_{net} W^T_{net} \bar{e} + \beta \bar{e}^*$, as inferred from *Equations 21 and 22*. Hence, the plasticity rule in general reads

$$\dot{W} \propto (u - W\bar{r})\bar{r}^T = \bar{e}\,\bar{r}^T \ , \quad \text{with } \bar{e} = \bar{r}'_{net} W^T_{net} \bar{e} + \beta \bar{e}^*. \tag{27}$$

## Proving theorem 1 (rt-DeEP)

The implicit assumption in Theorem 1 is that $\dot{u}$ exists in the distributional sense for $t > -\infty$, which is the case for delta-functions in $r_{in}$ and step-functions in $u^*$. Both parts (i) and (ii) of the Theorem are based on the requirement of stationary action $\delta A = 0$, and hence on $u$ satisfying the Euler-Lagrange equations in the form of *Equation 22*, $\left(1 + \tau \frac{d}{dt}\right)\frac{\partial L}{\partial u} = 0$. From the solution $\frac{\partial L}{\partial u_i} = c\,e^{-\frac{t-t_0}{\tau}}$ we conclude that for initialization at $t_0 = -\infty$ we have $\frac{\partial L}{\partial u} = 0$ for all $t$. It is the latter stronger condition that we require in the proof. With this, the main ingredient of the proof follows is the mathematical argument of *Scellier and Bengio, 2017*, according to which the total and partial derivative of $L$ with respect to $W$ are identical, and this in our case is true for any time $t$,

$$\frac{dL}{dW} = \frac{\partial L}{\partial u}^T \frac{du}{dW} + \frac{\partial L}{\partial W} = \frac{\partial L}{\partial W}, \tag{28}$$

For convenience we considered $\frac{\partial L}{\partial u}$ to be a column vector, deviating from the standard notations (see tutorial end of sec:Integration). Analogously to *Equation 28*, we infer $\frac{dL}{d\beta} = \frac{\partial L}{\partial \beta}$. Reading *Equation 28* from the right to the left, we conclude that the learning rule $\dot{W} \propto -\frac{\partial L}{\partial W} = \bar{e}\,\bar{r}^T$ for all $\beta > 0$ is gradient descent on $L$, i.e., $\dot{W} \propto -\frac{dL}{dW}$. This total derivative of $L$ can be analyzed for large and small $\beta$.

(i) We show that in the limit of large $\beta$, $\dot{W}$ becomes gradient descent on the mismatch energy $E^M = \frac{1}{2}\|\bar{e}\|^2$. For this we first show that there is a solution of the self-consistency equation $u = F(u) = W\bar{r} + \bar{r}'_{net} W^T_{net} \bar{e} + \beta\,\bar{e}^*$ that is uniformly bounded for all $t$ and $\beta$. For this we assume that the transfer function $\rho(u)$ is non-negative, monotonically increasing, and bounded, that its derivative $\rho'(u)$ is bounded too, and that the input rates $r_{in}$ and the target potentials $u^*_o$ are also uniformly bounded. To show that under these conditions we always find a uniformly bounded solution $u(t)$, we first consider the case where the output voltages are clamped to the target, $u_o = u^*_o$ such that $\bar{e}^* = 0$. For simplicity, we assume that $\rho'(u) = 0$ for $|u| \geq c_0$. For voltages $u$ with $u_i \leq c_0$ the recurrent input current $W\bar{r}$ is bounded, say $|(W\bar{r})_j| \leq c_1$ for some $c_1 > c_0$. When including the error term $\bar{r}'_{net} W^T_{net} \bar{e}$, the total current still remains uniformly bounded, say $|F(u)_j| \leq c_2$ for all $u$ with $u_i \leq c_0$. Because for larger voltages $u_i > c_0$ the error term vanishes due to a vanishing derivative $\rho'(u_i) = 0$, the mapping $F(u)$ maps the $c_2$-box $u$ (for which $|u_i| \leq c_2$) onto itself. Brouwer's fixed point theorem then tells us that there is a fixed point $u = F(u)$ within the $c_2$-box. The theorem requires the continuity of $F$, and this is assured if the neuronal transfer $\bar{r} = \rho(u)$ is continuous.

We next relax the voltages of the output neurons from their clamped stage, $u_o = u^*_o$. Remember that these voltages satisfy $u_o = (W\bar{r} + \bar{r}'_{net} W^T_{net} \bar{e} + \beta\,\bar{e}^*)_o = F(u)_o$ at any time $t$. We determine the correction term $\beta\,\bar{e}^*_o$ such that in the limit $\beta \to \infty$ we get $u_o = F(u)_o = u^*_o$. The correction remains finite, and in the limit must be equal to $\lim_{\beta \to \infty} \beta\,\bar{e}^*_o = u^*_o - (W\bar{r} + \bar{r}'_{net} W^T_{net} \bar{e})_o$. For arbitrary large nudging strength $\beta$, the output voltage $u_o$ deviates arbitrary little from the target voltage, $u_o = u^*_o + o(1/\beta)$, with target error $\bar{e}^*_o = \frac{1}{\beta}\left(u - W\bar{r} - \bar{r}'_{net} W^T_{net} \bar{e}\right)_o$ shrinking like $c_2/\beta$. Likewise, also for non-output neurons $i$, the self-consistency solution $u_i = F(u)_i$ deviates arbitrarily little from the solution of the clamped state. To ensure the smooth drift of the fixed point while $1/\beta$ deviates from 0 we require that the Jacobian of $F$ at the fixed point is invertible.

Because the output $\bar{e}^*_o$ shrinks with $1/\beta$, the cost shrinks quadratically with increasing nudging strength, $C = \frac{1}{2}\|\bar{e}^*\|^2 = o\left(\frac{1}{\beta^2}\right)$, and hence the cost term $\frac{\beta}{2}\|\bar{e}^*\|^2$ that enters in $L = E^M + \frac{\beta}{2}\|\bar{e}^*\|^2$

vanishes in the limit $\beta \to \infty$. In this large $\beta$ limit, where $\bar{e}_o^* = 0$ and hence the outputs are clamped, $u_o = u_o^*$, the Lagrangian reduces to the mismatch energy, $L = E^{\mathrm{M}}$. Along the least-action trajectories, we, therefore, get $\dot{W} \propto -\frac{\partial L}{\partial W} = -\frac{dL}{dW} = -\frac{dE^{\mathrm{M}}}{dW}$. The first equality uses *Equation 28*, and the second uses $L = E^{\mathrm{M}}$ just derived for $\beta = \infty$. This is a statement (*i*) of Theorem 1. In the case of successful learning, $E^{\mathrm{M}} = 0$, we also conclude that the cost vanishes, $C = 0$. This is the case because $E^{\mathrm{M}} = 0$ implies $E^{\mathrm{M}}{}_o = 0$ for all output neurons $o$. Since $E^{\mathrm{M}}{}_o = \frac{1}{2}\bar{e}_o^2 = \frac{1}{2}(\bar{r}'_{\mathrm{net}} W_{\mathrm{net}}^{\mathrm{T}} \bar{e} + \beta\,\bar{e}^*)_o^2$, we conclude that $\bar{e}_o = 0$, and if the output neurons do not feed back to the network (which we can assume without loss of generality), we conclude that $\bar{e}_o^* = 0$.

(ii) To consider the case of small $\beta$, we use that the cost $C$ can be expressed as $C = \frac{\partial L}{\partial \beta}$. This is a direct consequence of how $C$ enters in $L = \frac{1}{2}\|\bar{e}\|^2 + \frac{\beta}{2}C$, see *Equation 16* and *Scellier and Bengio, 2017*. We now put this together with *Equation 28* and the finding that $\frac{\partial L}{\partial \beta} = \frac{dL}{d\beta}$. Since for the Lipschitz continuous function $L$ in $u$, $W$, and $\beta$ ($L$ is even smooth in these arguments), the total derivatives interchange (which is a consequence of the Moore-Osgood theorem applied to the limits of the difference quotients), we then get at any $t$,

$$\frac{dC}{dW} = \frac{d}{dW}\frac{\partial L}{\partial \beta} = \frac{d}{dW}\frac{dL}{d\beta} = \frac{d}{d\beta}\frac{dL}{dW} = \frac{d}{d\beta}\frac{\partial L}{\partial W} = -\frac{d}{d\beta}\bar{e}\,\bar{r}^{\mathrm{T}}. \tag{29}$$

The last expression is calculated from the specific form of the Lagrangian *Equation 17*, using that by definition $\bar{e} = u - W\bar{r}$.

Finally, in the absence of output nudging, $\beta = 0$, we can assume vanishing errors, $\bar{e} = 0$, as they solve the self-consistency equation, $\bar{e} = \bar{r}'_{\mathrm{net}} W_{\mathrm{net}}^{\mathrm{T}} \bar{e}$ for all $t$, see *Equation 27*. For these solutions we have $\bar{e}\,\bar{r}^{\mathrm{T}}\big|_{\beta=0} = 0$. Writing out the total derivative of the function $g(\beta) = \bar{e}\,\bar{r}^{\mathrm{T}}$ with respect to $\beta$ at $\beta = 0$ as limit of the difference quotient, $\frac{dg(\beta)}{d\beta}\big|_{\beta=0} = \lim_{\beta\to 0}\frac{1}{\beta}\big(g(\beta) - g(0)\big) = \lim_{\beta\to 0}\frac{1}{\beta}g(\beta)$, using that $g(0) = 0$, we calculate at any $t$,

$$\frac{d\bar{e}\,\bar{r}^{\mathrm{T}}}{d\beta}\bigg|_{\beta=0} = \lim_{\beta\to 0}\frac{1}{\beta}\left(\bar{e}\,\bar{r}^{\mathrm{T}} - \bar{e}\,\bar{r}^{\mathrm{T}}\big|_{\beta=0}\right) = \lim_{\beta\to 0}\frac{1}{\beta}\bar{e}\,\bar{r}^{\mathrm{T}}. \tag{30}$$

Here, we assume that $\bar{e}\,\bar{r}^{\mathrm{T}}$ is evaluated at $\beta > 0$ (that itself approaches 0), while $\bar{e}\,\bar{r}^{\mathrm{T}}\big|_{\beta=0}$ is evaluated at $\beta = 0$. Combining *Equations 29 and 30* yields the cost gradient at any $t$,

$$-\frac{dC}{dW} = \lim_{\beta\to 0}\frac{1}{\beta}\bar{e}\,\bar{r}^{\mathrm{T}}. \tag{31}$$

This justifies the gradient learning rule $\dot{W}$ in *Equation 27*. Learning is stochastic gradient descent on the expected cost, where stochasticity enters in the randomization of the stimulus and target sequences $r_{\mathrm{in}}(t)$ and $u^*(t)$. For the regularity statement, see 'From implicit to explicit differential equations' in the sec:Integration. Notice that this proof works for a very general form of the Lagrangian $L$, until the specific expression for $\frac{\partial L}{\partial W}$. For a proof in terms of partial derivatives only, see Appendix 8, and for a primer on partial and total derivatives see Appendix 7.

## Instantaneous gradient descent on $C(u_o^*, \bar{r}_{\mathrm{in}})$

The cost $C = \frac{1}{2}\|u_o^* - u_o\|^2$ at each time $t$ is a function of the voltage $u_o$ of the output neurons and the corresponding targets. In a feedforward network, due to the instantaneity of the voltage propagation *Equation 23*, $u_o$ is in the absence of output nudging ($\beta = 0$) an instantaneous function of the voltage at the first layer, $u_1(t) = W_{\mathrm{in}}\bar{r}_{\mathrm{in}}(t) + u_1(t_0)\,e^{-\frac{t-t_0}{\tau}}$. For initialisation at $t_0 = -\infty$, the second term vanishes for all $t$ and hence $u_1(t) = W_{\mathrm{in}}\bar{r}_{\mathrm{in}}(t)$. The output voltage $u_o(t)$, therefore, becomes a function $F_W$ of the low-pass filtered input rate $\bar{r}_{\mathrm{in}}(t)$ that captures the instantaneous network mapping, $u_o(t) = F_W(\bar{r}_{\mathrm{in}}(t))$, and with this the cost also becomes an instantaneous function of $\bar{r}_{\mathrm{in}}$ and $u_o^*$, namely $C(t) = \frac{1}{2}\|u_o^*(t) - u_o(t)\|^2 = \frac{1}{2}\|u_o^*(t) - F_W(\bar{r}_{\mathrm{in}}(t))\|^2$.

For a general network, again assuming $t_0 = -\infty$, the voltage is determined by the vanishing gradient $\frac{\partial L}{\partial u} = f(u, t) = u - W\bar{r}(u) - \bar{e}(u) = 0$ with $\bar{e} = \bar{\epsilon} - \beta\bar{e}^*$, see *Equation 21*. For the inclusive treatment of the initial transient see Appendix 3 and Appendix 4. Remember that $\bar{r} = (\bar{r}_{\mathrm{in}}, \bar{r}_{\mathrm{net}}(u))^{\mathrm{T}}$ and $\bar{e}^* = u_o^* - u_o$. For a given $\bar{r}_{\mathrm{in}}$ and $u_o^*$ at time $t$, the equation $f(u, t) = 0$ can be locally solved for $u$ if the

Hessian $\boldsymbol{H} = \frac{\partial^2 L}{\partial \boldsymbol{u}^2} = \frac{\partial f}{\partial \boldsymbol{u}} = \mathbf{1} - \boldsymbol{W}_{\text{net}}\boldsymbol{\rho}' - \bar{\boldsymbol{e}}'$ is invertible, $\boldsymbol{u} = \boldsymbol{F}(\bar{r}_{\text{in}}, \boldsymbol{u}_o^*)$. This mapping can be restricted to the output voltages $\boldsymbol{u}_o$ on the left-hand side, while replacing $\boldsymbol{u}_o^* = \boldsymbol{u}_o + \bar{\boldsymbol{e}}_o^*$ in the argument on the right-hand side (even if this again introduces $\boldsymbol{u}_o$ there). With this, we obtain the instantaneous mapping $\boldsymbol{u}_o(t) = \boldsymbol{F}_W(\bar{r}_{\text{in}}(t), \bar{\boldsymbol{e}}_o^*(t))$ from the low-pass filtered input and the output error to the output itself. Notice that for functional feedforward network, the network weight matrix $\boldsymbol{W}_{\text{net}}$ is lower triangular, and for small enough $\beta$ the Hessian $\boldsymbol{H}$ is, therefore, always positive definite (see also Methods, Sect. From implicit to explicit differential equations).

## Proving theorem 2 (rt-DeEL)

Here, we restrict ourselves to layered network architectures. To prove Theorem 2 first assume that interneurons receive no nudging ($\beta^{\text{I}} = 0$) and only the lateral interneuron-to-pyramidal weights $\boldsymbol{W}^{\text{PI}}{}_l$ are plastic. This is already sufficient to prove the rt-DeEL theorem. Yet, simulations showed that learning the lateral pyramidal-to-interneuron weights $\boldsymbol{W}^{\text{IP}}{}_l$ via top-down nudging, so that the interneuron activity mimics the upper layer pyramidal neuron activity, helps in learning a correct error representation. We consider this case of learning $\boldsymbol{W}^{\text{IP}}{}_l$ later.

If the microcircuits is ought to correctly implement error backpropagation, all local prediction errors $\bar{e}_l$ must vanish in the absence of output nudging ($\beta = 0$) as there is no target error. Consequently, any remaining errors in the network are caused by a misalignment of the lateral microcircuit. We show how learning the interneuron-to-pyramidal weights $\boldsymbol{W}^{\text{PI}}{}_l$ corrects for such misalignments.

To define the gradient descent plasticity of the weights $\boldsymbol{W}^{\text{PI}}{}_l$ from the interneurons to the pyramidal neurons, we consider the apical error formed by the difference of top-down input and interneuron input, $\bar{\boldsymbol{e}}_l^A = \boldsymbol{B}_l \boldsymbol{u}_{l+1} - \boldsymbol{W}^{\text{PI}}{}_l \boldsymbol{u}^{\text{I}}{}_l$, and define the apical mismatch energy as $E_l^{\text{PI}} = \frac{1}{2}\|\bar{\boldsymbol{e}}_l^A\|^2$. Gradient descent along this energy with respect to $\boldsymbol{W}^{\text{PI}}{}_l$ yields

$$\dot{\boldsymbol{W}}_l^{\text{PI}} = \eta^{\text{PI}} e_l^A \boldsymbol{u}_l^{\text{I}\,\text{T}} = \eta^{\text{PI}} \left( \boldsymbol{B}_l \boldsymbol{u}_{l+1} - \boldsymbol{W}_l^{\text{PI}} \boldsymbol{u}_l^{\text{I}} \right) \boldsymbol{u}_l^{\text{I}\,\text{T}} \tag{32}$$

evaluated online while presenting input patterns from the data distribution to the network. We assume that the apical contribution to the somatic voltage is further modulated by the somatic spike rate, $\bar{r}_l' \bar{\boldsymbol{e}}_l^A$. After successful learning, the top-down input $\boldsymbol{B}_l \boldsymbol{u}_{l+1}$ is fully subtracted away by the lateral input in the apical compartment, and we have

$$\boldsymbol{B}_l \boldsymbol{u}_{l+1} = \boldsymbol{W}^{\text{PI}}{}_l \boldsymbol{u}^{\text{I}}{}_l. \tag{33}$$

Once this condition is reached, the network achieves a state where, over the activity space spanned by the data, top-down prediction errors throughout the network vanish,

$$\bar{\boldsymbol{e}}_l = \bar{r}_l' \bar{\boldsymbol{e}}_l^A = \bar{r}_l' \left( \boldsymbol{B}_l \boldsymbol{u}_{l+1} - \boldsymbol{W}^{\text{PI}}{}_l \boldsymbol{u}_l^{\text{I}} \right) = 0. \tag{34}$$

We show that this top-down prediction error, after the successful learning of the microcircuit, shares the properties of error-backpropagation for a suitable backprojection weights $\boldsymbol{B}$.

Due to the vanishing prediction errors, pyramidal cells only receive bottom-up input $\boldsymbol{u}_{l+1} = \boldsymbol{W}_{l+1} \bar{r}_l$. Using this expression as well as the expression for interneuron membrane potentials without top-down nudging ($\beta^{\text{I}} = 0$ in *Equation 9*), $\boldsymbol{u}^{\text{I}}{}_l = \boldsymbol{W}^{\text{IP}}{}_l \bar{r}_l$, and plugging both into *Equation 33*, we get

$$\boldsymbol{B}_l \boldsymbol{W}_{l+1} \bar{r}_l = \boldsymbol{W}^{\text{PI}}{}_l \boldsymbol{W}^{\text{IP}}{}_l \bar{r}_l. \tag{35}$$

Assuming that $\boldsymbol{W}^{\text{IP}}{}_l$ has full rank, and the low-pass filtered rates $\bar{r}_l$ span the full $n_l$ dimensions of layer $l$ when sampled across the data set, we conclude that

$$\boldsymbol{B}_l \boldsymbol{W}_{l+1} = \boldsymbol{W}^{\text{PI}}{}_l \boldsymbol{W}^{\text{IP}}{}_l. \tag{36}$$

In other words, the loop via upper layer and back is learned to be matched by a lateral loop through the interneurons.

*Equation 36* imposes a restriction on the minimal number of interneurons $n_l^{\text{I}}$ at layer $l$. In fact, the matrix product $\boldsymbol{B}_l \boldsymbol{W}_{l+1}$ maps a $n_l$-dimensional space onto itself via $n_{l+1}$-dimensional space. The maximal rank of the this matrix product is limited by the smallest dimension, i.e., $\text{rank}(\boldsymbol{B}_l \boldsymbol{W}_{l+1}) \leq \min(n_l, n_{l+1})$.

Analogously, $\mathrm{rank}(\boldsymbol{W}^{\mathrm{PI}}{}_l \boldsymbol{W}^{\mathrm{IP}}{}_l) \leq \min(n_l, n_l^{\mathrm{I}})$. But since the two ranks are the same according to *Equation 36*, we conclude that in general $n_l^{\mathrm{I}} \geq \min(n_l, n_{l+1})$ must hold, i.e., there should be at least as many interneurons at layer $l$ as the lowest number of pyramidal neurons at either layer $l$ or $l+1$. Note that by choosing $n_l^{\mathrm{I}} = n_{l+1}$ as in *Sacramento et al., 2018* (or $n_l^{\mathrm{I}} > n_{l+1}$ as in this work), the conditions is fulfilled.

With $\boldsymbol{u}^{\mathrm{I}}{}_l = \boldsymbol{W}^{\mathrm{IP}}{}_l \bar{\boldsymbol{r}}_l$ and *Equation 36*, the top-down prediction error from *Equation 34*, in the presence of output nudging ($\beta > 0$), can be written in the backpropagation form

$$\bar{\boldsymbol{e}}_l = \bar{\boldsymbol{r}}_l' \cdot (\boldsymbol{B}_l \boldsymbol{u}_{l+1} - \boldsymbol{W}^{\mathrm{PI}}{}_l \boldsymbol{u}_l^{\mathrm{I}}) = \bar{\boldsymbol{r}}_l' \cdot (\boldsymbol{B}_l \boldsymbol{u}_{l+1} - \boldsymbol{W}^{\mathrm{PI}}{}_l \boldsymbol{W}^{\mathrm{IP}}{}_l \bar{\boldsymbol{r}}_l) \tag{37a}$$

$$= \bar{\boldsymbol{r}}_l' \cdot (\boldsymbol{B}_l \boldsymbol{u}_{l+1} - \boldsymbol{B}_l \boldsymbol{W}_{l+1} \bar{\boldsymbol{r}}_l) = \bar{\boldsymbol{r}}_l' \cdot \boldsymbol{B}_l \left( \boldsymbol{u}_{l+1} - \boldsymbol{W}_{l+1} \bar{\boldsymbol{r}}_l \right) \tag{37b}$$

$$= \bar{\boldsymbol{r}}_l' \boldsymbol{B}_l \, \bar{\boldsymbol{e}}_{l+1} = \bar{\boldsymbol{r}}_l' \cdot \boldsymbol{B}_l \, \bar{\boldsymbol{r}}_{l+1}' \cdot \bar{\boldsymbol{e}}_{l+1}^A . \tag{37c}$$

Finally, the simulations showed that learning the lateral weights in the microcircuit greatly benefits from also adapting the pyramidal-to-interneuron weights $\boldsymbol{W}^{\mathrm{IP}}$ by gradient descent on $E^{\mathrm{IP}} = \frac{1}{2} \sum_l \| \boldsymbol{u}^{\mathrm{I}}{}_l - \boldsymbol{W}^{\mathrm{IP}}{}_l \bar{\boldsymbol{r}}_l \|^2$, using top-down nudging of the inhibitory neurons ($\beta^{\mathrm{I}} > 0$),

$$\dot{\boldsymbol{W}}^{\mathrm{IP}}_l = \eta^{\mathrm{IP}} \left( \boldsymbol{u}_l^{\mathrm{I}} - \boldsymbol{W}^{\mathrm{IP}}{}_l \bar{\boldsymbol{r}}_l \right) \bar{\boldsymbol{r}}_l^{\mathrm{T}} . \tag{38}$$

After learning we have $\boldsymbol{u}^{\mathrm{I}}{}_l = \boldsymbol{W}^{\mathrm{IP}}{}_l \bar{\boldsymbol{r}}_l$, and plugging in $\boldsymbol{u}_l^{\mathrm{I}} = (1 - \beta^{\mathrm{I}}) \boldsymbol{W}^{\mathrm{IP}}{}_l \bar{\boldsymbol{r}}_l + \beta^{\mathrm{I}} \boldsymbol{B}^{\mathrm{IP}}{}_l \boldsymbol{u}_{l+1}$ (*Equation 9*), we obtain $\boldsymbol{W}^{\mathrm{IP}}{}_l \bar{\boldsymbol{r}}_l = \boldsymbol{B}^{\mathrm{IP}}{}_l \boldsymbol{u}_{l+1}$. Since $\boldsymbol{u}_{l+1} = \boldsymbol{W}_{l+1} \bar{\boldsymbol{r}}_l$, we conclude as before,

$$\boldsymbol{W}^{\mathrm{IP}}{}_l = \boldsymbol{B}^{\mathrm{IP}}{}_l \boldsymbol{W}_{l+1} . \tag{39}$$

The top-down weights $\boldsymbol{B}^{\mathrm{IP}}{}_l$ that nudge the lower-layer interneurons has randomized entries and may be considered as full rank. If there are less pyramidal neurons in the upper layer than interneurons in the lower layer, $\boldsymbol{B}^{\mathrm{IP}}{}_l$ selects a subspace in the interneuron space of dimension $n_{l+1} < n_l^{\mathrm{I}}$. This seems to simplify the learning of the interneuron-to-pyramidal cell connections $\boldsymbol{W}^{\mathrm{PI}}$. In fact, this learning now has only to match the $n_{l+1}$-dimensional interneuron subspace embedded in $n_l^{\mathrm{I}}$ dimensions to an equal ($n_{l+1}$-)dimensional pyramidal cell subspace emedded in $n_l$ dimensions.

Learning of the interneuron-to-pyramidal cell connections works with the interneuron nudging as before, and combining *Equations 36 with 39* yields the 'loop consistency'

$$\boldsymbol{B}_l \boldsymbol{W}_{l+1} = \boldsymbol{W}^{\mathrm{PI}}{}_l \boldsymbol{B}^{\mathrm{IP}}{}_l \boldsymbol{W}_{l+1} . \tag{40}$$

The learning of the microcircuit was described in the absence of output nudging. Conceptually, this is not a problem as one could introduce a pre-learning phase where the lateral connections are first correctly aligned before learning of the feedforward weights begins. In simulations we find that both the lateral connections as well as the forward connections can be trained simultaneously, without the need for such a pre-learning phase. We conjecture that this is due to the fact that our plasticity rules are gradient descent on the energy functions $L$, $E^{\mathrm{PI}}$, and $E^{\mathrm{IP}}$, respectively.

## From implicit to explicit differential equations

The voltage dynamics is solved by a forward-Euler scheme $\boldsymbol{u}(t + dt) = \boldsymbol{u}(t) + \dot{\boldsymbol{u}}(t)\, dt$. The derivative $\dot{\boldsymbol{u}}(t)$ is calculated either through (*i*) the implicit differential *Equation 7a* yielding $\tau \dot{\boldsymbol{u}}(t) = \boldsymbol{h}(\boldsymbol{u}(t), \dot{\boldsymbol{u}}(t - dt))$, or (*ii*) by isolating $\dot{\boldsymbol{u}}(t)$ and solving for the explicit differential equation $\tau \dot{\boldsymbol{u}}(t) = \boldsymbol{g}(\boldsymbol{u}(t))$, as explained in Appendix 3 (after *Equation 51*).

(i) The implicit differential equation, $\tau \dot{\boldsymbol{u}}(t) = -\boldsymbol{u}(t) + \boldsymbol{W}\boldsymbol{r}(t) + \boldsymbol{e}(t)$, see *Equation 22*, is iteratively solved by assigning $\boldsymbol{r}(t) = \rho(\boldsymbol{u}(t)) + \rho'(\boldsymbol{u}(t))\dot{\boldsymbol{u}}(t - dt)$ and calculating the error $\boldsymbol{e}(t) = \bar{\boldsymbol{e}}(t) + \tau \dot{\bar{\boldsymbol{e}}}(t)$ with $\bar{\boldsymbol{e}}(\boldsymbol{u}) = \rho'(\boldsymbol{u})\boldsymbol{W}_{\mathrm{net}}^{\mathrm{T}} \left( \boldsymbol{u} - \boldsymbol{W}_{\mathrm{net}} \rho(\boldsymbol{u}) - \boldsymbol{W}_{\mathrm{in}} \bar{\boldsymbol{r}}_{\mathrm{in}} \right) + \beta \bar{\boldsymbol{e}}^*$ and $\dot{\bar{\boldsymbol{e}}}(t) = \bar{\boldsymbol{e}}'(\boldsymbol{u}(t))\dot{\boldsymbol{u}}(t - dt)$.

This iteration exponentially converges to a fixed point $\dot{\boldsymbol{u}}(t)$ on a time scale $\frac{dt}{1-k}$, where $1 - k > 0$ is the smallest Eigenvalue of the Hessian $\boldsymbol{H} = \frac{\partial^2 L}{\partial \boldsymbol{u}^2} = \mathbf{1} - \boldsymbol{W}_{\mathrm{net}} \rho' - \bar{\boldsymbol{e}}'$, see Appendix 3.

(ii) The explicit differential equation is obtained by eliminating the $\dot{\boldsymbol{u}}$ from the right-hand side of the implicit differential equation. Since $\dot{\boldsymbol{u}}$ enters linearly we get $\tau \boldsymbol{H} \dot{\boldsymbol{u}} = -\boldsymbol{f} - \tau \frac{\partial \boldsymbol{f}}{\partial t}$ with $\boldsymbol{f}(\boldsymbol{u}, t) = \frac{\partial L}{\partial \boldsymbol{u}} = \boldsymbol{u} - \boldsymbol{W}\bar{\boldsymbol{r}} - \bar{\boldsymbol{\epsilon}} - \beta \bar{\boldsymbol{e}}^*$. The explicit form is obtained by matrix inversion, $\dot{\boldsymbol{u}} = \boldsymbol{g}(\boldsymbol{u}, t) = -\frac{1}{\tau} \boldsymbol{H}^{-1} \left( \boldsymbol{f} + \tau \frac{\partial \boldsymbol{f}}{\partial t} \right)$, as the Hessian is invertible if it is strictly positive definite (which is typically the case, see Appendix 3, after *Equation 48*). The external input and the target enter through

$\frac{\partial f}{\partial t} = W_{\text{in}} \dot{r}_{\text{in}} + \beta \dot{u}_o^*$, where the derivative of the target voltage is only added for the output neurons $o$. This explicit differential equation is shown to be contractive in the sense that for each input trajectory $r_{\text{in}}(t)$ and target trajectory $u^*(t)$, the voltage trajectory $u(t)$ is locally attracting for neighbouring trajectories. This local attracting trajectory is the vanishing-gradient trajectory $f(u, t) = 0$, and the gradient remains 0 even if the input contains delta-functions, see Appendix 4.

## Moving and latent equilibria: a formal definition

We showed that the motor output ($u_o$), together with the low-pass filtered sensory input ($\bar{r}_{\text{in}}$) and the motor feedback ($\bar{e}_o^*$) is in a moving equilibrium, $u_o = F_W(\bar{r}_{\text{in}}, \bar{e}_o^*)$, see *Figure 3a*. In general, a dynamical system in $u$ that is given in an implicit form $G(x, \dot{x}, u, \dot{u}) = 0$ with external inputs $(x, \dot{x})$ is said to be in a moving equilibrium if the variable $u$ is an instantaneous function of the input $x$ at any point in time, $u = F(x)$. The fact that the implicit differential equation $G = 0$ represents a dynamical system in $u$ implies that, in principle, it has a representation in the explicit form $\dot{u} = g(u, x, \dot{x})$, guaranteed by an invertible Jacobian $\frac{\partial G}{\partial \dot{u}}$.

Our example is obtained from $G = (1 + \tau \frac{\mathrm{d}}{\mathrm{d}t}) f$ with $f(u, x) = \frac{\partial L}{\partial u}$ and $x = (\bar{r}_{\text{in}}, \bar{e}_o^*)$, leading to $x + \tau \dot{x} = (r_{\text{in}}, e_o^*)$. Given the paramterization of $G$ by the weights, we get the parametrized function $u = F_W(x)$, and this is restricted to the output components $u_o$ of $u$. The condition on the Jacobian translates to $\frac{\partial G}{\partial \dot{u}} = \frac{\partial f}{\partial u} = \frac{\partial^2 L}{\partial u^2}$ being invertible. Crucially, the description of the dynamics in the biological or physical substrate is not given in its explicit form $\dot{u} = g(u, x, \dot{x})$. However, it is given in an implicit form expressed as $\dot{u} = h(x, \dot{x}, u, \dot{u})$, where $\dot{u}$ still appears on the right-hand side. This 'hybrid' form is directly solved either in real time by the biophysical substrate itself, or by the forward-Euler scheme on clocked hardware, see (*i*) above. Notice that moving equilibria $u = F_W(x)$ with $x = (\bar{r}_{\text{in}}, \bar{e}_o^*)$ are able to capture complex temporal processing of the instantaneous input $r_{\text{in}}$. In fact, the low-pass filtering $\bar{r}_{\text{in}}$ can be obtained on various time scales through different $\tau_{\text{in}}$'s, and $F_W$ for a general network $W$ can be arbitrary complex. The task is to adapt $W$ such that the 'hybrid' dynamical system eventually implements the target mapping $u_o^* = F^*(x)$.

The Latent Equilibrium (*Haider et al., 2021*) can be analogously formalized as a dynamical system in $u$, implicitly given by $G(x, u, \dot{u}) = 0$, and having a solution of the form $u + \tau \dot{u} = F(x)$. Abbreviating again $f(u, x) = \frac{\partial L}{\partial u}$ with the same Lagrangian $L = \frac{1}{2}\|u - W\rho(u)\|^2 + \frac{\beta}{2} C$ as in the present NLA, the Latent Equilibrium is obtained for $G(x, u, \dot{u}) = f(u + \tau \dot{u}, x)$. The solution implies that the rate $r = \rho(u + \tau \dot{u}) = \rho(F(x))$ is an instantaneous function of $x = (r_{\text{in}}, e_o^*)$, here without low-pass filtering. As for moving equilibria, the crucial point is that the biophysical substrate implements a hybrid form of the dynamical system, now $\dot{u} = h(x, u, \dot{u})$, that is implicitly solved by the analog substrate, and also allows for a solution in clocked hardware. For an extended stability analysis of the moving and latent equilibria see Appendix 4.

## Simulation details

Solving the explicit differential equation seems to be more robust when the learning rate for $\dot{W}$ gets larger. The explicit form is also less sensitive to large Euler steps $dt$, see Appendix 3. By this reason, the ordinary differential equations (ODE) were solved in the explicit form when including plasticity $\dot{W}$. The algorithms are summarized as follows, once without interneurons (Algorithm 1), and once with interneurons (Algorithm 2):

Algorithm 1. **with projection neurons only, for Figures 3 and 4 (using the explicit ODE, i.e., Step 12 instead of 11)**

1: current state: $u(t)$, $W(t)$
2: # consider full vectors and matrices (padded with 0's for feedforward networks)
3:# drop time argument ($t$) for convenience
4: $\bar{r}_{\text{net}} \leftarrow \rho(u)$, $\bar{r} \leftarrow \left(\bar{r}_{\text{in}}, \bar{r}_{\text{net}}\right)^{\text{T}}$, $W \leftarrow \left(W_{\text{in}}, W_{\text{net}}\right)$
5: calculate weight derivatives
6: $\dot{W} \leftarrow \eta(u - W\bar{r})\bar{r}^{\text{T}}$
7: calculate low-pass-filtered errors
8: $\bar{e}_o^* \leftarrow u_o^* - u_o$, $\bar{e}_i^* = 0$ for non-output neurons
9: $\bar{e} \leftarrow \bar{r}'_{\text{net}}W_{\text{net}}^{\text{T}}(u - W\bar{r}) + \beta\bar{e}^*$
10: calculate temporal voltage derivatives either implicitly (11) or explicitly (12)
11: Implicit: $\tau\dot{u} \leftarrow -u + W(\bar{r} + \tau\dot{\bar{r}}) + (\bar{e} + \tau\dot{\bar{e}})$
12: Explicit: $f \leftarrow u - W\bar{r} - \bar{e}$, $H \leftarrow \frac{\partial f}{\partial u}$, $\dot{u} \leftarrow$ solve $\tau H(u)\dot{u} = -f - \tau\frac{\partial f}{\partial t}$ via Cholesky decomposition
13: update voltage and weights
14: $u \leftarrow u + \dot{u}dt$, $W \leftarrow W + \dot{W}dt$

Algorithm 2. **including plastic interneurons, for Figure 5 (using the explicit ODE, i.e., Step 13 instead of 12)**

1: current state: $u(t)$, $W(t)$, $u^{\text{I}}(t)$, $W^{\text{PI}}(t)$, $W^{\text{IP}}(t)$
2: # consider full vectors and matrices and drop time argument as in Algorithm 1
3: $\bar{r} \leftarrow \left(\bar{r}_{\text{in}}, \rho(u)\right)^{\text{T}}$
4: calculate weight derivatives
5: $\dot{W} \leftarrow \eta(u - W\bar{r})\bar{r}^{\text{T}}$
6: $\dot{W}^{\text{PI}} \leftarrow \eta^{\text{PI}}(Bu - W^{\text{PI}}u^{\text{I}})u^{\text{IT}}$
7: $\dot{W}^{\text{IP}} \leftarrow \eta^{\text{IP}}(u^{\text{I}} - W^{\text{IP}}\bar{r})\bar{r}^{\text{T}}$
8: calculate low-pass filtered errors
9: $\bar{e}_o^* \leftarrow u_o^* - u_o$ ($\bar{e}_i^* = 0$ for non-output neurons $i$)
10: $\bar{e} \leftarrow Bu - W^{\text{PI}}u^{\text{I}} + \beta\bar{e}^*$ ($B_{o,:} = W_{o,:}^{\text{PI}} = 0$ for output neurons o)
11: calculate temporal voltage derivatives either implicitly (12) or explicitly (13)
12: Implicit: $\tau\dot{u} \leftarrow -u + W(\bar{r} + \tau\dot{\bar{r}}) + (\bar{e} + \tau\dot{\bar{e}})$
13: Explicit: $f \leftarrow u - W\bar{r} - \bar{e}$, $H \leftarrow \frac{\partial f}{\partial u}$, $\dot{u} \leftarrow$ solve $\tau H(u)\dot{u} = -f - \tau\frac{\partial f}{\partial t}$ via Cholesky decomposition
14: update network state \For $X \in \{u, W, W^{\text{PI}}, W^{\text{IP}}\}$
15: $X \leftarrow X + \dot{X}dt$ End For
16: $u^{\text{I}} \leftarrow (1 - \beta^{\text{I}})W^{\text{IP}}\bar{r} + \beta^{\text{I}}B^{\text{IP}}u$

## Details for *Figure 3b*

Color coded snapshot of cortical local field potentials (LFPs) in a human brain from 56 deep iEEG electrodes at various locations, converted with the sigmoidal voltage-to-rate function $\bar{r}(u) = \frac{1}{1+e^{-u}}$ and plotted onto a standard Talairach Brain (*Talairach and Tournoux, 1988*). The iEEG data is from a patient with pharmacoresistant epilepsy and electrodes implanted during presurgical evaluation, extracted from the data release of *Burrello et al., 2019*. The locations of the electrodes are chosen in accordance with plausibilty, as the original positions of the electrodes were omitted due to ethical standards to prevent patient identification.

## Details for *Figure 3c*

Simulations of the voltage dynamics (*Equation 7a*) and weight dynamics (*Equation 8*), with learning rate $\eta = 10^{-3}$, step size $dt = 1$ ms for the forward Euler integration, membrane time constant $\tau = 10$ ms and logistic activation function. Weights were initialized randomly from a normal distribution $\mathcal{N}(0, 0.1^2)$ with a cut-off at ± 0.3. The number of neurons in the network $\mathcal{N}$ was $n = 96$, among them 56 output neurons $\mathcal{O} \subset \mathcal{N}$ that were simultaneously nudged, and 40 hidden neurons. During training, all output neurons were nudged simultaneously (with $\beta = 0.1$), whereas during testing, only 42 out of 56 neurons were nudged, the remaining 14 left to reproduce the traces. Data points of the iEEG signal were sampled with a frequency of 512 Hz. For simplicity, we, therefore, assumed that successive data points are separated by 2ms, and up-sampled the signal via simple interpolation to 1 ms resolution as required by our integration scheme. Furthermore, the raw values were normalized by dividing them by a factor of 200 to ensure that they are approximately in a range of ±1–2. Training and testing was done on two separate 8 s traces of the iEEG recording. Same data as in *Figure 3b1*.

## Details for *Figure 4*

Simulation of the neuronal and synaptic dynamics as given by *Equation 8*, *Equation 7a*, *Equation 7b*. For 5 ms, 10 ms, and 50 ms presentation time, we used an integration step size of $dt = 0.05\,\text{ms}$, $dt = 0.1\,\text{ms}$ and $dt = 0.5\,\text{ms}$, respectively (and $dt = 1\,\text{ms}$ otherwise). As an activation function, we used the step-linear function (hard sigmoidal) with $\bar{r}(u) = 0$ for $u \leq 0$, $\bar{r}(u) = 1$ for $u \geq 1$ and $\bar{r}(u) = u$ in between. The learning rate was initially set to $\eta = 10^{-3}$ and then reduced to $\eta = 10^{-4}$ after 22,000 s. The nudging strength was $\beta = 0.1$ and the membrane time constant $\tau = 10\,\text{ms}$. In these simulations (and only for these) we assumed that at each presynaptic layer $l = 0, 1, .., n - 1$ there is a first neuron indexed by 0 that fires with constant rate $\bar{r}_{l,0} = 1$, effectively allowing the postsynaptic neurons $\bar{r}_{l+1}$ to learn a bias through the first column of the weight matrix $W_{l+1}$. Weights were initialized randomly from a normal distribution $\mathcal{N}(0, 0.01^2)$ with a cut-off at ±0.03. For an algorithmic conversion see the scheme below. In *Figure 4c1*, 'rt-DeEP w/o lookahead' is based on the dynamics $\tau \dot{u} = -u + W\bar{r} + \bar{e}$. For '$\dot{u}$ w/o error + backprop,' we use $\tau \dot{u} = -u + Wr$ as the forward model (so without error terms on the membrane potential, but a prospective $r$), and calculate weight updates using error backpropagation. In 4c2, we provide three controls: the test error for (i) a standard shallow artificial neural network trained on MNIST (black dashed line), (ii) rt-DeEP without prospective coding (as in *Figure 4c1*), but in *Figure 4c2* with plasticity only turned on when the network is completely stationary, i.e., after waiting for several 100ms, such that synaptic weights are not changed during transients (orange dashed line, denoted by 'w/o transients'), and (iii) an equivalent artificial neural network, $u_l = W_l \bar{r}_{l-1}$, trained using error backpropagation (black dashed line, 'standard backprop').

## Details for *Figure 5*

Simulation of neuronal and synaptic dynamics with plastic microcircuit, i.e., the pyramidal-to-interneuron and lateral weights of the microcircuit learned during training.

For the results shown in *Figure 5c2*, the following parameters were used. As an activation function, we used a hard sigmoid function and the membrane time constant was set to $\tau = 10\,\text{ms}$. Image presentation time is 100ms. Forward, pyramidal-to-interneuron and interneuron-to-pyramidal weights were initialized randomly from a normal distribution $\mathcal{N}(0, 0.01^2)$ with a cut-off at ±0.03. All learning rates were chosen equal $\eta = 10^{-3}$ and were subsequently reduced to $\eta = 10^{-4}$ after 22,000 s training time. The nudging parameters were set to $\beta = 0.1$ and $\beta^{\text{I}} = \frac{0.1}{1.1}$. The feedback connections $B_l$ and the nudging matrices $B_l^{\text{IP}}$ were initialized randomly from a normal distribution $5\mathcal{N}(0, 0.01^2)$ with a cut-off at ±0.15. The used integration step size was $dt = 0.25$ ms. All weights were trained simultaneously. For an algorithmic conversion see the scheme below. The interneuron membrane potential was calculated by *Equation 9* with a linear transfer function.

## Acknowledgements

We would like to thank Federico Benitez, Jonathan Binas, Paul Haider, Kevin Max, Alexander Mathis, Alexander Meulemans, and Jean-Pascal Pfister for helpful discussions, Kaspar Schindler for providing the human intracortical EEG data, and the late Karlheinz Meier for his dedication and support throughout the early stages of the project. WS thanks Nicolas Zucchet for mathematical discussions and hints to the literature on time-varying optimal control. DD acknowledges support through the European Space Agency's Postdoctoral Research Fellowship Programme. We express our particular gratitude towards the Manfred Stärk Foundation for their continued support.

## Additional information

### Funding

| Funder | Grant reference number | Author |
|---|---|---|
| European Union 7th Framework Programme | 10.3030/720270 | Walter Senn Mihai A Petrovici |
| Swiss National Science Foundation | CRSII5180316 | Walter Senn |

| Funder | Grant reference number | Author |
|---|---|---|
| Swiss National Science Foundation | PZ00P3_186027 | João Sacramento |
| European Union 7th Framework Programme | 10.3030/785907 | Walter Senn Mihai A Petrovici |
| European Union 7th Framework Programme | 10.3030/945539 | Walter Senn Mihai A Petrovici |
| European Union 7th Framework Programme | 604102 | Walter Senn |
| Horizon Europe | 10.3030/101147319 | Walter Senn Mihai A Petrovici Jakob Jordan |

The funders had no role in study design, data collection and interpretation, or the decision to submit the work for publication.

## Author contributions

Walter Senn, Conceptualization, Resources, Formal analysis, Supervision, Funding acquisition, Validation, Investigation, Visualization, Methodology, Writing – original draft, Project administration, Writing – review and editing; Dominik Dold, Data curation, Software, Formal analysis, Validation, Investigation, Visualization, Methodology, Writing – original draft; Akos F Kungl, Data curation, Software, Formal analysis, Investigation, Methodology; Benjamin Ellenberger, Data curation, Software, Investigation, Visualization; Jakob Jordan, Software, Supervision, Investigation, Methodology, Writing – original draft; Yoshua Bengio, Conceptualization, Methodology, Writing – original draft; João Sacramento, Conceptualization, Formal analysis, Investigation, Methodology, Writing – original draft; Mihai A Petrovici, Conceptualization, Data curation, Software, Formal analysis, Supervision, Funding acquisition, Validation, Investigation, Visualization, Methodology, Writing – original draft, Project administration

## Author ORCIDs

Walter Senn (iD) https://orcid.org/0000-0003-3622-0497
Dominik Dold (iD) https://orcid.org/0000-0001-7626-9960
Benjamin Ellenberger (iD) https://orcid.org/0000-0002-4787-0471
Mihai A Petrovici (iD) https://orcid.org/0000-0003-2632-0427

Reviewer #1 (Public Review): https://doi.org/10.7554/eLife.89674.3.sa1
Reviewer #2 (Public Review): https://doi.org/10.7554/eLife.89674.3.sa2
Author response https://doi.org/10.7554/eLife.89674.3.sa3

# Additional files

## Supplementary files

MDAR checklist

## Data availability

The current manuscript is a computational study, and the modelling code is available at GitHub, copy archived at *Ellenberger, 2024*.

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

# Appendix 1

## Extracting the presynaptic voltage error

## Threshold-linear transfer functions

There is an important special case where the presynaptic voltage error can directly be extracted from presynaptic firing rates, without need to invert the transfer function via synaptic depression as shown below. This is the case when voltage errors in the upper layers are small, and the voltage-to-rate transfer function has derivatives $\rho' = 0$ or 1, so that $\bar{r}'_{l+1} = (\bar{r}'_{l+1})^2$. The condition is satisfied, for instance, for a doubly threshold-linear function (a 'doubly rectified linear unit', ReLu) defined by $\rho(\breve{u}) = 0$ for $\breve{u} < 0$ and $\rho(\breve{u}) = \breve{u}$ for $0 \leq \breve{u} \leq r_{\max}$, while $\rho(\breve{u}) = r_{\max}$ for larger voltages. In this case we calculate

$$\bar{e}_{l+1} = \bar{r}'_{l+1}\bar{e}^A_{l+1} = (\bar{r}'_{l+1})^2\bar{e}^A_{l+1} = \bar{r}'_{l+1}\bar{e}_{l+1} = \bar{r}'_{l+1}\cdot\left(\boldsymbol{u}_{l+1} - \boldsymbol{W}_{l+1}\bar{r}_l\right) \tag{41a}$$

$$\approx \rho(\boldsymbol{u}_{l+1}) - \rho(\boldsymbol{W}_{l+1}\bar{r}_l) = \bar{r}_{l+1} - \rho(\boldsymbol{W}_{l+1}\bar{r}_l). \tag{41b}$$

The approximation uses the Taylor expansion in $\boldsymbol{u}_{l+1}$ and assumes that $\bar{e}_{l+1}$ is small. The crucial point of *Equation 41a* is that the mismatch error defined on the voltage, $\bar{e}_{l+1} = \boldsymbol{u}_{l+1} - \boldsymbol{W}_{l+1}\bar{r}_l$, can be factorized into a product of the postsynaptic rate derivative, $\bar{r}'_{l+1}$, and the apical error, and hence it can be expressed as an error defined on the rate. Restricted to the segment $0 \leq \bar{u}_{l+1} \leq r_{\max}$ where the transfer function is linear and errors do not vanish, the same microcircuit delivers the feedback $\boldsymbol{B}_l\bar{r}_{l+1}$ to the apical tree through the top-down projections, and $-\boldsymbol{B}_l\rho(\boldsymbol{W}_{l+1}\bar{r}_l)$ through the lateral connections from the interneurons. While the plasticity rules for $\boldsymbol{W}^{\mathrm{PI}}_l$ and $\boldsymbol{W}^{\mathrm{IP}}_l$ stay the same, the top-down nudging of the interneurons, see *Equation 9*, can then be formulated based on the rate instead of the upper layer voltage, $\boldsymbol{u}^{\mathrm{I}}_l = (1 - \beta^{\mathrm{I}})\boldsymbol{W}^{\mathrm{IP}}_l\bar{r}_l + \beta^{\mathrm{I}}\boldsymbol{B}^{\mathrm{IP}}_l\bar{r}_{l+1}$, with transfer function of the interneuron again the (doubly) threshold-linear $\rho(\boldsymbol{u}^{\mathrm{I}}_l)$. Since voltages and rates are identical in the segment $0 \leq \bar{u}_{l+1} \leq r_{\max}$ for each component, *Equations 36 with 39* can still be inferred.

## Rate-to-voltage inversion by short-term synaptic depression

We wish to readout the voltage error also for other nonlinear transfer functions than clipped ReLu's. To do so, we take inspiration from the classical short-term synaptic depression model (*Tsodyks and Markram, 1997*; *Abbott et al., 1997*; *Varela et al., 1997*), see also *Appendix 1—figure 1a1* (that deviates from the release-independent synaptic depression, see *Campagnola et al., 2022*). We consider a dynamic vesicle release probability that is proportional to the pool size of available vesicles, $v(\bar{r})$, and this pool size is postulated to depend on past presynaptic rates,

$$p(\mathrm{release}\,|\,\bar{r}) \;\propto\; v(\bar{r}) = 1 + \frac{a}{1 + d\,\bar{r}}, \tag{42}$$

where $\bar{r}$ is the low-pass filtered presynaptic rate, $a$ and $d$ are constants. The proportionality factor is $\frac{1}{1+a}$, making a probability out of the vesicle pools size. The effective synaptic strength $B$ of a 'backprojecting top-down' connection is the product of the absolute synaptic strength $B_\circ$ and the vesicle pool size $v$, i.e., $B = B_\circ v(\bar{r})$. The contribution to the postsynaptic current of the synapse is $Wr$, and the contribution to the postsynaptic voltage is $W\bar{r}$.

We search for an activation function $\bar{r}(u)$ such that the postsynaptic voltage contribution is the scaled presynaptic potential, $B\bar{r}(u) = B_\circ u$. Plugging in the above expression for $B$ yields $B\bar{r}(u) = B_\circ v(\bar{r})\bar{r}(u) = B_\circ u$, and dividing $B_\circ$ out, $v(\bar{r})\bar{r}(u) = u$, see *Appendix 1—figure 1a2*. With the expression for $v(\bar{r})$ in *Equation 42* we obtain a quadratic equation in $\bar{r}$ that is solved by the non-negative and monotonically increasing function

$$\bar{r} = \frac{1}{2}\left(u - \theta + \sqrt{(u-\theta)^2 + 4u/d}\right) \geq 0\,, \text{ for } u \geq 0, \tag{43}$$

with 'smooth' threshold $\theta = (1 + a)/d$ and asymptote $\bar{r} \approx u - \theta$. This gives us a transfer function $\bar{r}(u)$ that qualitatively matches those observed for pyramidal neurons recorded in the steady state (*Anderson et al., 2000*; *Rauch et al., 2003*), see *Appendix 1—figure 1a3*.

The approach generalizes to other pairs of strictly monotonic neuronal activation and depression functions $\{\bar{r}(u), v(\bar{r})\}$, as long as $u$ is not driven below some minimal value, here $u_{\mathrm{rest}} = 0$, corresponding

to $\bar{r} = 0$. The last requirement can, for instance, be accomplished by offsetting the activation function into a regime that guarantees that $u$ stays positive.

In our simulations for *Figure 5*, we did not explicitly implement a dynamic vesicle pool, i.e., the right-hand side of $v(\bar{r})\,\bar{r} = u$, but instead directly used the recovered membrane potentials $u$.

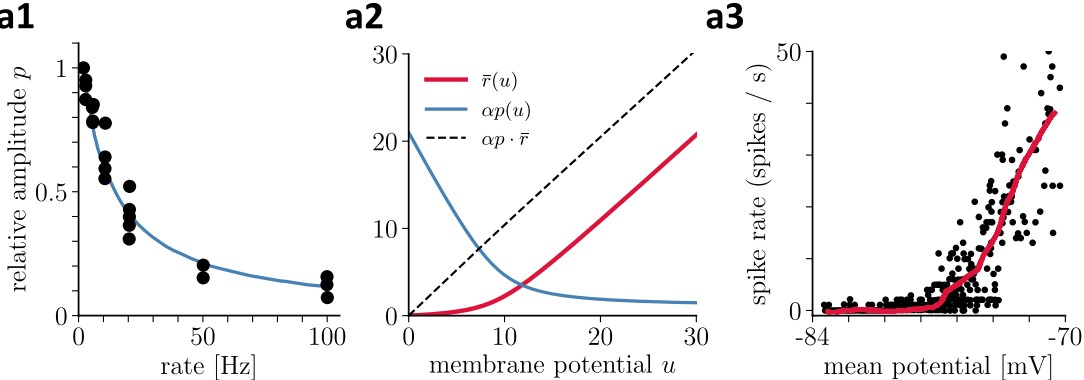

**Appendix 1—figure 1.** Recovering presynaptic potentials through short term depression. (**a1**) Relative voltage response of a depressing cortical synapse recreated from *Abbott et al., 1997*, identified as synaptic release probability $p$. (**a2**) The product of the low-pass filtered presynaptic firing rate $\bar{r}(u)$ times the synaptic release probability is $p(\bar{r})$ proportional to the presynaptic membrane potential, $p(\bar{r})\bar{r} \propto u$. (**a3**) Average in vivo firing rate of a neuron in the visual cortex as a function of the somatic membrane potential recreated from *Anderson et al., 2000*, which can be qualitatively identified as the stationary rate $\bar{r}(u)$ derived in *Equation 43*.

## Appendix 2

### Looking back and forward in time with derivatives

Since dealing with extrapolations into the future is a crucial notion of the paper we present here some of the calculations. The discounted future voltage was introduced in *Equation 4* as

$$\tilde{u}(t) = \frac{1}{\tau} \int_t^\infty u(t') e^{-\frac{t'-t}{\tau}} \, \mathrm{d}t' \ .$$

To show that $\tilde{u}$ satisfies $u = \tilde{u} - \tau \dot{\tilde{u}}$, we need to apply the Leibniz integral rule in calculating the derivative $\dot{\tilde{u}}$. This leads to

$$\frac{\mathrm{d}\tilde{u}}{\mathrm{d}t}(t) = -\frac{1}{\tau} u(t) + \frac{1}{\tau} \int_t^\infty u(t') \frac{1}{\tau} e^{-\frac{t'-t}{\tau}} \, \mathrm{d}t' \ .$$

Multiplying this equation by $\tau$ and using the definition of $\tilde{u}$ yields $\tau \dot{\tilde{u}}(t) = -u(t) + \tilde{u}(t)$, or $u = \tilde{u} - \tau \dot{\tilde{u}}$. By applying the Leibniz integral rule one also shows that $\bar{x}$, defined in *Equation 15*,

$$\bar{x}(t) = \frac{1}{\tau} \int_{-\infty}^t x(t') e^{-\frac{t-t'}{\tau}} \, \mathrm{d}t' \ ,$$

solves $\tau \dot{\bar{x}}(t) = -\bar{x}(t) + x(t)$. This differential equation can be written as $\left(1 + \tau \frac{\mathrm{d}}{\mathrm{d}t}\right)\bar{x} = x$, with lookahead operator $\left(1 + \tau \frac{\mathrm{d}}{\mathrm{d}t}\right)$ defined in *Equation 14*. To show that $\left(1 + \tau \frac{\mathrm{d}}{\mathrm{d}t}\right)x = \bar{x} + \tau \dot{\bar{x}} = x$, one applies partial integration to $\dot{\bar{x}}(t)$. Note that the equality $\bar{x} + \tau \dot{\bar{x}} = \bar{x} + \tau \dot{\bar{x}} = x$ only holds if we integrate from $-\infty$, and hence if the initialization of the trajectory is far back in the past as compared to the time constant $\tau$.

### Uniqueness of the rate function

In the main text we concluded from the postulate $r = \rho(u) + \tau \dot{\rho}(u)$ and the general relation $r = \bar{r} + \tau \dot{\bar{r}}$ that $\bar{r}(t) = \rho(u(t))$. This conclusion is a consequence of the uniqueness of a solution of an ordinary differential equation for a given initial condition that may include delta-functions on the right-hand side, see e.g., *Nedeljkov and Oberguggenberger, 2012*. In fact, we may consider both variables $\rho$ and $\bar{r}$ as solutions of the differential equation $\tau \dot{x} = -x + r$, with $x$ either $\rho$ or $\bar{r}$. Because the solution is unique, we conclude that $\rho = \bar{r}$.

### Learning the input time constants

In our applications we assumed that the input rates in the original mapping $\boldsymbol{u}_o^*(t) = \boldsymbol{F}^*(\bar{\boldsymbol{r}}_{\mathrm{in}}(t))$ are low-pass filtered by a common time constant $\tau_{\mathrm{in}(t),i}^* = \tau$ that is also shared as membrane time constant of the neurons. But in general, we may consider a subclass of network neurons (with voltage vector $\boldsymbol{u}_1$) that receive exclusively input rates $\boldsymbol{r}_{\mathrm{in}}$ and integrate these rates with corresponding time constants $\boldsymbol{\tau}_{\mathrm{in}}$. The prospective rates of these neurons, $\boldsymbol{r}_1 = \rho(\boldsymbol{u}_1) + \tau \dot{\rho}(\boldsymbol{u}_1)$, are then produced by the same voltage time constant $\tau$ of the down-stream neurons. The downstream voltage integration then cancels the presynaptic look-ahead in the presynaptic firing rate, leading again to an instantaneous voltage propagation across the entire network.

The general setting of learning to map time series is that input time series, $\boldsymbol{r}_{\mathrm{in}}(t)$ are low-pass filtered with given time constants $\boldsymbol{\tau}_{\mathrm{in}}^*$, and the target output time series $\boldsymbol{u}_o^*(t)$ are a function of these low-pass filtered inputs, $\boldsymbol{u}_o^*(t) = \boldsymbol{F}_o^*(\bar{\boldsymbol{r}}_{\mathrm{in}}^{\boldsymbol{\tau}_{\mathrm{in}}^*}(t))$. In the student network, the input time constants appear as parameters that can be learned by gradient descent to match the output of the teacher network, just as the synaptic weights are learned.

We ask for a learning rule so that the student network can extract the input time constants $\boldsymbol{\tau}_{\mathrm{in}}^*$ used by the teacher network. For this, we assume that only the neurons $\boldsymbol{u}_1$ obtain the external input and hence carry the time constant $\boldsymbol{\tau}_{\mathrm{in}}$. Because $\bar{\boldsymbol{r}}_{\mathrm{in}}^{\boldsymbol{\tau}_{\mathrm{in}}} = \boldsymbol{r}_{\mathrm{in}} - \boldsymbol{\tau}_{\mathrm{in}} \dot{\bar{\boldsymbol{r}}}_{\mathrm{in}}^{\boldsymbol{\tau}_{\mathrm{in}}}$, the gradient rule for the input time constant is

$$\dot{\boldsymbol{\tau}}_{\mathrm{in}} = -\frac{\partial L}{\partial \boldsymbol{\tau}_{\mathrm{in}}} = -\frac{\partial L}{\partial \bar{\boldsymbol{r}}_{\mathrm{in}}^{\boldsymbol{\tau}_{\mathrm{in}}}}^{\mathrm{T}} \frac{\partial \bar{\boldsymbol{r}}_{\mathrm{in}}^{\boldsymbol{\tau}_{\mathrm{in}}}}{\partial \boldsymbol{\tau}_{\mathrm{in}}} = -\boldsymbol{W}_{\mathrm{in}}^{\mathrm{T}} \left( \boldsymbol{u}_1 - \boldsymbol{W}_{\mathrm{in}} \bar{\boldsymbol{r}}_{\mathrm{in}}^{\boldsymbol{\tau}_{\mathrm{in}}} \right)^{\mathrm{T}} \dot{\bar{\boldsymbol{r}}}_{\mathrm{in}}^{\boldsymbol{\tau}_{\mathrm{in}}} \ .$$

(44)

This local learning rule also globally reduces the Lagrangian, and in the limits of $\beta \to \infty$ it is gradient descent on the mismatch energy, while in the limit $\beta \to 0$ it is gradient descent on the cost. The proof works as in Theorem 1. To learn a more complex mapping of time series that includes more complex temporal processing beyond a function of merely $\boldsymbol{u}_o^*(t) = \boldsymbol{F}_o^*(\bar{\boldsymbol{r}}_{\text{in}}^{\tau_{\text{in}}^*}(t))$, additional variables need to be introduced that form memories (see Appendix 6).

## Appendix 3

### From the implicit to the explicit differential equation

Global convergence to a moving equilibrium (i.e. to $\frac{\partial L}{\partial u} = 0$)

The original Euler-Lagrange equation is $\left(1 + \tau \frac{\mathrm{d}}{\mathrm{d}t}\right)\frac{\partial L}{\partial u} = 0$, 20, with Jacobian $\frac{\partial L}{\partial u} = f(u, t) = u - W\bar{r} - \bar{\epsilon} - \beta\bar{e}^*$. Remember the definition of $\bar{\epsilon} = \bar{r}'_{\mathrm{net}} W^{\mathrm{T}}_{\mathrm{net}}\bar{e}$ given in 21. While $\bar{e} = u - W\bar{r}$ is an error that may not vanish, the 'corrected error' $f = \frac{\partial L}{\partial u}$ is shown to vanish exponentially quick, leading to the moving equilibrium with $u = W\bar{r} - \bar{\epsilon} - \beta\bar{e}^*$. In fact, $\left(1 + \tau \frac{\mathrm{d}}{\mathrm{d}t}\right)\frac{\partial L}{\partial u} = f + \tau\dot{f} = 0$, has the general solution

$$f(u(t), t) = f(u_0, t_0)\, e^{-\frac{t - t_0}{\tau}} \longrightarrow 0 \tag{45}$$

as an exponentially decaying function in $t$ with $u(t_0) = u_0$. Note that the argument $u$ of the function $f(u, t) = u - W\bar{r} - \bar{\epsilon} - \beta\bar{e}^*$ does apriori not depend on time. It only becomes a function of time, $u(t)$, by requiring that $f + \tau\dot{f} = 0$. The voltage $u(t)$ becomes a non-trivial function of time because the time-dependence of the function $f(u, t)$, that apriori enters only through the second argument, expresses the temporally varying input $\bar{r}_{\mathrm{in}}(t)$, including a possible target output $u_o^*(t)$.

Implicit differential equation

To make the dependence of $\dot{u}$ on the right-hand-side of the implicit differential *Equation 22* more transparent, we rewrite this in the form

$$\dot{u} = -\frac{f}{\tau} - \frac{\partial f}{\partial t} + \frac{\partial}{\partial u}(W\bar{r} + \bar{\epsilon} + \beta\bar{e}^*)\,\dot{u} = -\frac{f}{\tau} - \frac{\partial f}{\partial t} + M\dot{u} = K(r_{\mathrm{in}}, u_o^*, u, \dot{u})\,. \tag{46}$$

The partial derivative of $f(u, t)$ with respect to time, $\frac{\partial f}{\partial t}$, captures the external drive from the inputs and output targets that do not depend on $u$, but may directly depend on time. In fact, instead of the argument $t$ of $f$, we could consider the two arguments $\bar{r}_{\mathrm{in}}$ and $u_o^*$, so that the partial derivative of $f(u, t) = f(u, \bar{r}_{\mathrm{in}}, u_o^*)$ with respect to $t$ can be written as

$$\frac{\partial f_i}{\partial t} = \frac{\partial f_i}{\partial \bar{r}_{\mathrm{in}}}\dot{\bar{r}}_{\mathrm{in}} + \frac{\partial f_i}{\partial u_o^*}\dot{u}_o^* = W_{i,\mathrm{in}}\dot{\bar{r}}_{\mathrm{in}} + \beta\,\delta_{io}\,\dot{u}_o^*, \tag{47}$$

where $\delta_{io}$ is the Kronecker delta that is equal to 1 if $ii$ is an output neuron, and 0 else. The partial derivatives of $f$ with respect to $u$ represents the (symmetric) Hessian of the Lagrangian, $H_{ij} = \frac{\partial f_i}{\partial u_j} = \frac{\partial^2 L}{\partial u_i \partial u_j} = \delta_{ij} - \frac{\partial}{\partial u_j}(W\bar{r} + \bar{\epsilon} + \beta\bar{e}^*)_i$, with $\delta_{ij}$ being the Kronecker-delta, $\bar{\epsilon}$ defined in 21 and $\bar{e}^*$ defined above 16. Remember that $W = (W_{\mathrm{in}}, W_{\mathrm{net}})$ and $\bar{r} = (\bar{r}_{\mathrm{in}}, \bar{r}_{\mathrm{net}}) = (\bar{r}_{\mathrm{in}}, \rho(u))$. In vectorial notation the Hessian of the Lagrangian is

$$H(u) = \frac{\partial f}{\partial u} = \frac{\partial^2 L}{\partial u\,\partial u} = 1 - W_{\mathrm{net}}\,\mathrm{diag}\left(\rho'(u)\right) - \frac{\partial}{\partial u}(\bar{\epsilon} + \beta\bar{e}^*)(u), \tag{48}$$

where 1 is the identity matrix. This Hessian is 'typically' positive definite, as we argue next. For a functionally feedforward network, the weight matrix $W_{\mathrm{net}}$ is lower triangular, and for small enough $\beta$ (for which $\bar{\epsilon}$ is small) the diagonal elements are all positive and hence the Hessian $H$ is invertible. Since $H$ is also symmetric, it is, therefore, positive definite. In the general case, we require that $W_{\mathrm{net}}$ has Eigenvalues smaller than 1, since otherwise the recurrent network dynamics may explode. If $\rho'(u)$ is also smaller than 1, and $\beta$ still small enough, we conclude again that the Hessian is invertible (and hence positive definite). Notice that the property of $H = \frac{\partial^2 L}{\partial u\,\partial u}$ being symmetric is a general property of differentiable functions and ways weaker than the requirement that the recurrent weight matrix $W_{\mathrm{net}}$ is symmetric, as classically required for fixed point convergence in Hopfield-type networks.

Since in the implicit differential equation of *Equation 46* the $\dot{u}$ also arises on the right-hand-side, we need to show that this *Equation 46*, for fixed $(r_{\mathrm{in}}, u_o^*, u)$, can be solved for $\dot{u}$. To find the solution $\dot{u}$ for given arguments $(r_{\mathrm{in}}, u_o^*, u)$ in $K$, we search for a fixed point of the mapping $\dot{u}^{(i+1)} = K(\dot{u}^{(i)})$. In the argument $\dot{u}$, the mapping is affine, $\dot{u}^{(i+1)} = -\frac{f}{\tau} - \frac{\partial f}{\partial t} + M\dot{u}^{(i)}$, with matrix $M(u) = \frac{\partial}{\partial u}(W\bar{r} + \bar{\epsilon} + \beta\bar{e}^*)$ appearing in *Equation 46*. The Banach fixed point theorem asserts that if $K(\dot{u})$ is strictly contracting with $k$, i.e., if $\|K(\dot{u}^{(2)}) - K(\dot{u}^{(1)})\|^2 \leq k\|\dot{u}^{(2)} - \dot{u}^{(1)}\|$ for all pairs of inputs and $0 \leq k < 1$, then the iteration (here with iteration time step $dt$) locally converges to a fixed point $\dot{u} = K(\dot{u})$. Because

$K(\dot{u}^{(2)}) - K(\dot{u}^{(1)}) = M(\dot{u}^{(2)} - \dot{u}^{(1)})$, the mapping is $k$-contractive if the eigenvalues of $M$ have absolute value smaller than $k$. This is the case if the Hessian $H(u) = 1 - M(u) = 1 - W_{\text{net}}\rho'(u) - \bar{e}'(u)$, with $\bar{e} = \bar{\epsilon} - \beta\bar{e}^*$, is strictly positive definite (which is 'typically' the case, see above).

## Stable solution of the implicit differential equation

According to the above reasoning, the Banach fixed point theorem asserts (for the typically positive Hessian) that during convergence of the mapping $\dot{u}^{(i+1)} = K(\dot{u}^{(i)})$ the distance to the fixed point is bounded by a constant times $e^{-i\frac{1-k}{dt}}$, with iteration index $i$ and 'virtual' Euler step $dt$. Formally,

$$\|K(\dot{u}^{(i+1)}) - K(\dot{u}^{(i)})\| = \|M(\dot{u}^{(i+1)} - \dot{u}^{(1)})\| \leq k\|\dot{u}^{(i+1)} - \dot{u}^{(i)}\| \leq c_0\, e^{-i\frac{1-k}{dt}} \longrightarrow \mathbf{0}. \tag{49}$$

Crucially, because the mapping is affine in $\dot{u}$, the convergence is global in $\dot{u}$ while fixing the other arguments $(r_{\text{in}}, u_o^*, u)$ of $K$.

In an analog physical device that implements exactly this feedback circuit in continuous time, the $dt$ becomes truly infinitesimal and in this sense the convergence is instantaneous. If $dt$ remains finite, $\dot{u}^{(i)}$ converges to a moving target because the mapping $\dot{u}^{(i+1)} = K(\dot{u}^{(i)})$ changes with time. The target should not move quicker than the time scale $\frac{dt}{1-k}$ of the $\dot{u}^{(i)}$ convergence (but, fortunately, this convergence time-scale can be made arbitrarily quick by reducing $dt$). Given a time course of the input rate $r_{\text{in}}(t)$ and target $u_o^*(t)$ that has bounded variation, the $dt$ can be chosen so that convergence becomes arbitrary quick, and in the limit instantaneous. As a consequence, for small enough $dt$, the $\dot{u}$ becomes the same on the left- and right-hand side of the implicit differential *Equation 46*.

If $r_{\text{in}}(t)$ contains well-separated delta-functions, while otherwise still having bounded variations, the reasoning still applies since at any point in time, except at the time points of the isolated delta-function, $f(u,t) = 0$. This statement is proven in the Appendix 4.

In practical simulations, a caveat is in place when the learning rate becomes too big and $\dot{W}$ starts to change the neuronal dynamics. In this case, the Hessian becomes $H = 1 - (W\text{diag}(\rho') + \dot{W}\text{diag}(\rho)) - \bar{e}'$, and the Eigenvalues may become negative due to the $\dot{W}$ term. Simulations can in fact become unstable with a big learning rate, and this is more pronounced if also the Euler $dt$ is large. The explicit differential equation avoids the fast iteration towards a moving target and hence allows for larger $dt$. This in particular pays out in the presence of a high learning rate (although the Cholesky decomposition also requires positive definiteness). For this reason, the large-scale simulations involving plasticity are performed with the explicit form described next.

## Explicit differential equation

To isolate $\dot{u}$ in the implicit differential equation, we rewrite $\left(1 + \tau\frac{d}{dt}\right)f = 0$ again as

$$\tau\dot{f} = \tau\left(\frac{\partial f}{\partial u}\dot{u} + \frac{\partial f}{\partial t}\right) = -f, \quad \text{or} \quad \tau\frac{df_i}{dt} = \tau\left(\sum_j \frac{\partial f_i}{\partial u_j}\dot{u}_j + \frac{\partial f_i}{\partial t}\right) = -f_i. \tag{50}$$

Plugging the Hessian $H(u) = \frac{\partial f}{\partial u}$ from *Equation 48* into *Equation 50*, we obtain the voltage dynamics from *Equation 22* in the equivalent form

$$\tau H(u)\dot{u} = -f - \tau\frac{\partial f}{\partial t}. \tag{51}$$

In our applications, the Hessian $H$ appears to be invertible (although this may not be the case for arbitrary networks), and *Equation 51* can be solved for the unique $\dot{u}$ using the Cholesky decompositions. In fact, if $H$ is invertible, the system implicit ordinary differential equations from *Equation 51* can be converted into a system of explicit ordinary differential equations (with $f(u,t) = \frac{\partial L}{\partial u} = u - W\bar{r} - \bar{\epsilon} - \beta\bar{e}^*$ and $H$ given in *Equation 48*),

$$\dot{u} = g(u,t) = -\frac{1}{\tau}H^{-1}(u)\left(f(u) + \tau\frac{\partial f}{\partial t}\right), \tag{52}$$

and as such it has a unique solution for any given initial condition. *Equation 52* represents the explicit solution of the iteration $\dot{u}^{(i+1)} = K(\dot{u}^{(i)})$ after the convergence to the fixed point $\dot{u} = K(\dot{u})$. The condition for the convergence is the same as the condition for solving for $\dot{u}$, namely an invertible (and hence positive definite) Hessian.

Because $\frac{\partial f_i}{\partial t} = W_{i,\text{in}}\dot{\bar{r}}_{\text{in}} + \beta\,\delta_{io}\dot{u}_o^*$, see *Equation 47*, the regularity requirement for $\tau\dot{u}$ to be integrable is satisfied even if $\bar{r}_{\text{in}}(t)$ and $u_o^*(t)$ contain step functions (and $r_{\text{in}}(t)$ delta-functions), see Appendix 4 for details.

Even in the presence of such step-functions, the Euler-Lagrange equations $\left(1 + \tau\frac{\mathrm{d}}{\mathrm{d}t}\right)f = 0$ lead to an $f$ that is a decaying exponential, $f_i(u(t), t) = c_i\,e^{-\frac{t-t_0}{\tau}}$, see *Equation 45*. For initialization at $t_0 = -\infty$ we have $f = \frac{\partial L}{\partial u} = 0$ at any time. In fact, *Equation 52* is equivalent to $\left(1 + \tau\frac{\mathrm{d}}{\mathrm{d}t}\right)f = 0$, and hence any solution of *Equation 52*, even if $\frac{\partial f}{\partial t}$ contains a delta-function, is also a solution of $\left(1 + \tau\frac{\mathrm{d}}{\mathrm{d}t}\right)f = 0$. Possible jumps in $\bar{r}_{\text{in}}$ or $u_o^*$ are compensated by the jumps they induce in $u$ (see below for the full mathematical description with a simple example). To give an intuition, we assume that a recurrent network of our prospective neurons has separate fixed points for the two constant input currents $r_{\text{in}}^{(1)}$ and $r_{\text{in}}^{(2)}$. This network still shows an overall relaxation time of $\tau_{\text{in}}$ (but not longer!) when the input rates instantaneously switch from $r_{\text{in}}^{(1)}$ to $r_{\text{in}}^{(2)}$. Nevertheless, at any moment during this relaxation process, gradient learning of the mapping $u_o(t) = F_o(\bar{r}_{\text{in}}(t))$ towards $u_o^*(t) = F_o^*(\bar{r}_{\text{in}}(t))$ is still guaranteed (Theorem 1).

## Link to time-varying optimal control

The explicit differential equation, *Equation 52*, is a special case of the one in *Simonetto et al., 2020*, *Equation 20*, where the function to be minimized (their $f$) can take a general (Lipschitz continuous) form (hence their $f$ is our Lagrangian, $f \equiv L$). To avoid inverting the Hessian, an iteration algorithm can be applied similar to our implicit form, although more involved to deal with the more general form of $L$ (*Simonetto and Dall'Anese, 2017*). The idea of tracking the solution of a time-varying optimization problem with a linear look-ahead in time has been introduced introduced in *Zhao and Swamy, 1998*.

## The NLA keeps stability as compared to an infinitesimally fast ($\tau \to 0$) equilibrium propagation

What have we gained as compared to making the time constant in Equilibrium propagation very small? The answer lies in the stability. To make the comparison explicit, we remind us of the Equilibrium Propagation which we formulate in terms of a Lagrangian. In both cases, the NLA and the Equilibrium Propagation, the Lagrangians are identical. We have chosen the notation $\bar{r} = (\bar{r}_{\text{in}}, \rho(u))$ to express that the quantities can be interpreted as a low-pass filtering of $r + \tau r$. In the Equilibrium propagation, they are called $r = (r_{\text{in}}, \rho(u))$, but this only represents a renaming of the input rates (and a notation change from $\bar{r}$ to $r$). Hence, the Lagrangian $L = \frac{1}{2}\|u - W\bar{r}\|^2 + \beta C$ in the NLA with $\bar{r} = (\bar{r}_{\text{in}}, \rho(u))$ is rewritten by $L = \frac{1}{2}\|u - Wr\|^2 + \beta C$ with $r = (r_{\text{in}}, \rho(u))$.

The only true difference is that in the NLA we apply the combined lookahead operator $\left(1 + \tau\frac{\mathrm{d}}{\mathrm{d}t}\right)\frac{\partial}{\partial u}$ to the Lagrangian $L$, while in the Equilibrium Propagation one applies the reduced operator $\frac{\partial}{\partial u}$ to $L$. We claim that the lookahead makes the difference for 'living' systems that learnt to optimize in-time with respect to future quantities. The Euler-Lagrange equations for the Equilibrium Propagation reduce to $\frac{\partial L}{\partial u} = u - Wr - \rho'(u)W^{\mathrm{T}}e - \beta e^* = 0$, where now $e = u - Wr$. Since these are not differential equations (no $\dot{u}$ in there), one introduces the gradient descent dynamics $\tau\dot{u} = -\frac{\partial L}{\partial u} = -u + Wr + \rho'(u)W^{\mathrm{T}}e + \beta e^*$ and waits for convergence for fixed input and fixed target. If $\tau = 0$ in both the NLA and the Equilibrium propagation, we obtain the same implicit and stationary equations in $u$.

Because the equation is implicit in $u$, it has to be solved first for $u$. While the Equilibrium propagation suggests to keep the inputs constant until a fixed point is reached, the NLA dynamically moves along the trajectory of fixed points while the inputs are changing. In the Equilibrium propagation we cannot simply take the limit $\tau \to 0$ since then the dynamics either disappears (when $\tau$ remains on the left, $\tau\Delta u \to 0$) or explodes (when $\tau$ is moved to the right, $\frac{\mathrm{d}t}{\tau} \to \infty$), leading to either too small or too big jumps.

## Appendix 4

### Contraction analysis and delta-function inputs

#### Contraction property

We next show when the voltage dynamics obtained from $\left(1 + \tau \frac{d}{dt}\right)\frac{\partial L}{\partial u} = 0$ is locally contracting, i.e., when neighboring trajectories of the explicit differential equations, for given inputs and targets, locally converge. This is a different stability property as compared to the (global) convergence properties for the implicit differential equation stated in *Equations 45 and 49*.

For the contraction analysis we rewrite *Equation 51* in the form $G(y, \dot{u}) = \left(1 + \tau \frac{d}{dt}\right)\frac{\partial L}{\partial u} = f + \tau \dot{f} = 0$, where the explicit time dependence of $E$ and $f$ is a short-cut to express the dependence on $y = (r_{\text{in}}, u_o^*, u)$. According to the implicit function theorem, at any point in time when $\frac{\partial G}{\partial \dot{u}}$ is invertible, we can locally write $\dot{u} = g(y)$. When absorbing the dependence of $\dot{u}$ on $(r_{\text{in}}, u_o^*)$ into an explicit time dependence, we can rewrite this differential equation as $\dot{u} = g(u, t)$, see *Equation 52*, when explicitly expressing the dependency of $g$ on $u$ only (instead of all the variables $y$). This differential equation is contractive and thus stable if the Jacobian of $g$ with respect to $u$ is uniformly negative definite (*Lohmiller and Slotine, 1998*). The contraction analysis tells that locally, where $\frac{\partial G}{\partial \dot{u}}$ is invertible, we can express $\dot{u}$ as a function $\dot{u} = g(y)$, that has derivative $\frac{\partial g}{\partial y} = -\left(\frac{\partial G}{\partial \dot{u}}\right)^{-1} \frac{\partial G}{\partial y}$ according to the implicit function theorem. Restricted to the $u$-component of $y$ we get the Jacobian

$$\frac{\partial g}{\partial u} = -\left(\frac{\partial G}{\partial \dot{u}}\right)^{-1} \frac{\partial G}{\partial u} = -\frac{1}{\tau}\mathbf{1} - H^{-1}\frac{\partial H}{\partial u}\dot{u} = -\frac{1}{\tau}\mathbf{1} - \frac{d\log H}{dt}. \tag{53}$$

To prove *Equation 53* we note that according to *Equation 51* we have $G(y, \dot{u}) = f + \tau H(u)\dot{u} + \tau \frac{\partial f}{\partial t} = 0$, and with this calculate $\frac{\partial G}{\partial \dot{u}} = \tau H$, with $H_{ij} = \frac{\partial f_i}{\partial u_j}$ specified above. Since according to *Equation 47* the term $\frac{\partial f}{\partial t}$ does not depend on $u$, we also calculate $\frac{\partial G}{\partial u} = H + \tau \frac{\partial H}{\partial u}\dot{u} = H + \tau \frac{dH}{dt}$. Hence, $\frac{\partial g}{\partial u} = -\frac{1}{\tau}H^{-1}\left(H + \tau \frac{dH}{dt}\right) = -\frac{1}{\tau}\mathbf{1} - \frac{d\log H}{dt}$.

If we had $\frac{\partial g}{\partial u} = -\frac{1}{\tau}\mathbf{1}$ alone, $\frac{\partial g}{\partial u}$ would be obviously negative definite, guaranteeing the local contraction property (*Lohmiller and Slotine, 1998*). To understand this statement, we consider the local representation of neighboring trajectories. The linear approximation of the voltage dynamics $u(t)$, starting with $u(t_\circ)$ in a dim-$u$-dimensional neighbourhood around some point $u_\circ(t_\circ)$, is $\frac{d}{dt}(u - u_\circ) = \frac{\partial g(u_\circ, t)}{\partial u}(u - u_\circ) = -\frac{1}{\tau}(u - u_\circ)$. While $u(t)$ is the linear approximations of the original voltage dynamics starting at $u(t_\circ)$, the trajectory $u_\circ(t)$ is the true voltage dynamics starting at the 'center' $u_\circ(t_\circ)$ of the neighborhood. This shows the exponential local contraction of surrounding trajectories to $u_\circ$ at times $t$ around $t_\circ$, would we have $\frac{\partial g}{\partial u} = -\frac{1}{\tau}\mathbf{1}$.

The additional log-term in *Equation 53* may cause a local violation of this contraction property. However, the additional term in *Equation 53* becomes small for large or small voltages for which we assume that the curvature of the transfer function also becomes small, $\rho''(u) \approx \mathbf{0}$. In fact, based on *Equation 48* we calculate $\frac{\partial H}{\partial u}$ being a function of $\rho''(u)$ that vanishes with vanishing $\rho''(u)$,

$$\frac{\partial H(u)}{\partial u} = -W_{\text{net}} \operatorname{diag}\left(\rho''(u)\right) - \frac{\partial^2}{\partial u^2}(\bar{e} + \beta \bar{e}^*)(u) = o(\rho''(u)). \tag{54}$$

Plugging *Equation 54* into *Equation 53* we conclude that $\frac{\partial g}{\partial u} \approx -\frac{1}{\tau}\mathbf{1}$ where the curvature of the transfer function vanishes, $\rho''(u) \approx \mathbf{0}$. Yet, $\rho''$ may strongly deviate from 0 for intermediate values of $u$. For a rectified linear unit (ReLu), $\rho(u) = u$ for $u \geq 0$ and 0 else, for instance, $\rho''$ is a delta-function at 0. Even if the deviation from $\rho''(u) = \mathbf{0}$ in this case is on a set of measure 0, integrating across voltages $u = 0$ can cause a jump in the voltage dynamics, preventing a strict local contraction property everywhere. The contraction property, in the example of a ReLu, only holds almost everywhere. Because the ReLu is an important transfer function in practical applications, we elaborate on this example further.

#### Comparison of the NLA with the latent equilibrium

Here, we point out that the appearance of a delta-function for ReLu's in the NLA may represent a technical challenge, that this specific challenge is tamed in the simpler version of the NLA principle

formulated as Latent Equilibrium (*Haider et al., 2021*), and that the Latent Equilibrium may represent a viable new approach for time-varying optimization in general.

The Euler-Lagrange equation for the presented NLA, $\left(1 + \tau \frac{\mathrm{d}}{\mathrm{d}t}\right)\frac{\partial L}{\partial \boldsymbol{u}} = 0$, *Equation 20*, with Jacobian $\frac{\partial L}{\partial \boldsymbol{u}} = \boldsymbol{f}(\boldsymbol{u}, t) = \boldsymbol{u} - \boldsymbol{W}\bar{\boldsymbol{r}} - \bar{\boldsymbol{\epsilon}} - \beta \bar{\boldsymbol{e}}^*$, implies second-order derivatives with respect to the voltage $\boldsymbol{u}$. This is the case because the total derivative with respect to time, $\frac{\mathrm{d}}{\mathrm{d}t}$, applied to the error $\bar{\boldsymbol{\epsilon}} = \bar{\boldsymbol{r}}'_{\mathrm{net}} \boldsymbol{W}_{\mathrm{net}}^{\mathrm{T}} \bar{\boldsymbol{e}}$ contained in $\boldsymbol{f}(\boldsymbol{u}, t)$, implies the expression $\frac{\mathrm{d}}{\mathrm{d}t}\bar{\boldsymbol{r}}'_{\mathrm{net}} = \rho''(\boldsymbol{u})\dot{\boldsymbol{u}}$, with $''$ equivalent to $\frac{\partial^2}{\partial \boldsymbol{u} \partial \boldsymbol{u}}$. These second-order derivatives enter both in the implicit form (*Equations 46 and 51*) and the explicit differential *Equation 52*. When integrating these equations across the components $u = 0$, a jump in $u$ emerges that must be explicitly included in the simulations. It is the same jump that may transiently violate the contraction property as shown above (*Equations 53 and 54*, but does not violate the exponential convergence from *Equation 45*, see below).

In the Latent Equilibrium, our Lagrangian $L = \frac{1}{2}\|\boldsymbol{u} - \boldsymbol{W}\bar{\boldsymbol{r}}\|^2 + \beta C$ with $\bar{\boldsymbol{r}} = (\bar{\boldsymbol{r}}_{\mathrm{in}}, \rho(\boldsymbol{u}))$ is replaced by $L(\boldsymbol{u}, \dot{\boldsymbol{u}}) = \frac{1}{2}\|\breve{\boldsymbol{u}} - \boldsymbol{W}\boldsymbol{r}\|^2 + \beta C$ with $\boldsymbol{r} = (\boldsymbol{r}_{\mathrm{in}}, \rho(\breve{\boldsymbol{u}}))$ and $\breve{\boldsymbol{u}} = \boldsymbol{u} + \tau \dot{\boldsymbol{u}}$. If we were asking for stationary trajectories $\boldsymbol{u}(t)$ of $\int L(\boldsymbol{u}, \dot{\boldsymbol{u}})\mathrm{d}t$ with respect to variations in $\boldsymbol{u}$, we would obtain the unstable Euler-Lagrange equations $\frac{\partial L}{\partial \boldsymbol{u}} - \frac{\mathrm{d}}{\mathrm{d}t}\frac{\partial L}{\partial \dot{\boldsymbol{u}}} = \left(1 - \tau \frac{\mathrm{d}}{\mathrm{d}t}\right)\frac{\partial L}{\partial \boldsymbol{u}} = 0$, solved by an exponentially growing function of time, $\frac{\partial L}{\partial \boldsymbol{u}} = \boldsymbol{c}\, e^{\frac{t - t_0}{\tau}}$, unless $\boldsymbol{c} = 0$. To force the trajectory moving along $\frac{\partial L}{\partial \boldsymbol{u}} = 0$, one again requires that variations of the action $\int L \, \mathrm{d}t$ are stationary with respect to a prospective voltage, this time with respect to $\breve{\boldsymbol{u}} = \boldsymbol{u} + \tau \dot{\boldsymbol{u}}$. Because the Lagrangian for the Latent Equilibrium can be expressed as a function $L(\breve{\boldsymbol{u}})$ of $\breve{\boldsymbol{u}}$ only, not of $\dot{\breve{u}}$, the Euler-Lagrange equations becomes $\frac{\partial L}{\partial \breve{\boldsymbol{u}}} = \breve{\boldsymbol{u}} - \boldsymbol{W}\boldsymbol{r} - \rho'(\breve{\boldsymbol{u}})\boldsymbol{W}^{\mathrm{T}}\boldsymbol{e} - \beta \boldsymbol{e}^* = 0$, where now $\boldsymbol{e} = \breve{\boldsymbol{u}} - \boldsymbol{W}\boldsymbol{r}$.

The Latent Equilibrium is consistent with the central idea of the NLA to deal with future quantities. While the presented version of the NLA deals with the discounted future voltage $\tilde{\boldsymbol{u}}$ (leading to the operator $\left(1 + \tau \frac{\mathrm{d}}{\mathrm{d}t}\right)\frac{\partial}{\partial \boldsymbol{u}}$ applied to the Lagrangian), the Latent Equilibrium deals with the linear approximation $\breve{\boldsymbol{u}}$ of the future voltage (leading to the operator $\frac{\partial}{\partial \breve{\boldsymbol{u}}}$ applied to the Lagrangian). Both, the NLA and the Latent Equilibrium differ in this crucial aspect from the Equilibrium Propagation that deals with the current voltage $\boldsymbol{u}$ (leading to the operator $\frac{\partial}{\partial \boldsymbol{u}}$ applied to the Lagrangian $L(\boldsymbol{u})$, without appearance of $\dot{\breve{u}}$, and hence without intrinsic dynamics, see above).

For the Latent Equilibrium, it is not obvious to find global convergence properties as expressed in *Equations 45 and 49* for the presented version of the NLA. However, the Latent Equilibrium satisfies the strict local contraction property $\frac{\partial \boldsymbol{g}}{\partial \boldsymbol{u}} = -\frac{1}{\tau}\boldsymbol{1}$. In fact, writing the Euler-Lagrange equations as $\boldsymbol{G}(\boldsymbol{y}, \dot{\boldsymbol{u}}) = \frac{\partial L}{\partial \breve{\boldsymbol{u}}} = 0$ with the abbreviation $\boldsymbol{y} = (\boldsymbol{r}_{\mathrm{in}}, \dot{\boldsymbol{r}}_{\mathrm{in}}, \boldsymbol{u}_o^*, \dot{\boldsymbol{u}}_o^*, \boldsymbol{u})$, leads to $\frac{\partial \boldsymbol{G}}{\partial \breve{\boldsymbol{u}}} = \frac{\partial \boldsymbol{G}}{\partial \boldsymbol{u}} = \frac{1}{\tau}\frac{\partial \boldsymbol{G}}{\partial \dot{\boldsymbol{u}}}$, and the linear approximation obtained from plugging these partial derivatives into *Equation 53* yields the claimed contraction property $\frac{\partial \boldsymbol{g}}{\partial \boldsymbol{u}} = -\frac{1}{\tau}\boldsymbol{1}$ for the Latent Equilibrium. The Jacobian $\frac{\partial \boldsymbol{G}}{\partial \dot{\boldsymbol{u}}}$ is invertible if the Hessian $\boldsymbol{H} = \frac{\partial^2 L}{\partial \breve{\boldsymbol{u}} \partial \breve{\boldsymbol{u}}}$ is (that looks as in *Equation 48*, but with $\boldsymbol{u}$ replaced by $\breve{\boldsymbol{u}}$).

Crucially, for the Latent Equilibrium, the analogous theorems on real-time dendritic error propagation and learning (rt-DeEP and rt-DeEL) hold. The key property is that the trajectories of the Latent Equilibrium are stationary solution of the corresponding action $A = \int L \, \mathrm{d}t$, so that the variation of $A$ induced by $\boldsymbol{W}$ reduces to the partial derivative of $L$ with respect to $\boldsymbol{W}$.

The above strict contraction property that for non-vanishing errors only holds for the Latent Equilibrium, but not for the NLA, may be a reason why the simulations of the error-based updates in our hands seem to be more stable for the Latent Equilibrium as compared to the NLA. This does not withstand the fact that the NLA equations are used in the context of time-varying optimal control (see above). It would be interesting to consider also the Latent Equilibrium as a tool for non-stationary, time-varying optimization (*Simonetto et al., 2020*).

## Delta-function inputs keep $\frac{\partial L}{\partial \boldsymbol{u}} = 0$

We next explain in more details why delta-functions in the input rates $\boldsymbol{r}_{\mathrm{in}}(t)$ for the NLA the stationarity ('equilibrium') condition $\frac{\partial L}{\partial \boldsymbol{u}} = 0$ is always satisfied (the delta-function in $\boldsymbol{r}_{\mathrm{in}}$ for the NLA corresponding to step-function in $\boldsymbol{r}_{\mathrm{in}}$ for the Latent Equilibrium). We reconsider the explicit differential equation $\dot{\boldsymbol{u}} = \boldsymbol{g}(\boldsymbol{u}, t) = -\frac{1}{\tau}\boldsymbol{H}^{-1}\left(\boldsymbol{f} + \tau \frac{\partial \boldsymbol{f}}{\partial t}\right)$ given in *Equation 52*, with $\boldsymbol{f}(\boldsymbol{u}, t) = \frac{\partial L}{\partial \boldsymbol{u}} = \boldsymbol{u} - \boldsymbol{W}\bar{\boldsymbol{r}} - \bar{\boldsymbol{\epsilon}} - \beta \bar{\boldsymbol{e}}^*$ and $\boldsymbol{H}$ given in *Equation 48*.

To simplify matters, we consider a single delta-function at $t = 0$ as input in the absence of output nudging. In this case, we get $\frac{\partial \boldsymbol{f}}{\partial t} = \frac{\partial \boldsymbol{f}}{\partial \boldsymbol{r}_{\mathrm{in}}}\dot{\boldsymbol{r}}_{\mathrm{in}} + \frac{\partial \boldsymbol{f}}{\partial \boldsymbol{r}_{\mathrm{in}}}\dot{\boldsymbol{u}}_o^* = \boldsymbol{W}_{\mathrm{in}}\boldsymbol{\delta}_{\mathrm{in}}(t)$, where the input matrix $\boldsymbol{W}_{\mathrm{in}}$ is typically sparse (not all network neurons receive external input), and $\boldsymbol{\delta}_{\mathrm{in}}(t)$ is a vector of delta-functions restricted to the input neurons. Following (*Nedeljkov and Oberguggenberger, 2012*), Proposition 2.1, we can then write the explicit differential equation, *Equation 51*, in the form

$$\dot{\boldsymbol{u}} = \boldsymbol{g}(\boldsymbol{u}, t) = \check{\boldsymbol{g}}(\boldsymbol{u}, t) + \boldsymbol{H}(\boldsymbol{u})^{-1} \boldsymbol{W}_{\text{in}} \boldsymbol{\delta}_{\text{in}}(t), \tag{55}$$

where $\check{\boldsymbol{g}}(\boldsymbol{u}, t)$ is globally Lipschitz continuous. Due to the Lipschitz continuity the change in $\boldsymbol{u}$ evoked by $\check{\boldsymbol{g}}(\boldsymbol{u}, t)$ during a small time interval $[-\varepsilon, \varepsilon]$ around $t = 0$ vanishes when this interval shrinks, $\varepsilon \to 0$. To quantify the change in $\boldsymbol{u}$ during these intervals it is, therefore, enough to consider $\dot{\boldsymbol{u}} = \boldsymbol{H}(\boldsymbol{u})^{-1} \boldsymbol{W}_{\text{in}} \boldsymbol{\delta}_{\text{in}}(t)$, or equivalently $\boldsymbol{H}(\boldsymbol{u})\dot{\boldsymbol{u}} = \boldsymbol{W}_{\text{in}} \boldsymbol{\delta}_{\text{in}}(t)$. To estimate the jump induced by the delta-functions, we consider some mollifier $\phi_\varepsilon(t) = \varepsilon^{-1}\phi(t/\varepsilon)$, where $\phi(t)$ is a smooth function on the interval $[-1, 1]$ with integral 1. By $\phi_{\text{in},\varepsilon}(t)$ we denote the vector of mollifiers centered at the delta-functions of the input neurons. We now consider the two differential equations, with the second approximating the first on the interval $[-\varepsilon, \varepsilon]$, but without regular term $\check{\boldsymbol{g}}(\boldsymbol{u}, t)$,

$$\tau \boldsymbol{H}(\boldsymbol{u}_\varepsilon) \dot{\boldsymbol{u}}_\varepsilon = \tau \boldsymbol{H}(\boldsymbol{u}_\varepsilon) \check{\boldsymbol{g}}(\boldsymbol{u}, t) + \boldsymbol{W}_{\text{in}} \phi_{\text{in},\varepsilon}(t) \tag{56a}$$

$$\tau \boldsymbol{H}(\check{\boldsymbol{u}}_\varepsilon) \dot{\check{\boldsymbol{u}}}_\varepsilon = \boldsymbol{W}_{\text{in}} \phi_{\text{in},\varepsilon}(t), \quad \check{\boldsymbol{u}}_\varepsilon(-\varepsilon) = \boldsymbol{u}_\varepsilon(-\varepsilon). \tag{56b}$$

We assume that for all $t \in [-\varepsilon, \varepsilon]$ the matrices $\boldsymbol{H}(\boldsymbol{u}_\varepsilon(t))$ and $\boldsymbol{H}(\check{\boldsymbol{u}}_\varepsilon(t))$ are invertible, so that the two *Equation 56a*, *Equation 56b* can be turned into an explicit differential equations. Analogously to the 1-dimensional case (*Nedeljkov and Oberguggenberger, 2012*), we conclude that the solution of *Equation 56a*, *Equation 56b* on the interval $[-\varepsilon, \varepsilon]$ converge to each other, $\sup_{t \in [-\varepsilon, \varepsilon]} |\boldsymbol{u}_\varepsilon(t) - \check{\boldsymbol{u}}_\varepsilon(t)| \to 0$ for $\varepsilon \to 0$. As a consequence, the jump of $\check{\boldsymbol{u}}_\varepsilon$ at $t = 0$ converges to the corresponding jump of $\boldsymbol{u}_\varepsilon$ in the various dimensions.

To calculate the jump of $\check{\boldsymbol{u}}_\varepsilon$ we have to integrate $\tau \boldsymbol{H}(\check{\boldsymbol{u}}_\varepsilon) \dot{\check{\boldsymbol{u}}}_\varepsilon$ across the time interval $[-\varepsilon, \varepsilon]$. Instead of integrating $\dot{\check{\boldsymbol{u}}}_\varepsilon$, we first integrate $\tau \boldsymbol{H}(\check{\boldsymbol{u}}_\varepsilon) \dot{\check{\boldsymbol{u}}}_\varepsilon$ given in *Equation 56b*. Moving from right to left yields

$$\boldsymbol{W}_{\text{in}} \boldsymbol{I}_{\text{in}} = \int_{-\varepsilon}^{\varepsilon} \boldsymbol{W}_{\text{in}} \phi_{\text{in},\varepsilon}(t) \, dt = \tau \int_{-\varepsilon}^{\varepsilon} \boldsymbol{H}(\check{\boldsymbol{u}}_\varepsilon(t)) \dot{\boldsymbol{v}}_\varepsilon(t) \, dt = \tau \int_{\check{\boldsymbol{u}}_\varepsilon(-\varepsilon)}^{\check{\boldsymbol{u}}_\varepsilon(\varepsilon)} \boldsymbol{H}(\check{\boldsymbol{u}}_\varepsilon) \, d\check{\boldsymbol{u}}_\varepsilon, \tag{57}$$

where $\boldsymbol{I}_{\text{in}}$ is the index vector of the input neurons having a delta-function, i.e., $I_{\text{in},j} = 1$ if there is a delta-input, and else 0. Note that $\dot{v}_{\varepsilon,j} \, dt = dv_{\varepsilon,j}$. Because $\boldsymbol{H}$ is itself a derivative, $\boldsymbol{H} = \frac{\partial f}{\partial \boldsymbol{u}}$, we can explicitly calculate the latter integral (also for $\beta > 0$, but for clarity here only done for $\beta = 0$). The last integral in *Equation 57* is defined as a vector with $i$-th component being

$$\left( \int_{\check{\boldsymbol{u}}_\varepsilon(-\varepsilon)}^{\check{\boldsymbol{u}}_\varepsilon(\varepsilon)} \boldsymbol{H}(\check{\boldsymbol{u}}_\varepsilon) \, d\check{\boldsymbol{u}}_\varepsilon \right)_i = \sum_{j=1}^{N} \int_{\check{u}_{\varepsilon,j}(-\varepsilon)}^{\check{u}_{\varepsilon,j}(\varepsilon)} H_{ij}(u_j) \, du_j = \sum_{j=1}^{N} \left( \delta_{ij} u_i - W_{ij} \rho(u_j) \right) \Big|_{\check{u}_{\varepsilon,j}(-\varepsilon)}^{\check{u}_{\varepsilon,j}(\varepsilon)} = \left( \boldsymbol{v}_\varepsilon(\varepsilon) - \boldsymbol{v}_\varepsilon(-\varepsilon) \right)_i, \tag{58}$$

where in the second last equality we used that $H_{ij}(\boldsymbol{u}) = \delta_{ij} - W_{ij} \rho'(u_j)$ does only depend on the component $u_j$, see *Equation 48*. In the last equality we introduced the 'network voltage error' $\check{\boldsymbol{v}}_\varepsilon = \check{\boldsymbol{u}}_\varepsilon - \boldsymbol{W}_{\text{net}} \rho(\check{\boldsymbol{u}}_\varepsilon)$. Following the 1-dimensional case treated in *Equation 66*, Proposition 1.2 we introduce the 'jump function' (called $G(y)$ in the cited work)

$$\boldsymbol{J}(\check{\boldsymbol{u}}_\varepsilon) = \tau (\check{\boldsymbol{v}}_\varepsilon - \check{\boldsymbol{v}}_\circ), \tag{59}$$

with $\check{\boldsymbol{v}}_\circ$ thought to represent $\check{\boldsymbol{v}}_\varepsilon(t_\circ)$ at some time $t_\circ$ before the delta-kick sets in. With this setting, *Equation 57* turns into

$$\boldsymbol{W}_{\text{in}} \boldsymbol{I}_{\text{in}} = \boldsymbol{J}(\check{\boldsymbol{u}}_\varepsilon(\varepsilon)) - \boldsymbol{J}(\check{\boldsymbol{u}}_\varepsilon(-\varepsilon)). \tag{60}$$

In the limit $\varepsilon \to 0$ we get a relation between $\boldsymbol{u}(t)$ immediately before and after the jump, $\boldsymbol{u}(-0)$ and $\boldsymbol{u}(+0)$, using that in this limit the boundary points of the trajectories also converge, $\check{\boldsymbol{u}}_\varepsilon(\pm\varepsilon) \to \boldsymbol{u}(\pm 0)$,

$$\boldsymbol{J}(\boldsymbol{u}(+0)) = \boldsymbol{J}(\boldsymbol{u}(-0)) + \boldsymbol{W}_{\text{in}} \boldsymbol{I}_{\text{in}}. \tag{61}$$

We now assume that the function $\boldsymbol{J}(\boldsymbol{u}) = \tau(\boldsymbol{v} - \boldsymbol{v}_\circ)$ is invertible around the jump. This is the case if the Jacobian $\frac{\partial \boldsymbol{J}(\boldsymbol{u})}{\partial \boldsymbol{u}}$ is invertible, and because $\boldsymbol{v} = \boldsymbol{u} - \boldsymbol{W}_{\text{net}} \rho(\boldsymbol{u})$, we require invertability of $\frac{\partial \boldsymbol{J}(\boldsymbol{u})}{\partial \boldsymbol{u}} = \tau \boldsymbol{H}(\boldsymbol{u})$, with Hessian defined in *Equation 48*.

In the case of invertability we get the voltage after the jump as

$$\boldsymbol{u}(+0) = \boldsymbol{J}^{-1} \left( \boldsymbol{J}(\boldsymbol{u}(-0)) + \boldsymbol{W}_{\text{in}} \boldsymbol{I}_{\text{in}} \right). \tag{62}$$

We next calculate the jump in $v$. This is easy since $v = u - W_{\text{net}}\rho(u)$ linearly enters in the function $J(u) = \tau(v - v_\circ)$. Plugging the explicit expression for $J$ into *Equation 61* we get $\tau(v(+0) - v_\circ) = \tau(v(-0) - v_\circ) + W_{\text{in}}I_{\text{in}}$, or

$$v(+0) = v(-0) + \frac{1}{\tau}W_{\text{in}}I_{\text{in}}. \tag{63}$$

Knowing the jump in $v$ helps to show that the equilibrium condition $\frac{\partial L}{\partial u} = 0$ is always satisfied, even immediately after the delta-in put, provided the initialization is at $t_0 = -\infty$. To show this, remember that in the absence of nudging we have $\frac{\partial L}{\partial u} = f(u, t) = u - W\bar{r} = u - W_{\text{net}}\rho(u) - W_{\text{in}}\bar{r}_{\text{in}} = v - W_{\text{in}}\bar{r}_{\text{in}}$. The jump size of $\bar{r}_{\text{in}}$ for a delta-function at $t=0$, $r_{\text{in}}(t) = \delta_{\text{in}}(t)$ is $\frac{1}{\tau}$. This is because $\bar{r}_{\text{in}}$ satisfies the differential equations $\dot{\bar{r}}_{\text{in}}(t) = -\frac{1}{\tau}\bar{r}_{\text{in}}(t) + \frac{1}{\tau}\delta_{\text{in}}(t)$, provided that $t_0 = -\infty$. Hence,

$$\bar{r}_{\text{in}}(+0) = \bar{r}_{\text{in}}(-0) + \frac{1}{\tau}I_{\text{in}}. \tag{64}$$

With *Equations 63 and 64* we conclude that $\frac{\partial L}{\partial u} = v - W_{\text{in}}\bar{r}_{\text{in}} = 0$ throughout.

## Appendix 5

### Example of a single recurrently connected neuron

To get an intuition for the instantaneity in the recurrent case we consider the example of a single, recurrently connected neuron. We also put this into the context of the Latent Equilibrium (*Haider et al., 2021*). Consider the weight vector $W = (W_{in}, W_{net})$ with an input rate $r_{in}(t)$ driving the postsynaptic voltage $u$. The postsynaptic rate is $r = \rho(u) + \tau\dot\rho(u)$, and its low-pass filter with respect to $\tau$ is $\bar r = \rho(u)$. As always, the low-pass filtering reaches back to an initialization at $t_0 = -\infty$, see *Equation 15*. The Lagrangian has the form

$$L = \frac{1}{2}\bar e^2 + \frac{\beta}{2}\bar e^{*2} = \frac{1}{2}\left(u - (W_{net}\rho(u) + W_{in}\bar r_{in})\right)^2 + \frac{\beta}{2}(u^* - u)^2 . \tag{65}$$

The Euler-Lagrange equations $\left(1 + \tau\frac{d}{dt}\right)\frac{\partial L}{\partial u} = 0$ are derived from $\frac{\partial L}{\partial u} = \bar e - \rho'(u)W_{net}\bar e - \beta\bar e^*$. Applying the look-ahead operator $\left(1 + \tau\frac{d}{dt}\right)$ (*Equation 14*), and abbreviating $\bar\epsilon = \rho'(u)W_{net}\bar e$, the Euler-Lagrange equations deliver the voltage dynamics,

$$\tau\dot u = -u + W_{net}\left(\rho(u) + \tau\dot\rho(u)\right) + W_{in}r_{in} + \epsilon + \beta e^* . \tag{66}$$

To simplify matters, we consider the nudging-free case, $\beta = 0$. This implies that $\bar e = \bar\epsilon = 0$. With $\dot\rho(u) = \rho'(u)\dot u$, we obtain the differential equation

$$\tau\left(1 - W_{net}\rho'(u)\right)\dot u = -u + W_{net}\rho(u) + W_{in}r_{in} . \tag{67}$$

Abbreviating $v = u - W_{net}\rho(u)$ as 'network voltage error,' the above differential equation reads as

$$\tau\dot v = -v + W_{in}r_{in} . \tag{68}$$

Integrating the effective voltage dynamics (*Equation 68*), assuming initialization infinitely far in the past, is equal to $v = W_{in}\bar r_{in}$. This equation is equivalent to the Euler-Lagrange equation $\left(1 + \tau\frac{d}{dt}\right)\frac{\partial L}{\partial u} = 0$ being integrate, and because the solution of the Euler Lagrange equation is $\frac{\partial L}{\partial u} = c\,e^{-\frac{t-t_0}{\tau}}$, we have (using $t_0 = -\infty$)

$$\frac{\partial L}{\partial u} = v - W_{in}\bar r_{in} = 0 . \tag{69}$$

### Voltage dynamics for a delta-function input

We next apply a delta-function in the input rate, say $r_{in}(t) = \delta(t)$ and consider the dynamics at the level of the voltage, *Equation 67*. As in *Equation 66*, Proposition 1.2 we introduce the 'jump function' (called $G(y)$ in the cited work)

$$J(u) = \tau\int_{u_\circ}^{u}\left(1 - W_{net}\rho'(y)\right)dy = \tau\left(\left(u - W_{net}\rho(u)\right) - v_\circ\right) = \tau(v - v_\circ) . \tag{70}$$

As in *Nedeljkov and Oberguggenberger, 2012*, Proposition 1.2, we show that the voltage $u$ makes a unique jump at the moment of the delta function that $J(u)$ is invertible around the jump.

We set $v_\circ = u_\circ - W_{net}\rho(u_\circ)$. Here, $u_\circ$ is some voltage before the jump, say $u_\circ = u(-1)$ evaluated at time $t = -1$, when the jump is at $t = 0$. When $u_0^- = u(-0)$ is the voltage immediately before the jump, the voltage immediately after the jump is $u_0^+ = u(+0)$ specified by

$$J(u_0^+) = J(u_0^-) + W_{in} . \tag{71}$$

The reason is that the $W_{in}$-scaled delta-function triggers a step of size $W_{in}$ when integrating over it as done in *Equation 70*. The new value $u_0^+$ is unique if $J$ is invertible, and looking at the defining integral in *Equation 70*, this is the case if $1 - W_{net}\rho'(u_0^+) \neq 0$.

The jump in $u$ translates into a jump in $v = u - W_{net}\rho(u)$ from $v_0^-$ to $v_0^+ = u_0^+ - W_{net}\rho(u_0^+)$. This endpoint can also be expressed as

$$v_0^+ = v_0^- + \frac{W_{in}}{\tau} . \tag{72}$$

To check this, we assume without loss of generality that $v_\circ = u_\circ - W_{\text{net}}\rho(u_\circ) = 0$. Then $J(u) = \tau(u - W_{\text{net}}\rho(u)) = \tau v$ and $J(u_0^+) = J(u_0^-) + W_{\text{in}} = \tau v_0^- + W_{\text{in}}$ according to **Equation 71**. Since also $J(u_0^+) = \tau v_0^+$, we conclude that $\tau v_0^+ = \tau v_0^- + W_{\text{in}}$, as claimed above.

We finally show that even far away from the initialization, the stationarity condition $\frac{\partial L}{\partial u} = 0$ holds before and immediately after the jump. In fact, for $t_0 = -\infty$, the evolution of the 'network voltage error' becomes

$$v(t) = 0 \text{ for } t \leq 0, \quad \text{and} \quad v(t) = \frac{W_{\text{in}}}{\tau} e^{-\frac{t}{\tau}} \text{ for } t > 0. \tag{73}$$

Here, we used that $v_0^- = 0$ and according to **Equation 72** the $v$ jumps to $v_0^- = \frac{W_{\text{in}}}{\tau}$. Remember, that for initialization far in the past, $\frac{\partial L}{\partial u} = 0$ is equivalent to $v = W_{\text{in}}\bar{r}_{\text{in}}$, see **Equation 69**. We, therefore, have to calculate the jump in $\bar{r}_{\text{in}}$ induced by the delta-input. Since $\bar{r}_{\text{in}}(t)$ is the solution of $\tau\dot{\bar{r}}_{\text{in}}(t) = -\bar{r}_{\text{in}}(t) + \delta(t)$ for $t_0 = -\infty$, we find that

$$\bar{r}_{\text{in}}(t) = 0 \text{ for } t \leq 0, \quad \text{and} \quad \bar{r}_{\text{in}}(t) = \frac{1}{\tau} e^{-\frac{t}{\tau}} \text{ for } t > 0. \tag{74}$$

Combing the two **Equations 73 and 74** proves that $\frac{\partial L}{\partial u} = v - W_{\text{in}}\bar{r}_{\text{in}} = 0$ holds true any moment in time, provided the initialization is far in the past.

One may ask why the delta-kink is different from resetting $v$ at a new initialization off from 0. The reason is that at $t = 0$ there is a cause for the jump in $r(0)$, while at $t_0$ there is no cause in $r(t_0)$. In fact, there is no jump initially, just the start of $v$ at some initial condition. Differently from the initialization at $t_0$, where $v(t_0) > 0$ implies $\frac{\partial L}{\partial u}(t) > 0$ for finite $t - t_0 > 0$, the jump of $v(0)$ at $t = 0$ to a positive value does leave $\frac{\partial L}{\partial u}(t) = 0$ for all $t > 0$, provided $t_0 = -\infty$.

## Linear transfer function

We first consider the case of a linear transfer-function $\rho(u) = 0$ (or threshold linear, being in the linear regime). Then the differential equation becomes

$$\tau\dot{u} = -u + \frac{W_{\text{in}}}{1 - W_{\text{net}}} r_{\text{in}}. \tag{75}$$

With initialization at $t = -\infty$ and low-pass filtering $\bar{r}_{\text{in}}^\tau$ defined in **Equation 15** the solution is

$$u(t) = \frac{W_{\text{in}}}{1 - W_{\text{net}}} \bar{r}_{\text{in}}^\tau(t). \tag{76}$$

The point is that the time constant is $\tau$ and is not $\frac{\tau}{1 - W_{\text{net}}}$, as this would be the case without prospective firing rate. In fact, for the 'classical' differential equation,

$$\tau\dot{u} = -u + W_{\text{net}}\rho(u) + W_{\text{in}} r_{\text{in}}, \tag{77}$$

and $\rho$ the identity, we obtain the differential equation $\tau_{\text{eff}}\dot{u} = -u + \frac{W_{\text{in}}}{1 - W_{\text{net}}} r_{\text{in}}$ with $\tau_{\text{eff}} = \frac{\tau}{1 - W_{\text{net}}}$ and solution $u = \frac{W_{\text{in}}}{1 - W_{\text{net}}} \bar{r}_{\text{in}}^{\tau_{\text{eff}}}$ that is now the low-pass filtering with respect to the effective time constant.

## Sigmoidal transfer function

For a sigmoidal transfer function $\rho(u) = \frac{\bar{r}_{\text{max}}}{1 + e^{\vartheta - u}}$, a positive feedback weight $W_{\text{net}} > 0$, and a constant external input $r_{\text{in}}$, say, the solutions $u(t)$ of **Equation 66** either converge to a fixed point or diverge. When converging, the voltage satisfies the fixed point condition

$$u = W_{\text{net}}\rho(u) + W_{\text{in}} r_{\text{in}}. \tag{78}$$

This fixed point equation can be numerically solved by time-discrete iteration process. But it can also be solved by a time-continuous process that underlies a neural or neuromorphic implementation. The prospective firing rate introduced in the NLA can be seen as a method to quickly find the fixed point in continuous time. When directly solving the implicit differential equation (as opposed to convert this into an explicit differential equation using e.g., the Cholesky decomposition), the fixed point is potentially found with a fewer number of steps.

To estimate the speed of convergence, we look at the initial speed when taking off at initial condition $u(0)$ between the unstable and stable fixed point. The initial speed for the classical differential equation, *Equation 77*, and the NLA version, *Equation 66*, are, respectively,

$$\dot{u}(0) = \frac{\Delta u(0)}{\tau},$$
(79a)

$$\dot{u}(0) = \frac{\Delta u(0)}{\tau \left(1 - W_{\text{net}} \rho'(u(0))\right)},$$
(79b)

where we set $\Delta u(0) = -u(0) + W_{\text{net}} \rho(u(0)) + W_{\text{in}} r_{\text{in}}(0)$. As $W_{\text{net}} \rho' > 0$, the initial convergence speed of the NLA solution is larger. The scheme has some resemblance to the Newton algorithm of finding zero's of a function by using its derivative.

## Appendix 6

## NLA for conductance-based neurons and least-action principle in physics

The mismatch energies and costs can be generalized in different ways. Here, we focus on a biophysical version of the mismatch energy that includes conductance-based neurons. This also relates to the least-action principle in physics. But the NLA can also be generalized to include other dynamica variables such as adaptive thresholds or synaptic short-term plasticity.

### Equivalent somato-dentritic circuit

For conductance based synapses, the excitatory and inhibitory conductances, $g_E$ and gI, are driven by the presynaptic firing rates and have the form $g_E(t) = W_E\, r(t)$, and analogously $g_I(t) = W_I\, r(t)$. The dynamics of a somatic voltage $u$ and a dendritic voltage $v$ reads as

$$c\,\dot{u} = g_L(E_L - u) + g_{sd}(v - u), \tag{80}$$

$$c_d\,\dot{v} = g_L(E_L - v) + g_E(E_E - v) + g_I(E_I - v) + g_{ds}(u - v) \tag{81}$$

where $c$ and $c_d$ are the somatic and dendritic capacitances, $E_{L/E/I}$ the reversal potentials for the leak, the excitatory and inhibitory currents, $g_{sd}$ the transfer conductance from the dendrite to the soma, and $g_{ds}$ in the other direction.

We consider the case when the dendritic capacitance $c_d$ is small as compared to the sum of conductances $g_d$ on the right-hand-side of *Equation 81*, yielding a fast dendritic time constant. In this case we can solve this equation in the steady state for $v$, plug this into *Equation 80*, and get

$$c\,\dot{u} = g\,(V - u), \tag{82}$$

with effective reversal potential $V = (g_L E_L + g_{sd}\frac{g_{ff}}{g_{ff}+g_{ds}} v_{ff})/g$, total conductance $g = g_L + g_{sd}\frac{g_{ff}}{g_{ff}+g_{ds}}$, feedforward dendritic voltage $v_{ff} = (g_L E_L + g_E E_E + g_I E_I)/g_{ff}$ and feedforward dendritic conductance $g_{ff} = g_L + g_E + g_I$. Because $g_{E/I} = W_{E/I}\, r(u_{pre}, \dot{u}_{pre})$, the conductance depends on the presynaptic voltage and its derivative. *Equation 82* describes the effective circuit that has the identical voltage time course as *Equations 80 and 81* with $\dot{v} = 0$, but with a single time-dependent 'battery voltage' $V$ and Ohmic resistance $R = 1/g$.

### Somato-dentritic mismatch power and action

The synaptic inputs $g_E(t)$ and $g_I(t)$ are continuously driving $V(t)$, and the best what one can hope for the dynamics of $u$ is that it traces $V$ with some integration delay determined by the time constant $\tau = c/g$. In fact, if $u$ follows the dynamics of *Equation 82*, then $u$ becomes the low-pass filtered target potential, $u = \bar{V}$, where $V(t)$ is filtered with the dynamic time constant $\tau(t)$. The defining equations for the low-pass filtering is

$$\bar{u} + \tau\dot{\bar{u}} = u, \tag{83}$$

and this self-consistency equation is equivalent to the explicit form

$$\bar{u}(t) = \int_{-\infty}^{t} dt'\, \frac{u(t')}{\tau(t')}\, e^{-\int_{t'}^{t} dt''\, \frac{1}{\tau(t'')}}. \tag{84}$$

To capture the voltage dynamics with our NLA principle we recall that the somatic voltage $u$ can be nudged by an 'apical voltage' $\bar{e}$ that causes a somatic voltage error $\bar{e} = u - \bar{V}$. The voltage error drives a current $I = g\bar{e}$ through the conductance $g$. The electrical power of this current $I$ driven by the voltage $\bar{e}$ is $P = I\bar{e}^2$. This motivates the definition of the mismatch power in a network of $N$ neurons by

$$\mathcal{P} = \sum_{i}^{N} \frac{g_i}{2}\,(u_i - \bar{V}_i)^2 = \frac{1}{2}\boldsymbol{g}^T(\boldsymbol{u} - \overline{\boldsymbol{V}})^2. \tag{85}$$

$\mathcal{P}$ is a virtual power that, nevertheless, is related to some physical flow of ions. Assume we could measure all the ions flowing in the original circuit of *Equations 80 and 81* (in the limit of small ratio $C_d/g_d$). From this flow, delete the ion flow that cancels at the level of electrical charge exchange due to the counter directed flow. The remaining effective ion flow defines an effective current flowing through the conductance $g$ with driving force $(V - u)$, *Equation 82*. If it were only this effective current in the reduced circuit, the voltage $u(t)$, starting at $u(0)$, would converge with time-constant $\tau$ to the low-pass filtering $\overline{V}$. Without additional 'hidden' current, the voltage $u$ would then instantaneously follow $\overline{V}$ that is itself given by the forward dendritic input conductances. The deviation of $u$ from $\overline{V}$, caused by some initial conditions in $u$ or by a feedback current from the network affecting $u$, builds the mismatch power $\mathcal{P}$. The feedback may originate from a target imposed downstream, and the neuron is 'free' in how to dynamically match $u$ and $\overline{V}$. It is, therefore, tempting to see $\mathcal{P}$ as a 'free power', and the NLA principle as minimizing the corresponding 'free energy'. In fact, the free-energy principle says that any self-organizing system that is in a dynamic equilibrium with its environment (here $u = \overline{V}$ in the absence of output nudging) must minimize its free energy (that here builds up by imposing a target; *Friston, 2010*).

The NLA principle states that the time-integral of $\mathcal{P}$ is minimized with respect to the look-ahead voltage $\tilde{u}$. We, therefore, define the physical mismatch energy as

$$A = \int_{t_1}^{t_2} \mathcal{P} \, \mathrm{dt} \, , \tag{86}$$

that has the units of energy. $\mathcal{E}_M$ takes the role of our neural action ($A$ in the main text) for conductance-based neurons.

## Euler-Lagrange equations for conductance-based neurons

The NLA for conductance-based neurons seeks to minimize $A = \int \mathcal{P}(u) \, \mathrm{dt}$ with respect to variations of $u$, such that $\frac{\delta A}{\delta u} = 0$. In the simplest example of the main text we considered prospective rates, $r = \rho(u) + \tau \dot{\rho}(u)$, so that the low-pass filtered rates become a function of the instantaneous voltage, $\bar{r} = \rho(u)$. These low-pass filtered presynaptic rates, $\bar{r}$, determine the postsynaptic voltage $u$. Analogously, the low-pass filtered reversal potential, $\overline{V}$, determines the postsynaptic voltage $u$, and we again postulate that $\overline{V}$ is an instantaneous function of the presynaptic voltage, $\overline{V} = \phi(u)$. Here, we argue that active dendritic mechanisms advance to postsynaptic reversal potential $V$, so that the delayed $\overline{V}$ again becomes instantaneous, similarly to the advancement of the apical dendritic potential observed in cortical pyramidal neurons (*Ulrich, 2002*), see also *Figure 2b*. With this instantaneity, the stationarity of the action with respect to generalized (compact and non-compact) variations, $\frac{\delta A}{\delta u} = 0$, translates to the condition $\frac{\partial \mathcal{P}}{\partial u} = 0$.

Calculating $\frac{\partial \mathcal{P}}{\partial u} = 0$ with $\mathcal{P}$ given in *Equation 85* and $\tau = c/g$ for the total conductance $g(u)$ specified after *Equation 82* is a bit more demanding. For a probabilistic version, where $\mathcal{P}$ is derived from the negative log-likelihood of a Gaussian density of the voltage, $\mathcal{P} = -\log p(u|\overline{V}) = \frac{1}{2}g^T(u - \overline{V})^2 - \frac{1}{2}\log g + \mathrm{const}$, the calculation is done in *Jordan et al., 2022*. In this probabilistic version, there is an additional normalization term that enters as $\log g$. In the deterministic version considered here, this log term is not present and we calculate

$$\frac{\partial \mathcal{P}}{\partial u} = g \cdot (u - \overline{V}) - \bar{\epsilon} \quad \text{with} \quad \bar{\epsilon} = \bar{r}' \cdot W_{\mathrm{net}}^T \left( (u - \overline{V})(E^{E/I} - \overline{V}) - \frac{1}{2}(u - \overline{V})^2 \right) . \tag{87}$$

Notice that the transpose $W_{\mathrm{net}}^T$ selects the downstream network neuron to backpropagate from there the first- and second-order errors. From $\frac{\partial \mathcal{P}}{\partial u} = 0$ and $\tau = c/g$ we conclude that

$$\frac{c}{\tau} \cdot (u - \overline{V}) - \bar{\epsilon} = 0 \quad \text{or} \quad u = \overline{V} + \frac{\tau}{c}\bar{\epsilon}. \tag{88}$$

We next apply the look ahead operator to the expression in this *Equation 88*. Assuming an initialization at $t_0 = -\infty$, the condition $\frac{\partial \mathcal{P}}{\partial u} = 0$ becomes equivalent to $\left(1 + \tau \frac{\mathrm{d}}{\mathrm{dt}}\right)\frac{\partial \mathcal{P}}{\partial u} = 0$, and hence *Equation 88* becomes equivalent to $g(u - \overline{V}) - \bar{\epsilon} + \tau g(\dot{u} - \dot{\overline{V}}) - \tau\dot{\bar{\epsilon}} + \tau\dot{g}(u - \overline{V}) = c\dot{u} - g(V - u) - \epsilon + \tau\dot{g}(u - \overline{V}) = 0$, and with the total conductance $g(u)$ specified after *Equation 82* this is calculated to become (see *Jordan et al., 2022*).

$$c\,\dot{\boldsymbol{u}} = \boldsymbol{g}(\boldsymbol{V} - \boldsymbol{u}) + \boldsymbol{\epsilon} + \dot{\tau}\bar{\boldsymbol{\epsilon}}. \tag{89}$$

A learning rule of the form $\dot{\boldsymbol{W}} \propto \frac{\partial \mathcal{P}}{\partial \boldsymbol{W}} = \bar{\boldsymbol{e}}_{\text{post}}\,\bar{\boldsymbol{r}}^{\text{T}}$ with an appropriate postsynaptic error $\bar{\boldsymbol{e}}_{\text{post}}$ can again be derived (see *Jordan et al., 2022*) for a single neuron in the probabilistic framework.

## Generalizations: Long memories, reinforcement learning

One could also extend the NLA principle by adding e.g., threshold adaptation that endows the dynamics with additional and longer time constants. For this, the rate function is parametrized by an additional threshold, $\bar{r} = \rho(u - \vartheta)$, an the Lagrangian is added by an error term on the threshold. Such an error addition can take the form $L = ... + \frac{1}{2}\|\vartheta - \bar{r}^{\tau_\vartheta}\|^2$, where $\bar{r}^{\tau_\vartheta}$ now represents a low-pass filtering of the rate with a long threshold adaptation time constant $\tau_\vartheta$. Neurons that show an additional threshold adaptation will still be able to instantaneously transmit a voltage jump through a prospective firing rate, but this will now also depend on the neuron's history. Short-term plasticity may be included in the same way. Due to the history dependence, the stationarity of the action $\delta A = 0$ cannot anymore be reduced to the stationarity of the Lagrangian at any moment in time. As a consequence, the errors will look-ahead into to future more than only based on the local derivatives.

Generalizations are further possible for the cost function that favor voltage regions with high cost, say corresponding to punishment, or negative cost, corresponding to punishment to reward. These extensions will be considered in future work.

## Voltage dynamics from the least-action principle in physics

We have shown that through the look-ahead in Hamilton's least-action principle, the notion of friction enters through the backdoor. In the least-action formalism in physics, friction is directly introduced by extending the Hamiltonian principle to a generalized D'Alembert principle, where at the level of the Euler-Lagrange equations the generalized force is equated to the dissipation force (*Flannery, 2005*). For an introduction to the least-action principle in physics see e.g., (*Feynman et al., 2011*), and for an introduction to the calculus of variation in general with a derivation of the Euler-Lagrange equation see e.g., (*Chachuat, 2007*).

The electro-chemical properties of a membrane can be captured by an equivalent circuit consisting of a battery voltage $V$, a capacitance $C$ and a resistance $R$, arranged in parallel. The voltage dynamics is derived from the Euler-Lagrange equations that are added by a dissipative force.

Formally, in the absence of an inductance defining the kinetic energy, the Lagrangian $\mathcal{L}$ becomes identical to the potential energy $\mathcal{U}$ that itself is

$$\mathcal{L} = \mathcal{U} = QV - Q^2/(2C), \tag{90}$$

where $Q$ represents the charge across the membrane. Looking for stationary trajectory of the action (time integral of the Lagrangian) with only the potential energy in the Lagrangian would not give any dynamics. In fact, the Euler-Lagrange equation $\partial_Q \mathcal{L} - d_t \partial_{\dot{Q}} \mathcal{L} = 0$ yields $Q = CV$. The potential energy is, therefore, extended by the dissipative Rayleigh energy $\mathcal{F}$ that introduces friction,

$$\mathcal{F} = R\dot{Q}^2/2. \tag{91}$$

According to D'Alembert's principle, the Rayleigh energy enters as a gradient at the level of the Euler-Lagrange equation. At this point, the theory steps out from the original principle by adding the Rayleigh gradient 'by hand' to the Euler-Lagrange equation (different from the NLA, see below). Otherwise, no first order differential equation with friction in the desired form would be obtained from the least-action principle in physics (if we were adding $\mathcal{F}$ to $\mathcal{L}$, for instance, the $d_t \partial_{\dot{Q}} \mathcal{L}$-term in the Euler-Lagrange equations would yield the term $R\ddot{Q}$ instead of $R\dot{Q}$). Hence, with D'Alembert's patch the dynamics is characterized by the Euler-Lagrange equation added by a dissipative force,

$$\partial_Q \mathcal{L} - d_t \partial_{\dot{Q}} \mathcal{L} + \partial_{\dot{Q}} \mathcal{F} = 0,$$

and this equation reduces to $-V + Q/C + R\dot{Q} = 0$. Identifying the charge by means of voltage across the capacitance, $Q = Cu$, this equation can also be written as $-V + u + RC\dot{u} = 0$, or

$$\tau \dot{u} = -u + V, \tag{92}$$

with $\tau = RC$, similarly as derived in *Equation 89*.

The link to our NLA principle is most easily made by assuming that the 'membrane' conductance $g = 1/R$ does not depend on the voltage $u$ (of the post- or presynaptic neuron, as this is the case in general, see explanation after *Equation 82*). In this simplified case, the Lagrangian from *Equation 86* becomes the power

$$\mathcal{P} = (u - \overline{V})^2/(2R). \tag{93}$$

The Euler-Lagrange equation with respect to $\tilde{u}$, applied to the action $A = \int \mathcal{P}\, \mathrm{d}t$ is then calculated by

$$\frac{\partial \mathcal{P}}{\partial \tilde{u}} - \frac{\mathrm{d}}{\mathrm{d}t}\frac{\partial \mathcal{P}}{\partial \dot{\tilde{u}}} = \left(1 + \tau\frac{\mathrm{d}}{\mathrm{d}t}\right)\frac{\partial \mathcal{P}}{\partial u} = \frac{1}{R}\left(u + \tau\dot{u} - (\overline{V} + \tau\dot{\overline{V}})\right) = 0, \text{ or} \tag{94}$$

$$\tau\dot{u} = -u + V. \tag{95}$$

We used that $\tau = RC$ while $u = \tilde{u} - \tau\dot{\tilde{u}}$ and $V = \overline{V} + \tau\dot{\overline{V}}$.

The dynamics derived from the NLA *Equation 95* is identical to the dynamics derived from the least-action principle with friction in physics (*Equation 92*). Hence, the minimization of the energy (i.e. the time-integral of the power, $A = \int \mathcal{P}\, \mathrm{d}t$) by looking ahead in time is equivalent to the stationarity of the physical action without looking ahead, but by taking account of the Rayleigh dissipation. The trick of looking ahead in time generates a dynamics out of an action $A$ that encompasses only a potential energy, no kinetic energy. Without looking ahead, no dynamics would emerge from such an action. Notice, though, that the NLA action has units of energy (the time-integral of a power), while the physical action has units of energy times time (the time-integral of energies).

# Appendix 7

## A tutorial on total and partial derivatives as used in the paper

The proof of Theorem 1 given in the Methods (Sect. Proving theorem 1 (rt-DeEP)) makes use of partial and total derivatives and follows the notation of *Scellier and Bengio, 2017* and *Meulemans et al., 2022*. As there is some variability (and community-dependent sloppiness) in the notation of partial and total derivatives here and in general, we provide some explanations on how this notation is interpreted, and why, for instance, total derivatives commute with each other, and also partial with each other (although not total with partial). For a standard basic textbook introducing the concepts of partial derivatives, total differentials, the implicit function theorem etc., see *Stewart, 2010*, being historically based on the classics of *Courant, 1934*.

i. In a differential geometric setting, the derivative of real-valued function $E(\boldsymbol{u})$ on a point $\boldsymbol{u}$ of a manifold (like the flat Euclidean space) is considered as a mapping of tangent vectors at $u$ to the real numbers. When interpreting the derivative as mapping, the $\frac{\partial E}{\partial \boldsymbol{u}}$ is living in the dual space of $u$ and is, therefore, a row vector if is a column vector. If a function $\boldsymbol{f}(\boldsymbol{u})$ of $\boldsymbol{u}$ is a vector valued, then its derivative is a matrix with entries $\left(\frac{\partial \boldsymbol{f}}{\partial \boldsymbol{u}}\right)_{ij} = \frac{\partial f_i}{\partial u_j}$, where $i$ is indexing the rows (i.e. running down) and $j$ is indexing the columns (i.e. running right). When $\boldsymbol{r} = \rho(\boldsymbol{u})$ is a column vector with $\rho$ applied to each component of $\boldsymbol{u}$, we consider the (partial) derivative $\boldsymbol{r}' = \rho'(\boldsymbol{u})$ for convenience as column vector with components $\rho'(u_i)$. To strictly follow the formalism, it should be a diagonal matrix.

ii. Because we introduced the error vector $\bar{\boldsymbol{e}}$ as column vector, it is easier to write $\frac{\partial L}{\partial \boldsymbol{u}} = \bar{\boldsymbol{e}} + ...$, where now $\frac{\partial L}{\partial \boldsymbol{u}}$ is also considered as column vector. To be consistent with the above, we should have written $\frac{\partial L}{\partial \boldsymbol{u}}^{\mathrm{T}} = \bar{\boldsymbol{e}} + ....$ The $.^{\mathrm{T}}$ appeared us as too heavy so that we neglected it, where it did not have further mathematical consequences (hoping it does not cause confusions). Sticking here to a column vector also renders the backpropagation error to be a column vector in *Equation 21*. This error then gets the classical form with the weight transpose, $\bar{\boldsymbol{\epsilon}} = \bar{\boldsymbol{r}}'_{\mathrm{net}} \boldsymbol{W}^{\mathrm{T}}_{\mathrm{net}} \bar{\boldsymbol{e}}$.

iii. We typically have real-valued functions of the form $E(\boldsymbol{u_\theta}, \boldsymbol{\theta})$, with $\boldsymbol{\theta} = (\boldsymbol{W}, \beta)$ being a vector of parameters, and $\boldsymbol{u}$ being a function of $\boldsymbol{\theta}$. To get the total derivative of $E$ with respect to $\boldsymbol{\theta}$ we consider the values $E$ as a function of $\boldsymbol{\theta}$. This can be done by introducing a new function $\mathcal{E}(\boldsymbol{\theta})$ defined as $\mathcal{E}(\boldsymbol{\theta}) = E(\boldsymbol{u_\theta}, \boldsymbol{\theta})$, where the components $\theta_i$ are considered as independent variables. The total derivative of $E$ with respect to $\boldsymbol{\theta}$ is then defined as vector-valued function $\frac{\mathrm{d}E}{\mathrm{d}\boldsymbol{\theta}} = \left(\frac{\mathrm{d}E}{\mathrm{d}\theta_1}, ..., \frac{\mathrm{d}E}{\mathrm{d}\theta_n}\right) = \left(\frac{\mathrm{d}\mathcal{E}}{\mathrm{d}\theta_1}, ..., \frac{\mathrm{d}\mathcal{E}}{\mathrm{d}\theta_n}\right) = \frac{\partial \mathcal{E}}{\partial \boldsymbol{\theta}}$ for a $n$-dimensional $\boldsymbol{\theta}$. It can be helpful to think of the components of this total derivative as a (total) directional derivatives in the unit directions. For the last ($n$-th) unit direction $\Delta \boldsymbol{\theta}^{(n)} = (0, .., 0, 1)$, for instance, we have $\frac{\mathrm{d}E}{\mathrm{d}\theta_n} = \lim_{\varepsilon \to 0} \frac{1}{\varepsilon} \left( E(\boldsymbol{u_{\theta + \varepsilon \Delta \theta^{(n)}}}, \boldsymbol{\theta} + \varepsilon \Delta \boldsymbol{\theta}^{(n)}) - E(\boldsymbol{u_\theta}, \boldsymbol{\theta}) \right)$.

iv. The cost gradient, $\frac{\mathrm{d}C}{\mathrm{d}\boldsymbol{W}} \propto (\boldsymbol{u} - \boldsymbol{W}\bar{\boldsymbol{r}})\bar{\boldsymbol{r}}^T$ has the same dimension as $\boldsymbol{W}$. Recall that by the cost gradient we mean $\frac{\mathrm{d}\mathcal{C}}{\mathrm{d}\boldsymbol{W}} = \frac{\partial \mathcal{C}}{\partial \boldsymbol{W}}$, where $\mathcal{C}$ is defined as $\mathcal{C}(\boldsymbol{W}) = C(\boldsymbol{u_o}(\boldsymbol{W}), \boldsymbol{u_o^*})$, with the voltage $\boldsymbol{u_o}$ of the output neurons being itself a function of $\boldsymbol{W}$.

v. To calculate the partial derivative $\frac{\partial}{\partial \boldsymbol{\theta}} E(\boldsymbol{u}, \boldsymbol{\theta})$ with respect to $\boldsymbol{\theta}$, we fix the first argument $\boldsymbol{u_\theta}$, even if for $\boldsymbol{u}$ we often plugged in the components of the trajectory $\boldsymbol{u} = \boldsymbol{u_\theta}(t)$ that now does depend on $\boldsymbol{\theta}$. In contrast, the total derivative is $\frac{\mathrm{d}}{\mathrm{d}\boldsymbol{\theta}} E(\boldsymbol{u}, \boldsymbol{\theta}) = \frac{\partial E(\boldsymbol{u}, \boldsymbol{\theta})}{\partial \boldsymbol{u}} \frac{\mathrm{d}\boldsymbol{u}}{\mathrm{d}\boldsymbol{\theta}} + \frac{\partial E(\boldsymbol{u}, \boldsymbol{\theta})}{\partial \boldsymbol{\theta}}$. Here, $\frac{\partial E(\boldsymbol{u}, \boldsymbol{\theta})}{\partial \boldsymbol{u}}$ is a row vector, as also $\frac{\partial E(\boldsymbol{u}, \boldsymbol{\theta})}{\partial \boldsymbol{\theta}}$, consistent with the convention (that we have broken in *Equation 28* to keep vectors as columns). When $\boldsymbol{u}$ is considered as trajectory, $\frac{\mathrm{d}\boldsymbol{u}}{\mathrm{d}\boldsymbol{\theta}}$ does not vanish in general, but it does when $\boldsymbol{u}$ is simply considered as independent variable.

vi. When replacing the argument $\boldsymbol{u}$ in $E(\boldsymbol{u}, \boldsymbol{\theta})$ by $\boldsymbol{u} = \tilde{\boldsymbol{u}} - \tau \dot{\tilde{\boldsymbol{u}}}$ we get the 'Lagrangian' $L(\tilde{\boldsymbol{u}}, \dot{\tilde{\boldsymbol{u}}}, \theta) = E(\tilde{\boldsymbol{u}} - \tau \dot{\tilde{\boldsymbol{u}}}, \theta)$. The partial derivative of $L$ with respect to $\tilde{\boldsymbol{u}}$, for instance, is then $\frac{\partial}{\partial \tilde{\boldsymbol{u}}} L(\tilde{\boldsymbol{u}}, \dot{\tilde{\boldsymbol{u}}}, \theta) = \frac{\partial}{\partial \boldsymbol{u}} E(\boldsymbol{u}, \boldsymbol{\theta}) \frac{\partial \boldsymbol{u}}{\partial \tilde{\boldsymbol{u}}} = \frac{\partial}{\partial \boldsymbol{u}} E(\boldsymbol{u}, \boldsymbol{\theta})$. The partial derivative $\frac{\partial}{\partial \tilde{\boldsymbol{u}}} L$ considers $L$ as a function of independent arguments $\tilde{\boldsymbol{u}}, \dot{\tilde{\boldsymbol{u}}}$ and $\boldsymbol{\theta}$.

vii. We also used that the total derivatives commute, in the current example $\frac{\mathrm{d}}{\mathrm{d}\boldsymbol{W}} \frac{\mathrm{d}}{\mathrm{d}\beta} E = \frac{\mathrm{d}}{\mathrm{d}\beta} \frac{\mathrm{d}}{\mathrm{d}\boldsymbol{W}} E$. This is generally true for derivatives of Lipschitz continuous functions, for which derivatives exist almost everywhere. The total derivatives (where they exist) then commute because the difference quotients in $\boldsymbol{W}$ and $\beta$ are uniformly bounded. The Moore-Osgood theorem tells that two limits, of which at least one is uniform in the other, can be commuted. This also applies to the double difference quotients involved in the definition of $\frac{\mathrm{d}}{\mathrm{d}\boldsymbol{W}} \frac{\mathrm{d}}{\mathrm{d}\beta}$. Remember

that the total derivative, for instance with respect to the $n$-th parameter, can be written as $\frac{\mathrm{d}E}{\mathrm{d}\theta_n} = \lim_{\varepsilon \to 0} \frac{1}{\varepsilon} \left( E(\boldsymbol{u}_{\boldsymbol{\theta}+\varepsilon\Delta\boldsymbol{\theta}^{(n)}}, \boldsymbol{\theta}+\varepsilon\Delta\boldsymbol{\theta}^{(n)}) - E(\boldsymbol{u}_{\boldsymbol{\theta}}, \boldsymbol{\theta}) \right)$. But note again that, while for Lipschitz functions partial derivatives commute among themselves, and total derivatives commute among themselves, partial and total derivatives in general do not commute (not even with additional smoothness conditions)!

# Appendix 8

## Proof of theorem 1, part (*ii*), using only partial derivatives

This section proves *Equations 29–31* in terms of only partial derivatives, banning nested functions and banning the main theorem to calculate total derivatives in higher dimensions as presented in item (*v*) of the above Tutorial. While a reduction to partial derivatives only may represent a conceptual simplification, it requires many more additional (unnested) functions to be defined (and in fact being the reason why in mathematics the concept of a total differential / total derivative is introduced, see e.g. *Courant, 1934*). – The cited three equations are also the core for the proof for Equilibrium Propagation (*Scellier and Bengio, 2017*), although there only applied in the steady state after the network converged to a constant activity.

We assume a network of $d$ neurons whose membrane potential is given by $\boldsymbol{u} \in \mathbb{R}^d$ and which are connected via weights $\boldsymbol{W} \in \mathbb{R}^{d \times d}$. By $\boldsymbol{\nabla}_{\boldsymbol{u}}$, we denote the gradient with respect to the membrane potentials, i.e., $\boldsymbol{\nabla}_{\boldsymbol{u}} = (\frac{\partial}{\partial u_1}, \dots, \frac{\partial}{\partial u_d})$. Similarly, $\boldsymbol{\nabla}_{\boldsymbol{W}}$ is a matrix containing the derivatives with respect to the weights, $(\boldsymbol{\nabla}_{\boldsymbol{W}})_{ij} = \frac{\partial}{\partial W_{ij}}$.

To prove the rt-DeEP theorem 1, we first have to make a few definitions and observations:

1. For a given $(\boldsymbol{W}, \beta)$, the dynamics yield certain membrane potentials $f_{\boldsymbol{u}} \in \mathbb{R}^d$. Formally, we define this as

$$f_{\boldsymbol{u}} : \mathbb{R}^{d \times d} \times \mathbb{R} \to \mathbb{R}^d, \quad (\boldsymbol{W}, \beta) \mapsto f_{\boldsymbol{u}}(\boldsymbol{W}, \beta). \tag{96}$$

   The $i$th element of $f_{\boldsymbol{u}}$ is denoted by $f_{u_i}$ and hence $\boldsymbol{\nabla}_{f_{\boldsymbol{u}}} = (\frac{\partial}{\partial f_{u_1}}, \dots, \frac{\partial}{\partial f_{u_d}})$.

2. We define the following functions:
   - the mismatch energy $E^{\mathrm{M}} : \mathbb{R}^d \times \mathbb{R}^{d \times d} \to \mathbb{R}$, $(\boldsymbol{u}, \boldsymbol{W}) \mapsto E^{\mathrm{M}}(\boldsymbol{u}, \boldsymbol{W}) = \frac{1}{2} \sum_{i=1}^d \left[ u_i - \sum_j W_{ij} \bar{r}_j \right]^2$,
   - the cost function $C : \mathbb{R}^d \to \mathbb{R}$, $\boldsymbol{u} \mapsto C(\boldsymbol{u}) = \frac{1}{2} \sum_{k \in \mathcal{O}} \|u_k^* - u_k\|^2$,
   - the Lagrangian $L : \mathbb{R}^d \times \mathbb{R}^{d \times d} \times \mathbb{R} \to \mathbb{R}$, $(\boldsymbol{u}, \boldsymbol{W}, \beta) \mapsto L(\boldsymbol{u}, \boldsymbol{W}, \beta) = E^{\mathrm{M}}(\boldsymbol{u}, \boldsymbol{W}) + \beta C(\boldsymbol{u})$. To make the dependency of the cost and energies on $\beta$ and $\boldsymbol{W}$ explicit, we further introduce three auxiliary functions $F_{\mathrm{M}}$, $F_{\mathrm{C}}$ and $F_{\mathrm{L}}$ :
   - for the mismatch energy $F_{\mathrm{M}} : \mathbb{R}^{d \times d} \times \mathbb{R} \to \mathbb{R}$, $(\boldsymbol{W}, \beta) \mapsto F_{\mathrm{M}}(\boldsymbol{W}, \beta) = E^{\mathrm{M}} \big( f_{\boldsymbol{u}}(\boldsymbol{W}, \beta), \boldsymbol{W} \big)$,
   - for the cost function $F_C : \mathbb{R}^{d \times d} \times \mathbb{R} \to \mathbb{R}$, $(\boldsymbol{W}, \beta) \mapsto F_C(\boldsymbol{W}, \beta) = C \big( f_{\boldsymbol{u}}(\boldsymbol{W}, \beta) \big)$,
   - for the Lagrangian $F_L : \mathbb{R}^{d \times d} \times \mathbb{R} \to \mathbb{R}$, $(\boldsymbol{W}, \beta) \mapsto F_L(\boldsymbol{W}, \beta) = F_{\mathrm{M}}(\boldsymbol{W}, \beta) + \beta F_C(\boldsymbol{W}, \beta)$

3. The Euler-Lagrange equations can be written as $\tau \frac{\mathrm{d}}{\mathrm{d}t} \boldsymbol{\nabla}_{\boldsymbol{u}} L = -\boldsymbol{\nabla}_{\boldsymbol{u}} L$.
   Hence, far enough away from initialization and for smooth enough input (and targets) we have $\boldsymbol{\nabla}_{f_{\boldsymbol{u}}} L = 0$ at all times, even when changing the network input continuously. Note that both the cost and the mismatch energies are defined on low-pass-filtered signals, and it is with respect to the low-pass filtered external input that the low-pass-filtered output error is minimized.

4. Without output nudging (i.e. $\beta = 0$), the output error vanishes and consequently all other prediction errors vanish as well, $\bar{\boldsymbol{e}} = \boldsymbol{u} - \boldsymbol{W}\bar{\boldsymbol{r}} = 0$. This can be easily shown for layered network architectures and holds true for arbitrary connections (e.g. recurrent networks) as long as $f_{\boldsymbol{u}}$ uniquely exists, i.e., $\boldsymbol{u} = \boldsymbol{W}\bar{\boldsymbol{r}}$ has a unique solution for $\boldsymbol{u}$. From the form of the mismatch energy, we then get

$$\boldsymbol{\nabla}_{\boldsymbol{W}} E^{\mathrm{M}} \big|_{\beta=0} = \left( \boldsymbol{W}\bar{\boldsymbol{r}} - \boldsymbol{u} \right) \bar{\boldsymbol{r}}^{\mathrm{T}} \big|_{\beta=0} = 0. \tag{97}$$

   Since we are assuming smooth functions, this also implies that

$$\lim_{\epsilon_\beta \to 0} \boldsymbol{\nabla}_{\boldsymbol{W}} E^{\mathrm{M}} \big|_{\beta=\epsilon_\beta} = 0. \tag{98}$$

5. From the assumption of well-behaved (smooth) functions, it also follows that partial derivatives commute $\boldsymbol{\nabla}_{\boldsymbol{W}} \frac{\partial}{\partial \beta} = \frac{\partial}{\partial \beta} \boldsymbol{\nabla}_{\boldsymbol{W}}$.

Our goal is to find a plasticity rule that minimizes the cost $C$, which we do by calculating $\boldsymbol{\nabla}_{\boldsymbol{W}} F_C \big|_{\beta=0}$. Similar to *Scellier and Bengio, 2017*, to achieve this, we first calculate the partial derivatives of $F_L$ with respect to the nudging strength $\beta$

$$\frac{\partial F_L}{\partial \beta} = \frac{\partial F_{\mathrm{M}}}{\partial \beta} + \beta \frac{\partial F_C}{\partial \beta} + F_C \tag{99a}$$

$$= \sum_{i=1}^{d} \left( \frac{\partial E^{\mathrm{M}}}{\partial f_{u_i}} \frac{\partial f_{u_i}}{\partial \beta} + \beta \frac{\partial C}{\partial f_{u_i}} \frac{\partial f_{u_i}}{\partial \beta} \right) + C \tag{99b}$$

$$= \sum_{i=1}^{d} \underbrace{\frac{\partial \left( E^{\mathrm{M}} + \beta C \right)}{\partial f_{u_i}}}_{=0} \frac{\partial f_{u_i}}{\partial \beta} + C \tag{99c}$$

$$= C, \tag{99d}$$

and the weights $W$

$$\frac{\partial F_L}{\partial W_{ij}} = \frac{\partial F_{\mathrm{M}}}{\partial W_{ij}} + \beta \frac{\partial F_C}{\partial W_{ij}} \tag{100a}$$

$$\sum_{k=1}^{d} \left( \frac{\partial E^{\mathrm{M}}}{\partial f_{u_k}} \frac{\partial f_{u_k}}{\partial W_{ij}} + \beta \frac{\partial C}{\partial f_{u_k}} \frac{\partial f_{u_k}}{\partial W_{ij}} \right) + \frac{\partial E^{\mathrm{M}}}{\partial W_{ij}} \tag{100b}$$

$$= \sum_{k=1}^{d} \underbrace{\frac{\partial \left( E^{\mathrm{M}} + \beta C \right)}{\partial f_{u_k}}}_{=0} \frac{\partial f_{u_k}}{\partial W_{ij}} + \frac{\partial E^{\mathrm{M}}}{\partial W_{ij}} \tag{100c}$$

$$= \frac{\partial E^{\mathrm{M}}}{\partial W_{ij}}, \tag{100d}$$

or in vectorized form

$$\boldsymbol{\nabla}_{\boldsymbol{W}} F_L = \boldsymbol{\nabla}_{\boldsymbol{W}} E^{\mathrm{M}}. \tag{101}$$

With these identities in place, we can calculate the plasticity rule:

$$- \lim_{\epsilon_\beta \to 0} \boldsymbol{\nabla}_{\boldsymbol{W}} F_C = \lim_{\epsilon_\beta \to 0} \boldsymbol{\nabla}_{\boldsymbol{W}} \frac{\partial F_L}{\partial \beta} \tag{102a}$$

$$\lim_{\epsilon_\beta \to 0} \frac{\partial}{\partial \beta} \boldsymbol{\nabla}_{\boldsymbol{W}} F_L \tag{102b}$$

$$= \lim_{\epsilon_\beta \to 0} \lim_{\delta \to 0} \frac{1}{\delta} \left( \boldsymbol{\nabla}_{\boldsymbol{W}} F_L \big|_{\beta = \epsilon_\beta + \delta} - \boldsymbol{\nabla}_{\boldsymbol{W}} F_L \big|_{\beta = \epsilon_\beta} \right) \tag{102c}$$

$$= \lim_{\delta \to 0} \frac{1}{\delta} \lim_{\epsilon_\beta \to 0} \left( \boldsymbol{\nabla}_{\boldsymbol{W}} E^{\mathrm{M}} \big|_{\beta = \epsilon_\beta + \delta} - \boldsymbol{\nabla}_{\boldsymbol{W}} E^{\mathrm{M}} \big|_{\beta = \epsilon_\beta} \right) \tag{102d}$$

$$= \lim_{\delta \to 0} \frac{1}{\delta} \left( \lim_{\epsilon_\beta \to 0} \boldsymbol{\nabla}_{\boldsymbol{W}} E^{\mathrm{M}} \big|_{\beta = \epsilon_\beta + \delta} - \underbrace{\lim_{\epsilon_\beta \to 0} \boldsymbol{\nabla}_{\boldsymbol{W}} E^{\mathrm{M}} \big|_{\beta = \epsilon_\beta}}_{=0 \text{ from (equation 98)}} \right) \tag{102e}$$

$$= \lim_{\delta \to 0} \frac{1}{\delta} \boldsymbol{\nabla}_{\boldsymbol{W}} E^{\mathrm{M}} \big|_{\beta = \delta} \tag{102f}$$

$$\approx \frac{1}{\delta} \boldsymbol{\nabla}_{\boldsymbol{W}} E^{\mathrm{M}} \big|_{\beta = \delta} \quad \text{for } \delta \ll 1. \tag{103g}$$

where we used that limits can be exchanged for smooth functions. Using the definition of $E^{\mathrm{M}}$, we obtain a plasticity rule that minimizes the cost function

$$-\lim_{\epsilon_\beta \to 0} \boldsymbol{\nabla}_{\boldsymbol{W}} C\big|_{\beta=\epsilon_\beta} \approx -\frac{1}{\epsilon_\beta} \boldsymbol{\nabla}_{\boldsymbol{W}} E^{\mathrm{M}}\big|_{\beta=\epsilon_\beta} = \frac{1}{\epsilon_\beta}\left(\boldsymbol{u} - \boldsymbol{W}\bar{\boldsymbol{r}}\right)\bar{\boldsymbol{r}}^{\mathrm{T}}\big|_{\beta=\epsilon_\beta} \quad \text{for } \epsilon_\beta \ll 1. \tag{103}$$

In practice, the prefactor $\frac{1}{\epsilon_\beta}$ is absorbed into the learning rate.

