## [Editor Report · eLife assessment]

This manuscript describes a potentially **important** theoretical framework to link predictive coding, error-based learning, and neuronal dynamics. The provided evidence is **solid**, but some details would benefit from additional clarification. The exposition of the manuscript is targeted for a specialist audience.

---

## [Referee Report · Reviewer #1 (Public Review)]

The manuscript considers a hierarchical network of neurons, of the type that can be found in sensory cortex, and assumes that they aim to constantly predict sensory inputs that may change in time. The paper describes the dynamics of neurons and rules of synaptic plasticity that minimize the integral of prediction errors over time.

The manuscript describes and analyses the model in great detail, and presents multiple and diverse simulations illustrating the model's functioning. However, the manuscript could be made more accessible and easier to read. The paper may help to understand the organization of cortical neurons, their properties, as well as the function of its particular components (such as apical dendrites).

---

## [Referee Report · Reviewer #2 (Public Review)]

Neuroscientists often state that we have no theory of the brain. The example of theoretical physics is often cited, where numerous and quite complex phenomena are explained by a compact mathematical description. Lagrangian and Hamiltonian pictures provide such powerful 'single equation'. These frameworks are referred to as 'energy', an elegant way to turn numerous differential equations into a single compact relationship between observable quantities (state variables like position and speed) and scaling constants (like the gravity constant or the Planck constant). Such energy pictures have been used in theoretical neuroscience since the 1980s.

The manuscript "neuronal least-action principle for real-time learning in cortical circuits" by Walter Senn and collaborators describes a theoretical framework to link predictive coding, error-based learning, and neuronal dynamics. The central concept is that an energy function combining self-supervised and supervised objectives is optimized by realistic neuronal dynamics and learning rules when considering the state of a neuron as a mixture of the current membrane potential and its rate of change. As compared with previous energy functions in theoretical neuroscience, this theory captures a more extensive range of observations while satisfying normative constraints. Particularly, no theory had to my knowledge related adaptive dynamics widely observed in the brain (referred to as prospective coding in the text, but is sometimes referred to as adaptive coding or redundancy reduction) with the dynamics of learning rules.

The manuscript first exposes the theory of two previously published papers by the same group on somato-dendritic error with apical and basal dendrites. These dynamics are then related to an energy function, whose optimum recovers the dynamics. The rest of the manuscript illustrates how features of this model fits either normative or observational constraints. Learning follows a combination of self-supervised learning (learning to predict the next step) and supervised learning (learning to predict an external signal). The credit assignment problem is solved by an apical-compartment projecting set of interneurons with learning rules whose role is to align many weight matrices to avoid having to do multiplexing. An extensive method section and supplementary material expand on mathematical proofs and make more explicit the mathematical relationship between different frameworks.

Experts would say that much of the article agglomerates previous theoretical papers by the same authors that have been published recently either in archival servers or in conference proceedings. A number of adaptations to previous theoretical results were necessary, so the present article is not easily reduced to a compendium of previous pre-prints. However, the manuscript is by no means easy to read. Also, there remain a few thorny assumptions (unobserved details of the learning rules or soma-dendrites interactions), but the theory is likely going to be regarded as an important step towards a comprehensive theory of the brain.

---

## [Author Response]

The following is the authors’ response to the original reviews.

**Reviewer #1 (Recommendations For The Authors):**
Major comments:(1) It is nice that the authors compared their model to the one "without lookahead" in Figure 4, but this comparison requires more evidence in my opinion, as I explain in this comment. The model without lookahead is closely related or possibly equivalent to the standard predictive coding. In predictive coding, one can make the network follow the stimulus rapidly by reducing the time constant tau. However, as the time constant decreases, the network would become unstable both in simulations (due to limited integration time step) and physical implementation (due to noise). Therefore I wonder if the proposed model has an advantage over standard predictive coding with an optimized time constant. Hence I suggest to also add a comparison between the proposed model, and the predictive coding with parameters (such as tau) optimized independently for each model. Of course, we know that the time-constant of biological neurons is fixed, but biological neurons might have had different time constants (by changing leak conductance) and such analysis could shed light on the question of why the neurons are organized the way they are.

The comparison with a predictive network for which the neuronal time constants shrink towards 0 is in fact helpful. We added two news subsections in the SI that formally compares the NLA with other approaches, Equilibrium propagation and the Latent Equilibrium, with a version of Equilibrium Propagation also covering the standard predictive coding you describe (SI, Sect.C and D). The Subsection C concludes: “In the Equilibrium propagation we cannot simply take the limit t0 since then the dynamics either disappears (when tau remains on the left, t Du0) or explodes (when t is moved to the right, dt/ t  ∞), leading to either too small or too big jumps.”

We have also expanded the passage on the predictive coding in the main text, comparing our instantaneous network processing (up to a remaining time constant tin) with experimental data from humans (see page 10 of the revised ms). The new paragraph ends with:

“Notice that, from a technical perspective, making the time constants of individual cortical neurons arbitrarily short leads to network instabilities and is unlikely the option chosen by the brain (see SI Sect. C, Comparison to the Equilibrium Propagation).”

A new formal definition of the moving equilibrium in the Methods (Sect. F) helps to understand this notion of being in a balanced equilibrium state during the dynamics. This formal definition directly leads to the contraction analysis in the SI, Sect. D, showing why the Latent Equilibrium is always contractive, while the current form of the NLA may show jumps at the corner of a ReLu (since a second order derivative of the transfer function enters in the error propagation).

The reviewer perhaps has additional simulations in mind that compare the robustness of the different models. However, as this paper is more about presenting a novel concept with a comprehensive theory (summing up to 45 pages), we prefer to not add more than the simulations necessary to check the statements of the theorems.

(2) I found this paper difficult to follow, because the Results sections went straight into details, and various elements of the model were introduced without explaining why they are necessary. Furthermore, the neural implementation was introduced after the model simulations. I suggest reorganizing the manuscript, to describe the model following Marr's levels of description and then presenting the results of simulations. In particular, I suggest starting the Results section by explaining what computation the network is trying to achieve (describe the setup, function L, define its integral over time, and explain that the goal is to find a model minimizing this integral). Then, I suggest presenting the algorithm the neurons need to employ to minimize this integral, i.e. their dynamics and plasticity (I wonder if r=rho(u) + tau rho(u)' is a consequence of action minimization or a necessary assumption - please clarify it). Next please explain how the algorithms could be implemented in biological neurons. Afterward please present the results of the simulation.

We are sorry to realize that we could not convey the main message clearly enough. After rewriting the paper and straightening the narrative, we hope it is simpler to understand now.

The paper does not suggest a new model to solve a task, and writing down the function to be minimized is not enough. The point of the NLA is that the time integral of our Lagrangian is minimized with respect to the prospective coordinates, i.e. the discounted future voltage. It is about the question how dynamic equations in biology are derived. Of course, we also solve these equations, prove theorems and perform simulations. But the main point that biology seems to deal with time differently than physics deals with time. Biology “thinks” in terms of future quantities, physics “thinks” in terms of current quantities. We tried to explain this better now in the Introduction, the Results (e.g. after Eq. 5) and the Methods.

(3) Understanding the paper requires background knowledge that most readers of eLife are unlikely to have, even if they are mathematically minded. For example, I am from the field of computational neuroscience, and I have never heard about Least Action principle from physics or the EulerLagrange equation. I felt lost after reading this paper, and to be able to write this review I needed to watch videos on the Euler-Lagrange equation. To help other readers, I have two suggestions: First, I feel that Eq 4-6 could be moved to the methods, because I found the concept of u~ difficult to understand, and it does not appear in the algorithm. Second, I advise to write in the Introduction, what knowledge is required to follow this paper, and point the readers to resources where they can find the required information. The authors may specify what background is required to follow the main text, and what is required to understand the methods.

We hope that after explaining the rationale better, it becomes clear that we cannot skip the equations for the prospective coordinates. Likewise, the Euler-Lagrange equations need to be presented in the abstract form, since these are the equations that are eventually transformed into the “model”. We tried to give the basic intuition for this in the main text. As we explained above, the equations asked to be skipped represent the essence of the proposal. It is about how to derive a model equations.

Moreover, we give more explanations in the Methods to understand the derivations, and we refer to the specifically sections in the SI for further details. We are aware that a full understanding of the theory requires some basic knowledge of the calculus of variation.

We are hesitating to write in the Introduction what type of knowledge is required to understand the paper. An understanding can be on various levels. Moreover, the materials that are considered to be helpful depend on the background. While for some it is a Youtube, for some Wikipedia, and for others it is a textbook where specific ingredients can be extracted. But we do cite two textbooks in the Results and more in the SI, Sect. F, when referring to the principle of least action in physics and the mathematics, including weblinks.

Minor commentsEq.3: The Authors refer to this equation as a Lagrangian. Could you please clarify why? Is the logic to minimize the energy subject to a constraint that Cost = 0?

Thanks for asking. The cost is not really a constraint, it is globally minimized, in parallel steps. We are explaining this right after Eq. 3. “We `prospectively' minimize L locally across a voltage trajectory, so that, as a consequence, the local synaptic plasticity for W will globally reduce the cost along the trajectory (Theorem 1 below).”

We were adding two sentence that explain why this function in Eq. 3 is called a Lagrangian: “While in classical energy-based approaches L is called the total energy, we call it the `Lagrangian' because it will be integrated along real and virtual voltage trajectories as done in variational calculus (leading to the Euler-Lagrange equations, see below and SI, Sect. F)”

p.4, below Eq. 5 - Please explain the rationale behind NLA, i.e. why is it beneficial that "the trajectory u˜(t) keeps the action A stationary with respect to small variations δu˜"? I guess you wish to minimize L integrated over time, but this is not evident from the text.

Hmm, yes and no. We wish to minimize the cost, and on the way there minimize the action. Since the global minimization of C is technically difficult, one looks for stationary trajectory as defined in the cited sentence, while minimizing L with respect to W, to eventually minimize the cost.

In the text we now explain after Eq. 5:

“The motivation to search for a trajectory that keeps the action stationary is borrowed from physics. The motivation to search for a stationary trajectory by varying the near-future voltages ũ instead of u is assigned to the evolutionary pressure in biology to 'think ahead of time'. To not react too late, internal delays involved in the integration of external feedback need to be considered and eventually need to be overcome. In fact, only for the 'prospective coordinates' defined by looking ahead into the future, even when only virtually, will a real-time learning from feedback errors become possible (as expressed by our Theorems below).”

Bottom of page 8. The authors say that in the case of single equilibrium and strong nudging the model reduced to the Least Control Principle. Does it also reduce to Predictive coding for supervised learning? If so, it would be helpful to state so.

Yes, in this case the prediction error in the apical dendrite becomes the one of predictive coding. We are stating this now right at the end of the cited sentence:

“In the case of strong nudging and a single steady-state equilibrium, the NLA principle reduces to the Least-Control Principle (Meulemans et al., 2022) that minimizes the mismatch energy E^M for a constant input and a constant target, with the apical prediction error becoming the prediction error from standard predictive coding (Rao & Ballard, 1999).”

In the Discussion we also added a further point (iv) to compare the NLA principle with predictive coding. Both “improve” the sensory representation, but the NLA does in favor of an output, and the predictive coding in favor of the sensory prediction itself (see Discussion).

Whenever you refer to supplementary materials, please specify the section, so it is easier for the reader to find it.

Done. Sorry to not have done it earlier. We are now also indicate specific sections when referring to the Methods.

**Reviewer #2 (Recommendations For The Authors):**
There are no major issues with this article, but I have several considerations that I think would greatly improve the impact, clarity, and validity of the claims.(1) Unifying the narrative. There are many many ideas put forward in what feels like a deluge. While I appreciate the enthusiasm, as a reader I found it hard to understand what it was that the authors thought was the main breakthrough. For instance, the abstract, results, introduction, and discussion all seem to provide different answers to that question. The abstract seems to focus on the motor error idea. The introduction seems to focus on the novel prospective+predictive setup of the energy function. The discussion lists the different perks of the theory (delay compensation, moving equilibrium, microcircuit) without referring to the prospective+predictive setup of the energy function.

Thanks much for these helpful hints. Yes, the paper became an agglomerate of many ideas, also own to the fact that we wish to show how the NLA principle can be applied to explain various phenomenology in neurosicence. We now simplified the narrative to this one point of providing a novel theoretical framework for neuroscience, and explaining why this is novel and why it “suddenly works” (the prospective minimization of the energy).

As you can see from the dominating red in the revised pdf, we did fully rewrite Abstract, Introduction and Discussion under the narrative of the NLA and prospective coding.

(2) Laying out the organization of the notation clearly. There are quite a few subtle distinctions of what is meant by the different weight matrices (omnibus matrix then input vs recurrent then layered architecture), different temporal horizon formalisms (bar, not bar, tilde), different operators (L, curly L, derivative version, integral version). These different levels are introduced on the fly, which makes it harder to grasp. The fact that there are many duplicate notations for the same quantities does not help the reader. For instance u_0 becomes equal to u_N at one point (above Eq 25). Another example is the constant flipping between integrated and 'current input' pictures. So laying out the multiple layers early, making a table or a figure for the notation, or sticking with one level would help convey the idea to a wide readership.

Thanks for the hints. We included the table you suggested, but put it to the SI as it became a full page itself. We banned the curly L abbreviating the look-ahead operator.

The “change of notation” you are alluding to is tricky, though. In a recurrent layer, the index of the output neuron is called o. In a forward network with N layer, the index of the output neurons becomes the last layer N. One has to introduce the layer index l anway for the deeper layers l < N, and we found it more consistent to explain that, while switching from the recurrent to the forward network, the voltage of the output layer becomes now u_o = u_N. There are more of these examples, like the weight matrix W splitting into a intrinsic network part W_net across which errors backpropagate, and a part conveying the input, W_in, that has to be excluded when writing the backpropagation formula for general networks. Again, in the case of the feedforward networks, the notation reduces to W_l, with index l coding for the layer. Presenting the general approach and a specific example may appear as we would duplicate notations – we haven’t found a solution here.

(3) Separate the algorithm from the implementation level. I particularly struggled with separating the ideas that belonged to the algorithm level (cost function, optimization objectives) and the biophysics. The two are interwoven in a way that does not have to be. Particularly, some of the normative elements may be implemented by other types of biophysics than the authors have in mind. It is for this reason that I think that separating more clearly what belongs to the implementation and algorithm levels would help make the ideas more widely understood. On this point, a trigger point for me was the definition of the 'prospective input rates' e_i, which comes in the second paragraph.

We are very sorry to have made you thinking that the 'prospective input rates' would be e_i. The prospective input rates are r_i. The misunderstanding likely appeared by an unclear formulation from our side that is now corrected (see first and second paragraph of the Results where we introduce r_i and e_i).

From a biophysical perspective, it is quite arbitrary to define the input to be the difference between the basal input and the somatic (prospective) potential. It sounds like it comes from some unclear normative picture at this point. But the authors seem to have in mind to use the fact that the somatic potential is the sum of apical and basal input, that's the biophysical picture.

We hope to have disentangled the normative and biophysical view in the 2nd and 3rd paragraph of the Results, respectively. We introduce the prospective error ei as abstract notion in the first paragraph, while explaining that it will be interpreted as somato-dendritic mismatch error in neuron I in the next paragraph. The second paragraph contains the biophysical details with the apical and basal morphology.

(4) Experts and non-expert would appreciate an explanation of why/how the choice of state variables matters in the NLA. The prospective coding state variables cannot be said to be the naïve guess. Why does the simple u, dot{u} not work as state variables applied on the same energy function, as would be a naïve application of the Lagrangian ideas?

We are very glad for this hint to present an intuition behind the variation of the action with respect to a prospective state, instead of the state itself. The simple L(u, dot{u}) does not work because one does not obtain the first-order voltage dynamics compatible with the biophysics. We made an effort to explain the intuition to non-experts and experts in an additional paragraph right after presenting the voltage and error dynamics (Eq. 7 on page 4).

Here is how the paragraph starts (not displaying the formulas here):

“From the point of view of theoretical physics, where the laws of motion derived from the least-action principle contain an acceleration term (as in Newton's law of motion, like … for a harmonic oscillator), one may wonder why no second-order time derivative appears in the NLA dynamics. As an intuitive example, consider driving into a bend. Looking ahead in time helps us to reduce the lateral acceleration by braking early enough, as opposed to braking only when the lateral acceleration is already present. This intuition is captured by minimizing the neuronal action A with respect to the discounted future voltages ũi instead of the instantaneous voltages ui.

Keeping up an internal equilibrium in the presence of a changing environment requires to look ahead and compensate early for the predicted perturbations.

Technically, …”

More details are given in the Methods after Eq. 20. Moreover, in the last part of the SI, Sect. F, we have made the link to the least-action principle in physics more explicitly. There we show how the voltage dynamics can be derived from the physical least-action principle by including the Rayleigh dissipation (Eq. 92 and 95).

(5) Specify that the learning rules have not been observed. Though the learning rules are Hebbian, the details of the rules have not to my knowledge been observed. Would be worth mentioning as this is a sticking point of most related theories.

We agree, and we do now explicitly write in the Discussion that the learning rule still awaits to be experimentally tested.

1. Some relevant literature. Chalk et al. PNAS (2018) have explored the relationship between temporal predictive coding and Rao & Ballard predictive coding based on the parameters of the cost function. Harkin et al. eLife (2023) have shown that 'prospective coding' also takes place in the serotonergic system, while Kim ... Ma (2021) have put forward similar ideas for dopamine, both may participate in setting the cost function. Instantaneous voltage propagation is also a focus of Greedy et al. (2023). The authors cite Zenke et al. for spiking error propagation, but there are biological references to that end.

Thanks much for these hints. We do now cite the book of Gerstner & Kistler on spiking neurons, and more specifically the spike-based approach for learning to represent signals (Brendel, .., Machens, Denève, PLoS CB, 2020). Otherwise, we had difficulties to incorporate the other literature that seems to us not directly related to our approach, even when related notions come up like predictive coding and temporal processing in Chalk et al. (2018), where various temporal coding schemes coding efficiency is studied as a function of the signal-to-noise ratio, or the apical activities in Greedy et al. (2022), where bursting, multiplexing and synaptic facilitation arises. We found it would confuse more than it would help if we would cite these papers too (we do already cite 95 papers).

(7) In the main text, theorem two is presented as proof without assumptions on the level of nudging, but the actual proof uses strong assumptions in that respect, relying on numerical ad hoc observations for the general case.

Thanks for pointing this out. We agree it is a better style to state all the critical assumptions in Theorem itself, rather than deferring them to the Methods. We now state: “Then, for suitable top-down nudging, learning rates, and initial conditions, the ….weights …evolve such that…”.

(8) In the discussion regarding error-backpropagation, it seems to me that it could be clarified that the current algorithm asks for a weight alignment between FF and FB matrices as well as between FB and interneuron circuit matrices. Whether all of these matrices can be learned together remains to be shown; neither Akrout, Kunin nor Max et al. have shown this explicitly. Particularly when there are other inputs to the apical dendrites from other areas.

Yes, it is difficult to learn to align all in parallel. Nevertheless, our simulations in fact do align the lateral and vertical circuits, at is also claimed in Theorem 2. Yet, as specified in the theorem, “for suitable learning rates” (that were all the same, but were commonly reduced after some training time, as previously explained in the Methods, Details for Fig. 5).

In the Discussion we now emphasis that, in general, simulating all the circuitries jointly from scratch in a single phase is tricky. We write:

“A fundamental difficulty arises when the neuronal implementation of the Euler-Lagrange equations requires an additional microcircuit with its own dynamics. This is the case for the suggested microcircuit extracting the local errors. Formally, the representation of the apical feedback errors first needs to be learned before the errors can teach the feedforward synapses on the basal dendrites. We showed that this error learning can itself be formulated as minimizing an apical mismatch energy. What the lateral feedback through interneurons cannot explain away from the top-down feedback remains as apical prediction error.

Ideally, while the network synapses targetting the basal tree are performing gradient descent on the global cost, the microcircuit synapses involved in the lateral feedback are performing gradient descent on local error functions, both at any moment in time.

The simulations show that this intertwined system can in fact learn simultaneously with a common learning rate that is properly tuned. The cortical model network of inter- and pyramidal neurons learned to classify handwritten digits on the fly, with 10 digit samples presented per second. Yet, the overall learning is more robust if the error learning in the apical dendrites operates in phases without output teaching but with corresponding sensory activity, as may arise during sleep (see e.g. Deperrois et al., 2022 and 2023).”

(9) The short-term depression model is assuming a slow type of short-term depression, not the fast types that are the focus of much recent experimental literature (like Campagnola et al. Science 2022).This assumption should be specified.

Thanks for hinting to this literature that we were not aware of. We are now citing the releaseindependent plasticity (Campagnola et al. 2022) in the context of our synaptic depression model.

(10) There seems to be a small notation issue: Eq 21 combines vectors of the size of the full network (bar{e}) and the size of the readout network (bar{e}star).

Well, for notational convenience we set the target error to e*=0 for non-output neurons. This way we can write the total error for an arbitrary network neuron as the sum of the backpropagated error plus the putative target error (if the neuron is an output neuron). Otherwise we would always have to distinguish between network neuron that may be output neurons, and those that are not. We did say this in the main text, but are repeating it now again right after Eq. 21. -- Notations are often the result of a tradoff.